# Ultrafast single-molecule imaging reveals focal adhesion nano-architecture and molecular dynamics

Takahiro K. Fujiwara[1], Taka A. Tsunoyama[2], Shinji Takeuchi[3], Ziya Kalay[1], Yosuke Nagai[3], Thomas Kalkbrenner[4], Yuri L. Nemoto[2], Limin H. Chen[2], Akihiro C.E. Shibata[1], Kokoro Iwasawa[1], Ken P. Ritchie[5], Kenichi G.N. Suzuki[1,6], and Akihiro Kusumi[1,2]

Using our newly developed ultrafast camera described in the companion paper, we reduced the data acquisition periods required for photoactivation/photoconversion localization microscopy (PALM, using mEos3.2) and direct stochastic reconstruction microscopy (dSTORM, using HMSiR) by a factor of ≈30 compared with standard methods, for much greater view-fields, with localization precisions of 29 and 19 nm, respectively, thus opening up previously inaccessible spatiotemporal scales to cell biology research. Simultaneous two-color PALM-dSTORM and PALM-ultrafast (10 kHz) single fluorescent-molecule imaging-tracking has been realized. They revealed the dynamic nanoorganization of the focal adhesion (FA), leading to the compartmentalized archipelago FA model, consisting of FA-protein islands with broad diversities in size (13–100 nm; mean island diameter ≈30 nm), protein copy numbers, compositions, and stoichiometries, which dot the partitioned fluid membrane (74-nm compartments in the FA vs. 109-nm compartments outside the FA). Integrins are recruited to these islands by hop diffusion. The FA-protein islands form loose ≈320 nm clusters and function as units for recruiting FA proteins.

## Introduction

In the companion paper (Fujiwara et al., 2023), we report the development of an ultra-high-speed camera system that has enabled the fastest single fluorescent-molecule imaging and tracking (SFMI) to date. Our camera system achieved a 100-µs resolution with a 20-nm localization precision for single Cy3 molecules for a frame size of 14 × 14 µm² (256 × 256 pixels) and a 33-µs resolution with a 34-nm localization precision for a frame size of 7.1 × 6.2 µm² (128 × 112 pixels; Table 1 in the companion paper; faster than video rate by factors of 330 and 1,000, respectively). Our ultrafast SFMI technique, which uses this ultrafast camera and the selected fluorophores, Cy3 and tetramethyrhodamine (TMR), has successfully detected the fast hop diffusion of membrane molecules in both the apical and basal plasma membrane (PM). This detection was previously only achievable in the apical PM by using less preferable 40-nm gold probes.

In this paper, we present the application of the developed camera system for dramatically reducing the data acquisition time required for single-molecule localization microscopy (SMLM), including photoactivation/photoconversion localization microscopy (PALM) and direct stochastic optical reconstruction microscopy (dSTORM). The advent of SMLM has greatly improved the spatial resolution of fluorescence microscopy, but it came with the cost of temporal resolution.

Obtaining a single SMLM image typically requires 250–20,000 frames, which means that the data acquisition time, and thus the time resolution of SMLM, is 1–10 min (Jones et al., 2011; Lelek et al., 2021). Consequently, the SMLM observations have generally been limited to fixed cells, precluding observations of the time-dependent changes of the cellular structures in live cells (Shcherbakova et al., 2014; Liu et al., 2015b; Nicovich et al., 2017; von Diezmann et al., 2017; Baddeley and Bewersdorf, 2018; Sigal et al., 2018). For more detailed discussions on this point, see Lelek et al. (2021). The objective of our present research was to overcome this critical limitation in the application of SMLM to cell biology by reducing the data acquisition times by using the developed ultrafast camera system. By making SMLM applicable to live cells, we aim to observe the time-dependent changes of subcellular structures at the SMLM spatial resolution.

In the present research, based on the developed ultrafast camera system, we established the optimal conditions for accelerating the data acquisition for PALM and dSTORM up to 1 kHz, ≈30× faster than normal video rate, with 29 and 19 nm single-molecule localization precisions, respectively, for a view-field as large as 640 × 640 pixels ≈35.3 × 35.3 µm², which can often encompass an entire live cell. By employing mEos3.2 and HMSiR probes (Zhang et al., 2012; Uno et al., 2014) for ultrafast PALM and dSTORM, respectively, the data acquisition time has

........................................................................................................................................................................................................
[1]Institute for Integrated Cell-Material Sciences (WPI-iCeMS), Kyoto University, Kyoto, Japan; [2]Membrane Cooperativity Unit, Okinawa Institute of Science and Technology Graduate University (OIST), Okinawa, Japan; [3]Photron Limited, Tokyo, Japan; [4]Carl Zeiss Microscopy GmbH, Jena, Germany; [5]Department of Physics and Astronomy, Purdue University, West Lafayette, IN, USA; [6]Institute for Glyco-core Research, Gifu University, Gifu, Japan.

Correspondence to Akihiro Kusumi: akihiro.kusumi@oist.jp.

been shortened from 1–10 min to 0.25–20 s, even for the largest view-field to date. Thus, simultaneous two-color ultrafast PALM and dSTORM can now be readily performed for imaging entire live cells with nanoscale spatial resolutions, provided the structure of interest does not appreciably change during the data acquisition periods of 0.25–20 s.

The 1-kHz data acquisition rate is not limited by the developed camera but by the availability of fluorophores. If fluorescent probes that can be excited and photobleached faster become available, then a data acquisition frame rate of 10 kHz (or even 45 kHz) would be possible with the developed camera system. This would further shorten the time required to obtain a single SMLM image to 25 ms–2 s (5.5 ms–0.44 s); i.e., under optimal conditions, the video-rate SMLM would become possible. Therefore, we emphasize here that, with the developed ultrafast camera, the availability of suitable fluorophores is the time-limiting factor for SMLM rather than the instrumentation.

Using the developed ultrafast SMLM methods, we examined the nanoscale architecture of the focal adhesion (FA), a micron-scale structure in the basal PM that serves as a scaffold for cellular attachment to and migration in/on the extracellular matrix (Fig. 1, A–C; Parsons et al., 2010; Gardel et al., 2010; Humphries et al., 2019; Yamada and Sixt, 2019; Doyle et al., 2022; Kanchanawong and Calderwood, 2023). Furthermore, we observed the ultrafast single-molecule dynamics in the nanoscale-resolved architecture image of the FA.

We and others previously discovered that the FA domain largely comprises the fluid membrane, in contrast to the earlier notion that the FA was a micron-scale, continuous, massive assembly of various proteins (like a large continent) or that subdivided by "canals" crisscrossing the land, like the city plan of Venice (the Venetian canal model; Saxton, 1982, 2010; Holcman et al., 2011; Shibata et al., 2012, 2013; Rossier et al., 2012; Changede and Sheetz, 2017; Tsunoyama et al., 2018; Orré et al., 2021; Fig. 1, D-a and -b). Based on these observations, we previously proposed the model of an "archipelago of FA-protein islands," in which nanoscale clusters of FA proteins are distributed in the fluid membrane (Fig. 1 D-c; Shibata et al., 2012, 2013; Rossier et al., 2012). We will more precisely define FA-protein islands later in this report.

In the present study, we applied the developed ultrafast single-molecule imaging methods, including ultrafast PALM, dSTORM, simultaneous two-color PALM-dSTORM, and simultaneous PALM-SFMI, to examine the molecular architecture and dynamics of the FA in live cells. We have revealed the nanoscale archipelago architecture of FA-protein islands, including those containing integrins β1 and β3, paxillin, FAK, talin, and vinculin. Our results indicate the broad diversity of FA-protein islands in terms of size (mostly in the range of 13–100 nm, with a mean island diameter of ≈30 nm), protein copy numbers, compositions, and stoichiometries of individual FA-protein islands. This broad diversity of the FA-protein islands might be critical for the FA's mechanotransduction function, responding to various types of forces and force loading rates. In addition, the FA-proteins are likely to exist in the FA as both oligomers and monomers.

Our analyses revealed that the fluid-membrane part of the FA, outside the FA-protein islands, is compartmentalized, similar to the bulk basal PM outside the FA. However, the compartment size is smaller (74 vs. 109 nm in the bulk basal PM) and the non-FA transmembrane protein transferrin receptor (TfR) undergoes intercompartmental hop movements at an average of every 36 ms, as compared with 24 ms in the bulk basal PM. The FA's key transmembrane receptor integrin β3 undergoes hop diffusion in the FA's fluid membrane region to reach the FA-protein islands, where it becomes immobilized for various durations. Therefore, by integrating the FA membrane compartmentalization with the model shown in Fig. 1 D-d, we propose the "Compartmentalized archipelago model of FA-protein island clusters and oligomers" (Fig. 1 D-e).

## Results

### Development of ultrafast PALM for practical live-cell observations

PALM imaging and other SMLM methods of live cells require a good balance among the time resolution (data acquisition frame number and rate), view-field size, and single-molecule localization precision (Shroff et al., 2008; Jones et al., 2011; Huang et al., 2013; see the bottom six rows in Table 1 in the companion paper). Therefore, we first established the optimal conditions for performing ultrafast PALM in live cells by using the photoconvertible fluorescent protein mEos3.2 (Zhang et al., 2012), which was fused to the C-terminus of mouse caveolin-1 (caveolin-1-mEos3.2), expressed at very low levels (<<1 caveolin-1-mEos3.2 molecule/caveola), and immobile in live human T24 epithelial cells.

We found that the duration of a single on-period (bright period) of mEos3.2 can be reduced to 1.2 ms or even less by increasing the 561-nm excitation laser intensity at the specimen plane to 100 $\mu$W/$\mu$m$^2$ or more (note that 1 $\mu$W/$\mu$m$^2$ = 0.1 kW/cm$^2$) using the total internal reflection (TIR) illumination mode (Fig. 2, A-a and B). However, the number of photons emitted by a single mEos3.2 molecule during an on-period reaches a maximum at a laser intensity of 30 $\mu$W/$\mu$m$^2$ at the sample plane (Fig. 2, A-b and c). This bell-shaped power dependence is not unique to mEos3.2, as Lin et al. (2015) reported that Alexa647 exhibits a similar trend. At a laser intensity of 30 $\mu$W/$\mu$m$^2$, the single-molecule localization precision is highest, reaching 29 ± 0.22 nm (Fig. 2 A-d and Fig. S1). Therefore, we employed this laser intensity, at which the on-period is 2 ms, and used a camera frame rate of 1 kHz for PALM imaging throughout this study because frame times much shorter than 2 ms or frame rates much higher than 0.5 kHz will not be useful, although the developed camera can reach 10–30 kHz for possibly faster PALM. The mean number of on-periods before mEos3.2 became photobleached was 1.4 based on the model of a monomeric blinking fluorophore (Hummer et al., 2016; see Fig. 2 A-e and its legend).

Although mEos3.2 is currently one of the best PALM probes available, it is impossible to perform PALM imaging faster than this data acquisition frame rate of 1 kHz. To achieve faster PALM imaging, we will need photoconvertible or photoactivatable fluorescent molecules with on-periods shorter than 2 ms. Despite this limitation, we emphasize here that, with the developed

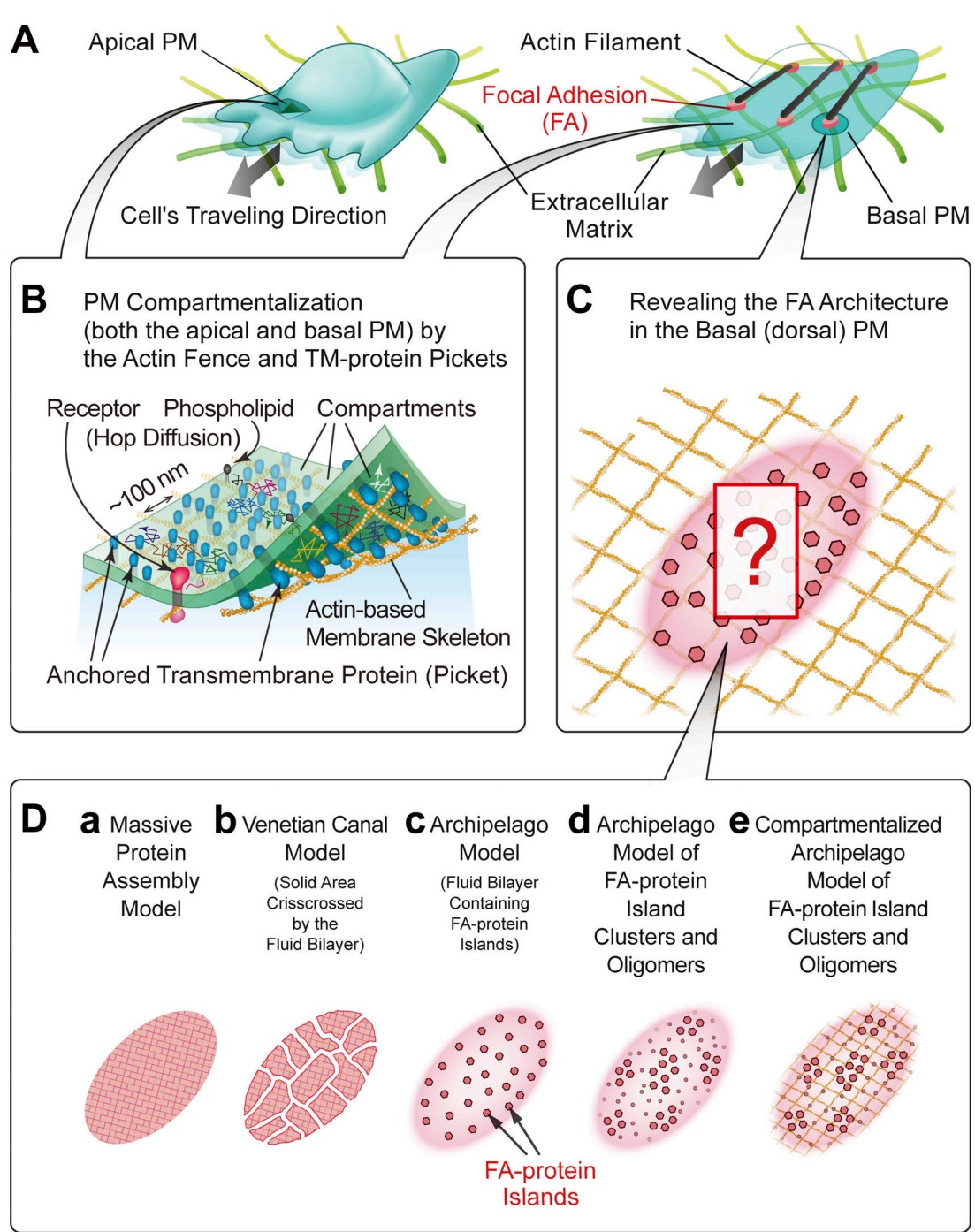

Figure 1. **The objectives of this study are twofold: (1) to establish the optimal conditions for ultrafast PALM of mEos3.2, ultrafast dSTORM of HMSiR, and their simultaneous imaging and (2) to apply these methods and ultrafast SFMI (its development is described in the companion paper) to elucidate the FA architecture and protein dynamics in the FA.** The second purpose, together with the previous results, is summarized in the figure. **(A and B)** Both the apical (dorsal) PM (A, left) and basal (ventral) PM (A, right) are compartmentalized in a nearly identical manner by actin-based membrane-skeleton meshes (fences; brown mesh in B) and rows of transmembrane-protein pickets anchored to and aligned along the actin fence (blue molecules in B), which induce the hop diffusion of virtually all membrane molecules in both the apical and basal PM (B). See the companion paper for these results. **(C)** Three fundamental questions about the FA molecular organization addressed here are as follows: (1) the characteristics of the FA-protein clusters/oligomers/islands, (2) the higher-order organizations of the FA-protein clusters/oligomers/islands, and (3) the possibility that the fluid membrane part in the FA is compartmentalized, like the bulk basal PM. For details, see D. **(D)** The three models of the molecular organization in the FA proposed previously (a–c) and the two new models (d and e) proposed here. The new models incorporate the formation of loose clusters of FA-protein islands (d) plus the compartmentalization of the fluid membrane part in the FA (e).

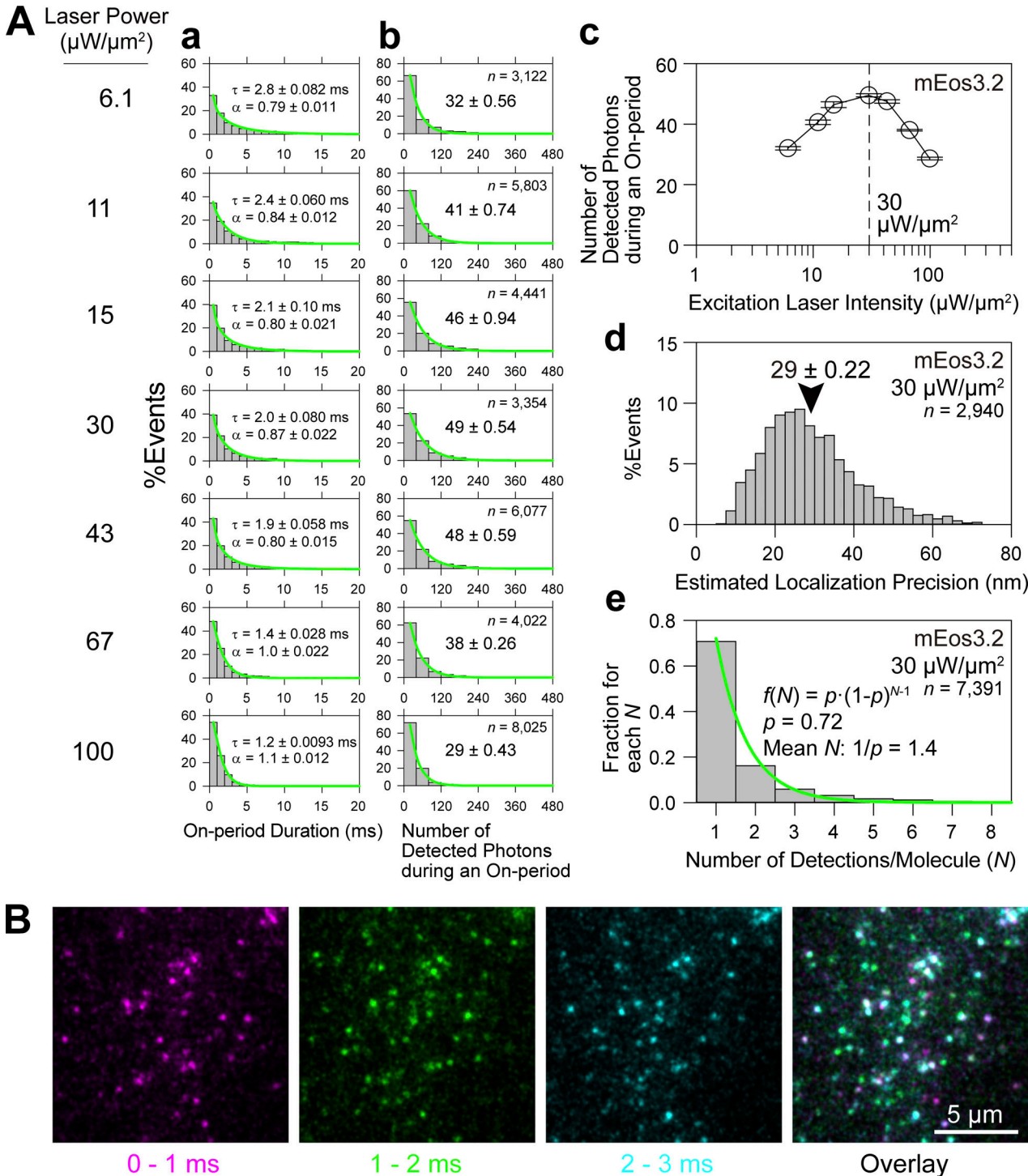

**Figure 2. Optimizing the excitation laser power density and the data acquisition frame rate for the ultrafast PALM imaging of mEos3.2 in living cells.**
**(A)** The mean number of photons emitted from a mEos3.2 molecule during an on-period is maximized (49 ± 0.54 photons) at a laser power density of 30 μW/μm² at the sample plane (c), providing a single-molecule localization precision of 29 ± 0.22 nm (d) with an on-period of ≈ 2 ms (a) and a mean number of on-periods before photobleaching of 1.4 (e). Caveolin-1-mEos3.2 expressed at very low levels (<<1 caveolin-1-mEos3.2 molecule/caveola) in the basal PM of T24 cells was imaged using live cells. (a) Histograms of individual fluorescent on-periods (with a gap closing of 1 frame) obtained at various laser power densities at the sample (indicated on the right of b). They could be fitted by stretched exponential functions $\varphi(t) = \varphi_0 e^{-(t/\tau)^\alpha}$, where $\varphi_0$ is the prefactor, $\alpha$ is the stretching exponent, and $\tau$ is the time constant (Morimatsu et al., 2007; mean ± SEM; SEM was determined as a 68.3% confidence limit for the fitting, which is also the same in b; the numbers of spots observed are the same as those indicated in the boxes in b). (b) Distributions of the numbers of detected photons during an on-period. The histograms could be fitted with single exponential decay functions, with the decay constants providing the mean numbers of detected photons during an on-period. n = number of observed spots. (c) Summary plot for the results in b, showing that the laser power density of 30 μW/μm² at the sample plane provides the maximal number of detected photons during an on-period of a single mEos3.2 molecule (49 ± 0.54 photons). (d) The mean localization precision for mEos3.2 in the basal PM was 29 ± 0.22 (SEM) nm under the optimized laser excitation conditions of 30 μW/μm². The localization precision for

each on-period of a single mEos3.2 molecule was estimated using the theoretical equation derived by Mortensen et al. (2010), employing an "excess noise" factor ($F$) of 1.2 determined for the developed camera system (see Fig. S2 of the companion paper). (e) The distribution of the number of on-events (localizations) for a single mEos3.2 molecule ($N$) at a laser power density of 30 µW/µm². Each detection was found by examining the proximity of the spots recorded at different frames, with a cutoff time of 3 s (Durisic et al., 2014) and a cutoff distance of 82 nm ($\sqrt{2} \times 2 \times$ [mean localization precision for mEos3.2 = 29 nm]). The histogram could be fitted well (green curve) with the geometric function $f(N) = p \bullet (1 - p)^{N-1}$ based on the model for a monomeric blinking fluorophore by Hummer et al. (2016), yielding the P value (fluorophore bleaching probability) = 0.72 and the mean number of detections (on events)/molecule ($1/p$) = 1.4. **(B)** The data acquisition at 1 kHz is nearly an optimal frame rate for PALM imaging using mEos3.2 as a probe. The figure shows typical consecutive single-frame images of caveolin-1-mEos3.2 molecules in the basal PM acquired every 1 ms (1 kHz) using an observation laser power density of 30 µW/µm². Based on the number of detections (localizations), caveolin-1-mEos3.2 was found to be expressed at 42 ± 5.2 copies/caveola, which only includes the fluorescent mEos3.2 and not the non-fluorescent mEos3.2, but was normalized by the overcounting of 1.4; $n$ = 15). While many spots exist in only a single image frame, some spots appear in two or three images (spot colors are changed every 1 ms). Using these spot images, the SD of the Gaussian spot profile was determined to be 129 ± 1.3 nm ($n$ = 50 caveolin-1-mEos3.2 molecules). This value was used for reconstructing the diffraction-limited images.

camera, the instrument is no longer the limiting factor for the data acquisition rate and the probes have become the rate-limiting factor. This bottleneck is similar to the challenge faced in achieving faster SFMI. The camera and instrument are no longer the limitations for achieving faster SFMI, and now the availability of fluorescent probes has become the limit.

However, we emphasize that even the 1-kHz frame rate is significantly faster than the video rate, shortening the PALM data acquisition time by a factor of 33 as compared to video-rate acquisition. For acquisitions of 250–20,000 frames commonly employed in SMLM (Jones et al., 2011; Lelek et al., 2021), the data acquisition time can be shortened to 0.25–20 s, which is a reasonably fast time frame to observe the morphological changes of subcellular structures in living cells. With the future advent of better photoconvertible/photoactivatable molecules, it may be possible to use 10–30 kHz, rather than 1 kHz, for data acquisition. This will allow us to obtain PALM images 10–30 times faster than the rate achieved here, potentially enabling PALM imaging at the video rate (for the acquisition of 330–1,000 frames). We attempted to use photoactivatable organic fluorophores but their on-periods were found to be even longer.

The single-molecule localization precision for mEos3.2 using scientific complementary metal-oxide-semiconductor (sCMOS) sensors, which are more commonly used in fluorescence microscopy, is 22 nm (256 × 256 pixels at a frame rate of 0.6 kHz; Huang et al., 2013). This is better by a factor of 1.3 as compared with the developed camera system (29 ± 0.22 nm; Fig. 2 A-d). The superior localization precisions obtained with the sCMOS sensors are primarily due to their higher quantum efficiencies of prevalent sCMOS sensors (70–80%) as compared with that of the photosensor in the image intensifier used here (≈40%; consistent with $[70/40]^{1/2} \approx 1.3$).

Typical PALM images of caveolin-1-mEos3.2 expressed at much higher levels (42 ± 5.2 caveolin-1-mEos3.2 molecule/caveola on average; $n$ = 15) in live and chemically-fixed T24 cells, generated in a reconstruction time window of 1 s (data acquisition at 1 kHz × 1,000 frames; 10 × 10 µm² in 181 × 181 pixels), are shown in Fig. 3, A-a and B-a. The obtained PALM images of caveolae are consistent with the known size of a caveola (60–80 nm in diameter; Parton and del Pozo, 2013), given the localization precision of 29 nm. The whole caveola occasionally moves on the PM (compare Fig. 3 A with Fig. 3 B), and when migration occurs, it can suddenly shift by ≈100 nm or so in <0.33 s, suggesting movement from one actin-induced compartment to an

adjacent one on the basal PM, as detected by ultrafast SFMI (Fujiwara et al., 2023).

## Ultrafast PALM imaging of large PM areas in live cells
The PALM data acquisition rate was limited to 1 kHz when using mEos3.2. However, the newly developed camera enables data acquisition with a frame size as large as 640 × 640 pixels (35.3 × 35.3 µm² with a pixel size of 55.1 nm) at 1 kHz, while maintaining the same single-molecule localization precision of 29 ± 0.22 nm. Therefore, the large majority of the basal PM, and often the entire basal PM, in a live cell can be imaged by PALM after 10 s of the 10,000-frame data acquisition. A PALM image of mEos3.2 fused to human paxillin, a representative FA structural protein, expressed on the basal PM of live T24 cells and obtained after the data acquisition at 1 kHz for 10,000 frames (10 s) is shown in Fig. 4 A (also see Video 1; compare with the diffraction-limited image in Fig. 4 B). This image size is the largest ever reported for PALM data obtained with this level of localization precision in the time scale of 10 s, which is useful for observing live cells (for comparisons with previous results, see Table 1 in the companion paper).

Paxillin was selected as our first target protein among the FA proteins because it is an important scaffolding protein that recruits structural and signaling molecules involved in cell movement (López-Colomé et al., 2017). The PALM images of live T24 cells expressing mEos3.2-paxillin (clonally selected after transfection) exhibited similar FA areas to those reported previously in both live cells (Fig. 4, C and D; Shroff et al., 2008; Orré et al., 2021) and fixed cells (Shroff et al., 2007; Changede and Sheetz, 2017; Deschout et al., 2017). In these T24 cells, the amount of expressed mEos3.2-paxillin was roughly estimated to be 0.9× of that of endogenous paxillin in non-transfected cells, and thus the total paxillin amount will be ≈1.9× after transfection, assuming that the endogenous paxillin expression level remains unchanged after the expression of mEos3.2-paxillin (see Fig. S2 A and its legend).

## The FA region is dotted with paxillin-enriched islands of 33-nm mean diameter in T24 epithelial cells
We utilized the SR-Tesseler software based on the Voronoï segmentation analysis (Levet et al., 2015) to quantitatively analyze the paxillin PALM images obtained in live cells. First, we identified the contours of the FA in the basal PM (Fig. 4, C–E; see the red contours in Fig. 4 F) using a thresholding polygon density factor of 1.45 (the paxillin copy number density inside the FA is

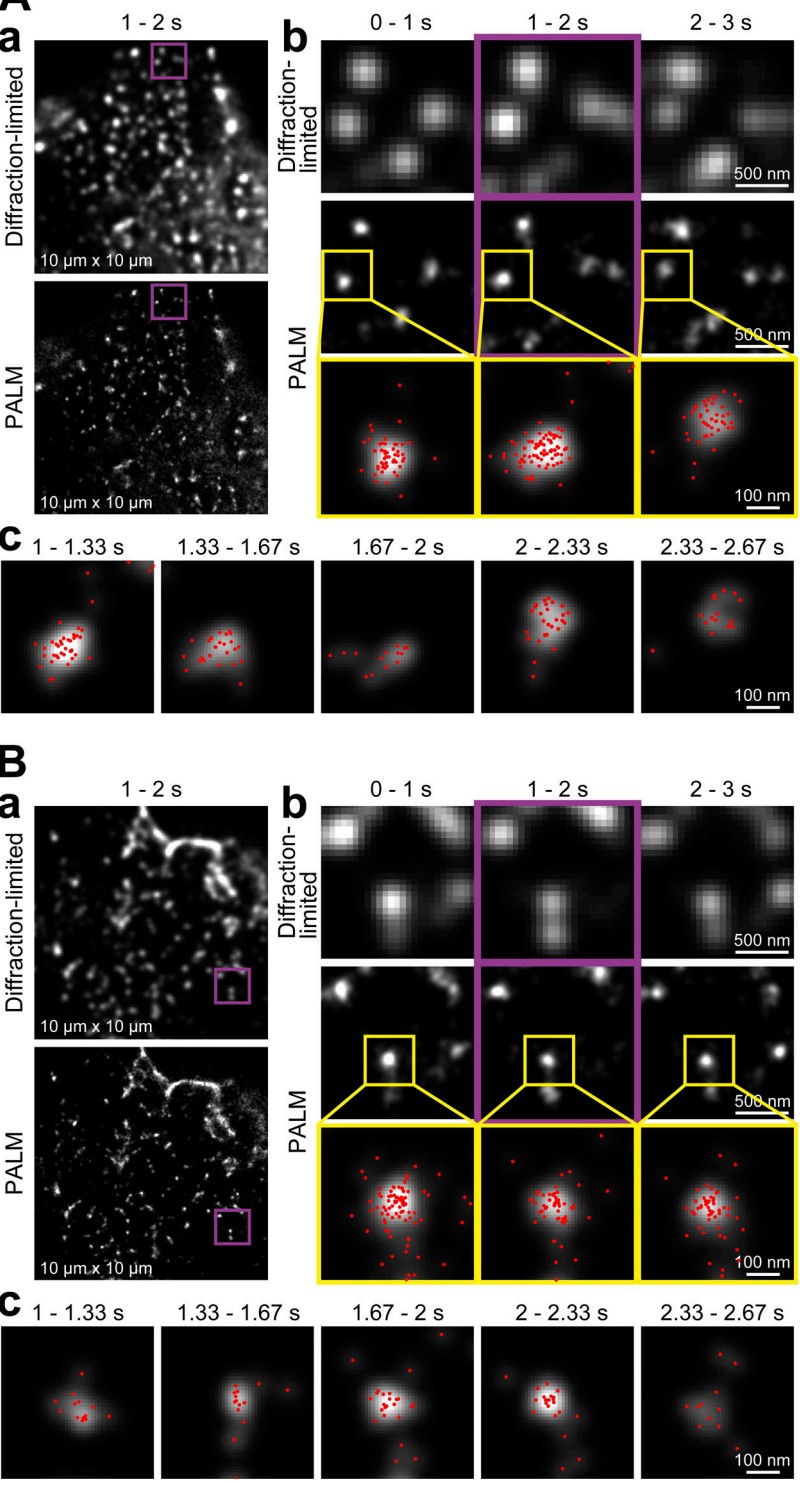

**Figure 3. Ultrafast PALM reveals the shape changes, migrations, and formation/disappearance of caveolae in 3 s in live cells. (A and B)** Caveolin-1-mEos3.2 expressed in the basal PM of T24 cells at a density of 42 ± 5.2 caveolin-1-mEos3.2 molecules/caveola ($n = 15$; see the legend of Fig. 2 B) was imaged in both live (A) and fixed cells (B), using identical ultrafast PALM imaging conditions. Data acquisitions were performed at a rate of 1 kHz for 3 s (3,000 frames), and PALM images were reconstructed using the data acquired for every 1 s (= 1,000 frames; b) and every 0.33 s (= 333 frames; c). (a) Diffraction-limited and PALM images of caveolae in 10 × 10-μm² observation areas, using a data acquisition period of 1 s. For the spatial resolution of these images, see Materials and methods. (b) Enlarged images of the purple-square regions in (a) showing time-dependent changes (every 1 s). The images in the middle column (top and middle rows) are the expanded images of the purple-square regions in a, for the data acquisition between 1 and 2 s. The regions surrounded by yellow squares in the middle row are magnified in the bottom row, and the localizations of single mEos3.2 molecules determined within each 1-s period are indicated by red dots. (c) The PALM image reconstruction performed every 0.33 s for the same caveola that is shown in the bottom row in b.

≥1.45× greater in all of the polygons in the FA than the average density of the entire basal PM in the image). The FA contours determined from the PALM image were slightly smaller than those from the diffraction-limited image using the minimum cross entropy thresholding and more sensitive to much smaller FAs/paxillin clusters outside larger FAs (Fig. 4 D).

Second, we further identified the nanoscale paxillin-enriched subregions within the FA, using a thresholding polygon density

factor of 1.45 (polygons surrounded by green contours in Fig. 4 F; the reason for selecting this density factor is described in Fig. S2 B and its legend). Here, we define the subregions with diameters of ≥13 nm containing ≥6 paxillin copies as "paxillin-enriched islands," using the island diameter definition of $2 \times \sqrt{polygon\ area\ for\ the\ island/\pi}$. Within a 13-nm diameter circular region, a maximum of ≈6 paxillin molecules could be accommodated, assuming that their cross-sectional

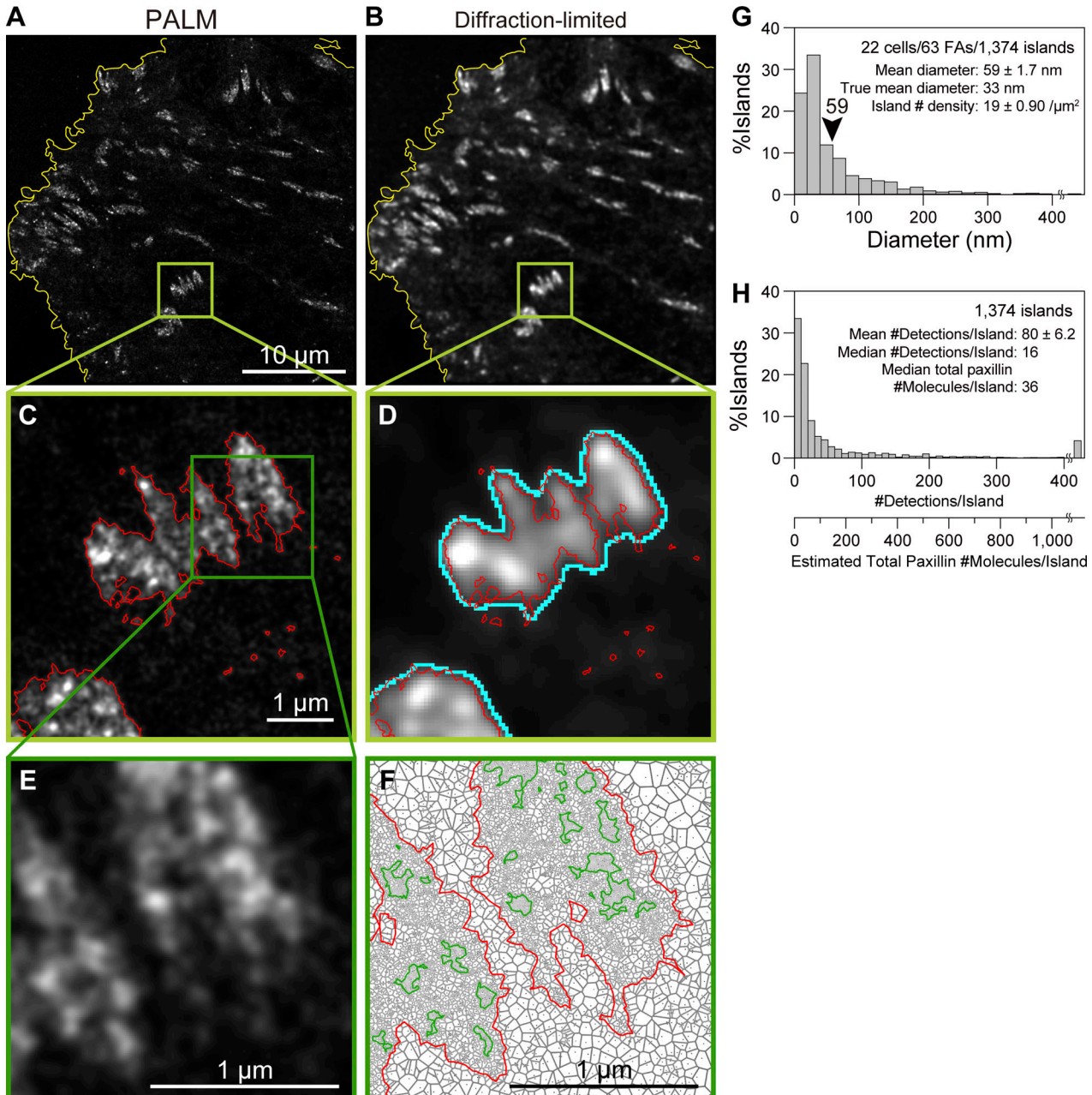

Figure 4. **Ultrafast PALM is capable of imaging a view field as large as 640 × 640 pixels (35.3 × 35.3 μm²), often encompassing almost an entire live cell, at a data acquisition rate of 1 kHz using mEos3.2 (linked to paxillin; a total of 10,000 frames obtained for 10 s), while at the level of a single FA, it reveals an archipelago architecture of paxillin-enriched islands with a 33 nm mean diamete**r. **(A and B)** Typical reconstructed PALM (A) and diffraction-limited (B) images of mEos3.2-paxillin on ≈ 2/3 of the entire basal PM of a T24 cell (data acquisition for a view field of 640 × 640 pixels ≈35.3 × 35.3 μm²). PALM image: 3,526 × 3,526 pixels with a pixel size of 10 nm. Diffraction-limited image: 640 × 640 pixels with a pixel size of 55.1 nm. The yellow contours outline the cell peripheries determined in the diffraction-limited image on the right, using Sauvola's local thresholding method with a local domain radius of 64 pixels, and k and r values of 0.5 and 128, respectively (Sauvola and Pietikäinen, 2000). Throughout this study, the reconstructions of PALM images with a pixel size of 10 nm were performed using the ThunderSTORM plugin for ImageJ (Ovesny et al., 2014) installed in the Fiji package (Schindelin et al., 2012) and Gaussian rendering with a localization precision of 29 nm (Fig. 2 A-d), whereas diffraction-limited images with a pixel size of 55.1 nm were generated by Gaussian rendering with a spot with an SD of 129 nm (Fig. 2 B). **(C and D)** Enlarged images of the domains enclosed in squares in A and B, respectively. The contours of the FAs in the PALM image shown in red (C) were determined by using the SR-Tesseler software based on Voronoï polygons (Levet et al., 2015), with a thresholding paxillin number density of 1.45. This contour is overlaid on the diffraction-limited image on the right (D). The contours of the FAs determined from the diffraction-limited image using the minimum cross entropy thresholding are shown in cyan (D). Comparison of the two FA contours indicates that the contour determined from the PALM image is slightly smaller than that determined from the diffraction-limited image and is more sensitive to much smaller FAs (or paxillin clusters outside larger FAs). **(E)** Further enlarged image from the squared domain in the PALM image in C. **(F)** The Voronoï polygon diagram of the PALM image in E, showing the contours of the FAs (red) and paxillin-enriched islands (dark green), using a thresholding paxillin number density of 1.45 for both contours. **(G)** Distribution of the paxillin-enriched island diameters obtained by the Voronoï tessellation analysis using the SR-Tesseler software. The mean diameter of the islands determined here was 59 nm, but after the correction for the effect of the 29-nm single mEos3.2-molecule localization precision on the

SR-Tesseler segmentation (Fig. S2 C), the actual mean diameter of the islands was estimated to be 33 nm. **(H)** Distribution of the number of detections (localizations) of mEos3.2-paxillin molecules per detected paxillin island. The additional x-axis scale shows the estimated total (= mEos3.2-conjugated + endogenous) paxillin copy number per island. Although these values obtained in G and H (the mean paxillin island diameter of 33 nm containing a median of 36 copies of paxillin/island) would be quite informative, they need to be interpreted with caution. First, larger islands might be collections of smaller islands that could not be resolved. Second, the islands containing many other FA proteins, but only <6 paxillin copies, would be missed by definition. Third, paxillin-enriched islands with smaller diameters would be missed even though their diameters are ≥13 nm: This was evaluated by Monte Carlo simulations, which showed that ~15, 29, and 82% of paxillin-enriched islands with diameters of 13, 20, and 30 nm, respectively, were detectable (when the protein density for the simulation was increased by a factor of 2 to 0.04 copies/nm$^2$, ≈30 and 57% of the 13- and 20-nm islands, respectively, were detectable). Approximately, 85–95% of the islands with ≥40 nm diameters were detectable (the localization precisions of 19 and 29 nm hardly affected the result). Taking these limitations into account, based on the data shown in G and H, we propose that the paxillin island diameters are mostly in the range of 13~100 nm and that the paxillin nanoclusters containing <6 paxillin copies might either exist alone or be located in the islands enriched in other FA proteins. Furthermore, smaller FA-protein islands might contain a total of ≥6 copies of various FA proteins, but the copy number of each constituent protein species might be <6.

---

diameter is ≈7 nm (The Stokes radius of BSA is 3.48 nm [Ikeda and Nishinari, 2000] and the molecular weights of paxillin and BSA are 68 and 66 kD, respectively. Therefore, although paxillin might be more elongated, we use this diameter as the 0th-order approximation.) and the fluorescent probe is located at the paxillin center when viewed from the top of the islands, warranting the consistency of these two threshold values for defining the paxillin-enriched islands. Since the location of the paxillin N-terminus, where the probe is linked, is spread over ≈20 nm in height (Kanchanawong et al., 2010), the paxillin molecules might be assembled in three dimensions. The thresholding paxillin copy number set at six copies/island is also reasonable because paxillin clusters containing ≤5 paxillin copies were found quite abundantly even outside the FAs. The presence of paxillin-enriched islands in the FA is consistent with that of the nanoscale substructures and clusters of FA proteins in the FA, as previously reported (Shroff et al., 2008; Patla et al., 2010; Rossier et al., 2012; Levet et al., 2015; Changede and Sheetz, 2017; Spiess et al., 2018; Orré et al., 2021).

By using the term "paxillin-enriched island," we allude to the possibilities that the paxillin-enriched islands might contain other FA proteins, and that other-FA-protein-enriched islands, as defined in the same way as paxillin-enriched islands, containing 0–5 copies of detected paxillin, might exist. Namely, we suggest the possibility of the existence of a variety of FA-protein islands containing various FA proteins with very flexible stoichiometries. This notion is consistent with the well-established results that FA proteins extensively interact with one another. We will revisit the concept of FA-protein islands when discussing the results shown in Fig. 6.

Using these image analysis methods based on the SR-Tesseler software, we obtained the distribution of paxillin-enriched island diameters (Fig. 4 G), providing the mean diameter of 59 ± 1.7 nm (the mean value is employed here for the comparison with the radius determined from the spatial autocorrelation of the detected PALM spots; see Fig. 5 F bottom). After correction for the effect of the 29-nm single-molecule localization precision of the PALM imaging, using the method described in Fig. S2 C, the true mean diameter of the paxillin-enriched islands was estimated to be 33 nm.

Next, we evaluated the median copy number of total (= mEos3.2-conjugated + endogenous) paxillins per island in these mEos3.2-paxillin–expressing T24 cells, which was 36. This was obtained from the median number of detections (localizations; the number of

spots in the raw PALM image) of mEos3.2-paxillin/island, which was 16 (Fig. 4 H; the much larger mean value, 80 ± 6.2, is not considered here because it is probably due to the presence of unresolved, much greater islands) using the following basic values: the mean number of on-events (detections) per mEos3.2 molecule (overcounting) = 1.4 (Fig. 2 A-e); ≈60% of mEos3.2 is fluorescent (Baldering et al., 2019); and the T24 cells used here express mEos3.2-paxillin at the level of 0.9× of that of endogenous paxillin; i.e., the total paxillin is 1.9× of the endogenous paxillin (see Fig. S2 A and its legend). The total paxillin copy number per island can then be calculated as (16/1.4/0.60)×1.9, resulting in 36.2. Since a circular island with a mean diameter of 33 nm can only accommodate up to 24 paxillin molecules in 2D (assuming the Stokes diameter of 7 nm for a paxillin molecule and the probe location in its center in 2D), as stated, the paxillin molecules might be assembled in three dimensions.

Although these values (the mean paxillin-enriched island diameter of 33 nm containing a median of 36 copies of paxillin/island) would be quite informative, they need to be interpreted with caution. First, larger islands might be collections of smaller unresolved islands. Second, the islands containing many other FA proteins, but only <6 paxillin copies, would be missed by definition. Therefore, we propose that the paxillin-enriched island diameters are mostly in the range of 13–100 nm and that the paxillin nanoclusters containing <6 paxillin copies might exist alone or be located in islands enriched in other FA proteins.

### Establishing mEos3.2-paxillin–rescued MEFs and the optimal data acquisition conditions for ultrafast dSTORM and simultaneous two-color ultrafast PALM-dSTORM

We next aimed to perform simultaneous two-color ultrafast SMLM. For these observations, we employed a previously established method that utilized mEos3.2 as the PALM probe (561-nm excitation) and HMSiR as the dSTORM probe (660-nm excitation; Takakura et al., 2017). HMSiR is suitable for live-cell dSTORM imaging because it spontaneously blinks in the normal cell culture medium without any further chemical reagents or photoswitching laser illumination, which might be toxic to cells (Uno et al., 2014). We first optimized the conditions for performing ultrafast live-cell dSTORM using HMSiR-labeled Halo-paxillin.

As specimens, we generated a clone of paxillin-null mouse embryonic fibroblast (MEF; Sero et al., 2011) rescued by the stable expression of mEos3.2-paxillin at a level of 0.64× that of

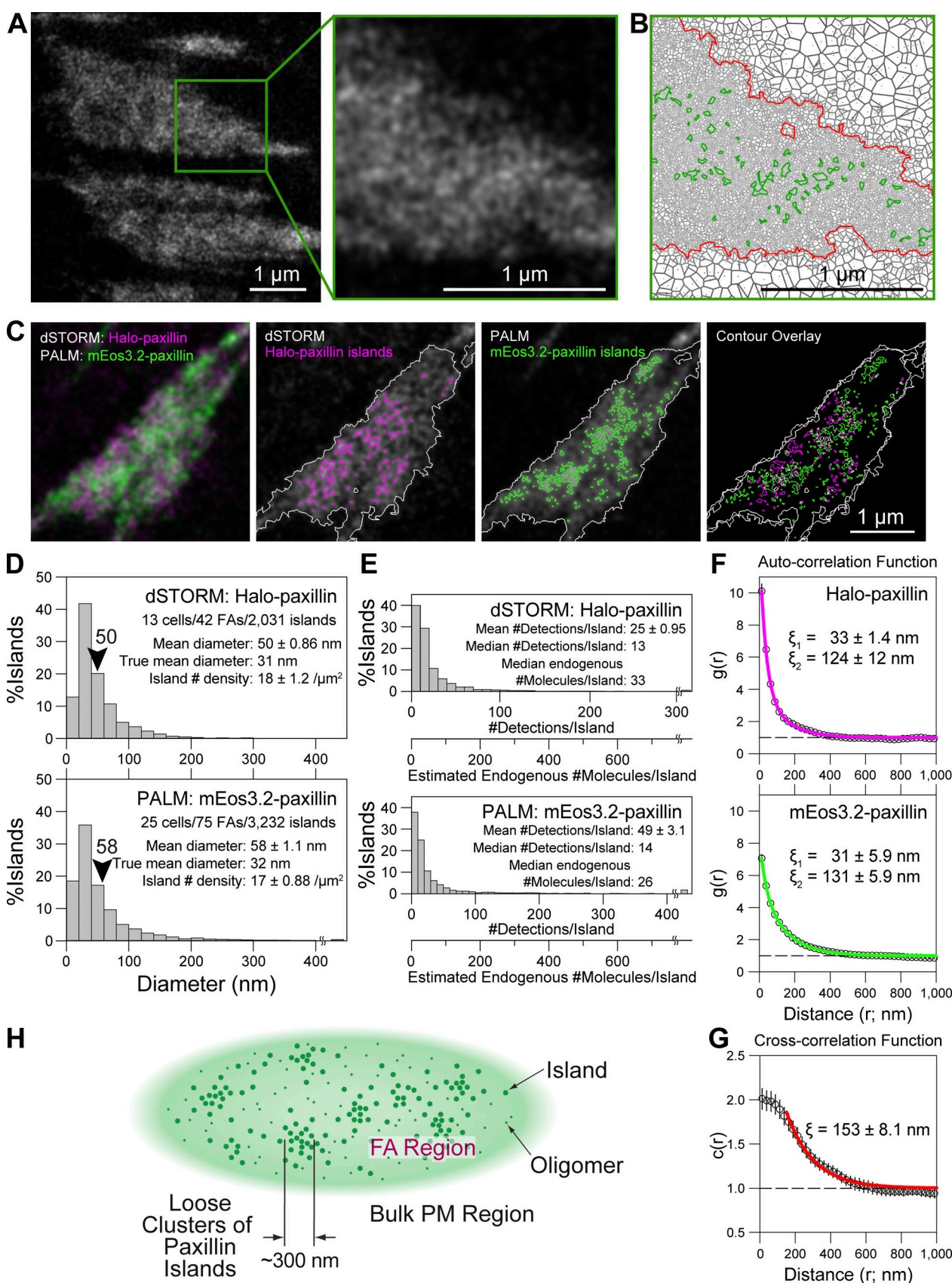

Figure 5. **Ultrafast simultaneous dSTORM (HMSiR-labeled Halo-paxillin) and PALM (mEos3.2-paxillin) in live mEos3.2-paxillin–rescued MEFs reveal that FAs contain paxillin-enriched islands of ≈32 nm in diameter with a median of ≈30 paxillin copies, which are not homogeneously distributed in the FA, but form island-enriched domains (loose island clusters) with a diameter of ≈300 nm. (A)** Typical ultrafast dSTORM image of HMSiR-labeled

Halo-paxillin expressed on the basal PM of a live MEF (data acquisition at 1 kHz for 10 s, with a total of 10,000 frames) and its expanded image. **(B)** The Voronoï diagram of the dSTORM image is shown on the left. The contours of the FA (red) and paxillin-enriched islands (green) are shown (see Fig. S2 B). **(C)** Left: A typical overlaid image of simultaneously recorded ultrafast live-cell dSTORM of HMSiR-labeled Halo-paxillin (magenta) and PALM of mEos3.2-paxillin (green) in an mEos3.2-paxillin–rescued MEF. Data were acquired at 1 kHz for 10 s, with a total of 10,000 frames for both dSTORM and PALM (three figures on the right). The paxillin-enriched islands detected by the Voronoï segmentation analysis of the image on the left (the images are superimposed here) exhibited only partial colocalizations of the paxillin-enriched islands, probably due to limited copy numbers of paxillin in the islands. **(D)** Distributions of the diameters of paxillin-enriched islands obtained from dSTORM and PALM images using the tessellation analysis. The mean diameters were 50 and 58 nm, respectively, but after the correction for single-molecule localization precisions (Fig. S2 C), the true mean diameter of the islands was estimated to be 32 nm for both the dSTORM and PALM results. **(E)** Distributions of the number of detections (localizations) of mEos3.2- and HMSiR-Halo–linked paxillin molecules per detected paxillin-enriched island. The additional x-axis scale shows the estimated endogenous paxillin copy number per island in the parental MEFs. **(F and G)** Auto-correlation (F) and cross-correlation (G) functions calculated for all of the fluorescent spots localized in the detected paxillin-enriched islands (polygons; calculated for each FA and the mean ± SEM was obtained). The best-fit functions are shown in color. **(H)** Schematic FA model, proposing that the FA consists of islands of paxillin (and other FA proteins) with a mean diameter of ≈32 nm, which are not homogeneously distributed in the FA, and instead form loose clusters of ≈300 nm in mean diameter. The FA region outside the paxillin-enriched islands is also probably enriched in paxillin monomers and oligomers (shown by green shading and smaller dots, respectively).

the endogenous paxillin in the parental cell line, which we call "mEos3.2-paxillin–rescued MEFs" (Fig. S3 A). Then, these cells were further transfected with the Halo-paxillin cDNA for transient expression. The expressed Halo-paxillin was labeled with HMSiR at ≈90% efficiency (Morise et al., 2019), and the dSTORM observations were performed using live cells that exhibited similar levels of HMSiR-labeled Halo-paxillin signals. Based on the spot densities detected by PALM (mEos3.2) and dSTORM (HMSiR), we found that these MEFs contain mEos3.2-paxillin and Halo-paxillin (including non-fluorescent molecules) at levels of 0.64× and 0.16× relative to endogenous paxillin, respectively (thus 0.8× altogether), in the parental MEF line (calculations are provided in the legend to Fig. S3 A).

In the subsequent parts describing PALM/dSTORM results, we report the results obtained by using mEos3.2-paxillin–rescued MEFs rather than the human epithelial T24 cells used for the experiments shown in Figs. 2, 3, and 4. This choice is based on the fact that the tagged-paxillin expression levels are well-controlled and quite comparable with the endogenous expression level (0.80×). Furthermore, MEFs are frequently employed in the FA research field (Diez et al., 2009; Seong et al., 2011; Cleghorn et al., 2015; Chen et al., 2016). In addition, MEFs tend to develop larger FAs than T24 epithelial cells, which makes MEFs more appropriate for examining the FA architecture and paxillin-enriched islands more closely.

Due to space limitations, the establishment of optimal dSTORM data acquisition conditions is described in Fig. S3 C and its legend. A typical dSTORM image of HMSiR-Halo-paxillin in a live MEF after the 10-s data acquisition (19-nm single-molecule localization accuracy at 1 kHz for 10,000 frames) is shown in Fig. 5 A. Its Voronoï polygon expression, using the SR-Tesseler software with a thresholding polygon density factor of 1.45 for both contours of FAs and paxillin-enriched islands (Fig. S2 B), is shown in Fig. 5 B.

### Simultaneous two-color ultrafast dSTORM and PALM imaging of live MEFs

Representative simultaneously obtained ultrafast dSTORM (HMSiR-labeled Halo-paxillin) and PALM (mEos3.2-paxillin) images after the 10-s data acquisition (both at 1 kHz for 10,000 frames) are overlaid and shown in Fig. 5 C. The Voronoï segmentation analysis revealed that some islands detected by

dSTORM overlapped with those detected by PALM, while others did not, and vice versa. This would be due to the presence of limited numbers of Halo- and mEos3.2-conjugated paxillin molecules in each island (further discussed later).

The distributions of the paxillin-enriched island diameters (Fig. 5 D) provided mean island diameters of 32 and 31 nm after correction for the effects of the 29- and 19-nm single-molecule localization errors for PALM (mEos3.2) and dSTORM (HMSiR) images, respectively (Fig. S2 C). These paxillin-enriched island diameters in MEFs are almost identical to those evaluated by PALM in human epithelial T24 cells (33 nm; Fig. 4 G).

The median copy number of paxillin molecules located in a paxillin-enriched island in a non-transfected parental MEF was estimated to be 26 copies/island from the PALM data and 33 copies/island from the STORM data (Fig. 5 E; calculations similar to those for T24 cells using the data shown in Fig. 4 H; actual calculations are shown in Materials and methods). Taking the average of these two values obtained from PALM and dSTORM results, we conclude that a paxillin-enriched island in the parental non-transfected MEF cell will contain a median of ≈30 endogenous paxillin molecules. This number is quite comparable with the median copy number of 36 copies of paxillin per island (mEos3.2-paxillin + endogenous paxillin) found in T24 cells. Thus, both the mean sizes and the median paxillin copy numbers in paxillin-enriched islands are similar between MEFs and human epithelial T24 cells. This result suggests that although the sizes and shapes of the entire FAs may vary among different cell types, the basic FA elements, such as paxillin-enriched islands, are apparently similar and that higher organizations of the FA-protein islands may differ among various cell types.

### Paxillin-enriched islands are not distributed homogeneously in the FA, but form island-enriched domains (loose island clusters) with ≈300-nm diameters

The spatial autocorrelation of the individual paxillin spots located within the paxillin-enriched islands (polygons) in the dSTORM and PALM images was examined. The autocorrelation function $g(r)$ (mean ± SEM for 25 FAs; Fig. 5 F) could be fitted using the function $g(r) = 1 + A_1 \times \exp(-r/\xi_1) + A_2 \times \exp(-r/\xi_2)$ ($A_1$ and $A_2 > 0$). The correlation lengths $\xi_1$ and $\xi_2$ provide the measures of the radii of two types of paxillin clusters ($2 \times \xi_1$ and $2 \times \xi_2$ for diameters; Sengupta et al., 2011; Veatch et al., 2012). The shorter correlation length

(2×$\xi_1$; 66 and 62 nm from the dSTORM and PALM data, respectively; Fig. 5 F) would represent the mean diameter of the paxillin-enriched islands prior to the correction for single-molecule localization errors. These values are quite consistent with those determined by the definition of 2 × $\sqrt{polygon\ area\ for\ the\ island/\pi}$, i.e., 50 and 58 nm, respectively, as shown in Fig. 5 D.

The longer correlation lengths (2×$\xi_2$s; 248 and 262 nm from the dSTORM and PALM data, respectively, in Fig. 5 F) can be attributed to the existence of spatial correlations among the paxillin-enriched islands. This result indicates that the paxillin-enriched islands are not randomly distributed within the FA, but rather form FA subdomains enriched in paxillin-enriched islands, i.e., loose clusters of paxillin-enriched islands with diameters of ≈260 nm (two effective digits for the arithmetic mean of 248 and 262 nm).

Next, the spatial cross-correlation between the fluorescent spots in the dSTORM images (HMSiR-Halo-paxillin) and those in the simultaneously recorded PALM images (mEos3.2-paxillin) was examined using all of the fluorescent spots localized within the paxillin-enriched islands (polygons; Sengupta et al., 2011; Veatch et al., 2012; Stone et al., 2017; Fig. 5 G). The c(r) shape was quite different from the g(r) shapes: the c(r) values are much smaller than the g(r) values in the range of 0 < r < 100 nm. This result is consistent with the observation that the paxillin-enriched islands detected by dSTORM and those by PALM did not extensively overlap (Fig. 5 C).

Since c(r) indicated long-range correlations (Fig. 5 G), we fitted c(r) in the range of r >150 nm, in which the radii of <2% of the paxillin-enriched islands fall as measured by dSTORM and PALM (Fig. 5 D), and thus avoided the correlations within an island. The fitting with a single exponential decay function yielded a (2×$\xi$) of 306 nm, which is quite consistent with the 260-nm diameter loose clusters of paxillin-enriched islands, evaluated by the autocorrelation analysis. Since the crosscorrelation function calculated here deals with more paxillin-enriched islands and is less affected by the presence of unresolved larger islands as compared with auto-correlation, it would be more sensitive to the correlations between islands (rather than within islands). Therefore, we consider the average diameter of the paxillin-island-enriched subregion (loose clusters of paxillin-enriched islands) to be ≈300 nm.

In summary, we conclude that the ≈32 nm diameter paxillin-enriched islands are not randomly distributed in the FA, but rather assemble into subdomains that are enriched in these islands (loose clusters of paxillin-enriched islands) with an average diameter of ≈300 nm. This finding further indicates that even though the paxillin-enriched islands detected by PALM and dSTORM might not overlap extensively, they do coexist within the same subdomains, as clarified by auto- and crosscorrelation analyses.

### Talin, FAK, and vinculin also form islands and loose clusters

The spatial distributions of the FA proteins other than paxillin were investigated. As FA proteins, we examined talin and talin-binding FA molecules (Liu et al., 2015a; Goult et al., 2018), including FAK, vinculin, integrin β3, and integrin β1 (paxillin also binds to talin). We performed simultaneous ultrafast dSTORM of one of the Halo-linked FA proteins (HMSiR labeled) other than paxillin and ultrafast PALM of mEos3.2-paxillin (Fig. 6, A and B). Using the Voronoï tessellation analysis and the same definition for the protein islands (≥13 nm in diameter; ≥6 specific protein copies), we detected paxillin-enriched islands by PALM and other-FA-protein-enriched islands by dSTORM and found that they partially overlap, but not extensively (Fig. 6 B). Similar limited overlaps were found even for the pair of mEos3.2-paxillin (PALM) and HMSiR-Halo-paxillin (dSTORM; Fig. 5 C), which was explained by the presence of limited numbers of mEos3.2-linked- and HMSiR-Halo-linked paxillin molecules in an island (each expressed at 0.64× and 0.16× the endogenous paxillin in the parental MEF line). We propose that the limited overlaps of these FA-protein islands with paxillin-enriched islands would be induced for the same reason. Furthermore, since the paxillin, talin, and talin-binding FA proteins examined here should interact extensively (Liu et al., 2015a; Goult et al., 2018), we do not believe that these FA proteins form separate islands to significant extents.

After the dSTORM localization error correction (Fig. S2 C), the mean island diameters of talin, FAK, vinculin, and integrins β3 and β1 fell within the range of 24–29 nm (31 nm for the paxillin-enriched islands; Fig. S4 A). Autocorrelation analyses of talin, FAK, and vinculin islands (Fig. 6 C) provided 2×$\xi_1$ values comparable with the mean diameters obtained from the island diameter distributions (Fig. S4 A; integrin results will be discussed in the next subsection).

The long-range auto-correlations provided the 2×$\xi_2$ range of 230–260 nm (using two effective digits) for talin, FAK, and vinculin, consistent with the 2×$\xi_2$ values for paxillin obtained by PALM and dSTORM (260 nm; Fig. 5 F). Furthermore, the crosscorrelation between the paxillin-enriched islands (by PALM) and the talin, FAK, and vinculin islands (by dSTORM, simultaneously performed with PALM of paxillin) at a spatial scale of r > 150 nm yielded a 2×$\xi$ value of 306–336 nm (Fig. 6 D). These values are consistent with the correlation length between the Halo-paxillin and mEos3.2 paxillin-enriched islands, which is ≈300 nm (Fig. 5 G and related main text). These results indicate that, although the talin, FAK, and vinculin islands might not extensively overlap with the paxillin-enriched islands, probably due to the smaller copy numbers of proteins in an island, at the level of the subregions enriched in the islands of paxillin, talin, FAK, and vinculin, they are concentrated in the same FA subregions with an average diameter of 300–340 nm. For conciseness, we will refer to these as "loose island clusters of 320 nm in diameter."

### FA-protein islands: Integrin copy numbers in an FA-protein island are smaller

In contrast, integrins β3 and β1 did not exhibit longer-range auto-correlation (Fig. 6 C; although they exhibited short-range auto-correlation; see the legend to Fig. 6 C). This result suggests that integrin islands, or more precisely the FA-protein islands containing detectable copy numbers of integrins (≥6 copies/islands), are scarce and thus located quite far apart. Accordingly, we suspect that the copy number of integrins in each FA-protein island would often be <6. Meanwhile, every FA-protein island is

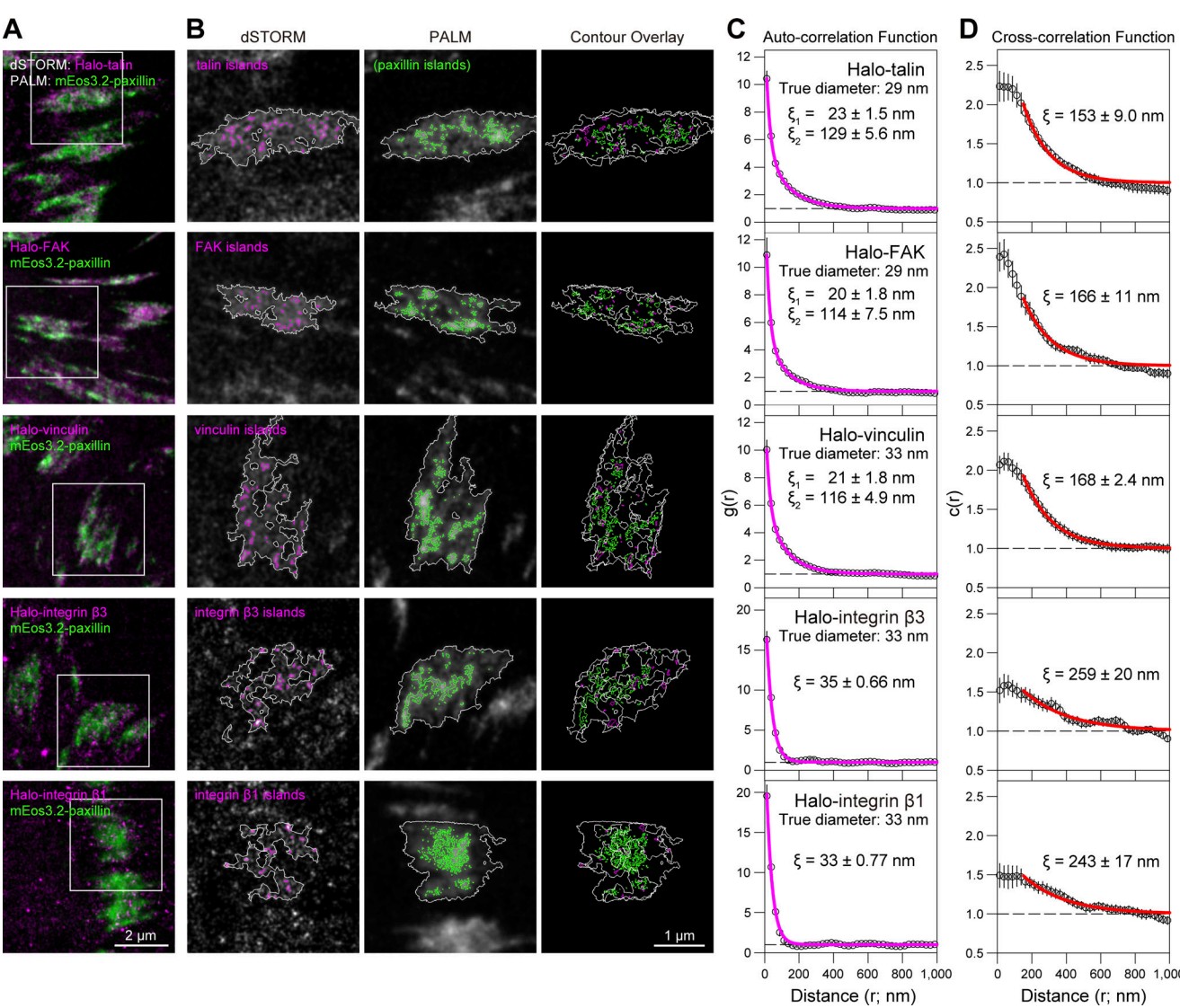

**Figure 6. Islands of various FA proteins and their loose clusters of ≈320 nm in diameter. (A)** Typical overlaid PALM and dSTORM images simultaneously obtained in a live MEF for mEos3.2-paxillin and five other HMSiR-labeled Halo-linked FA molecules (data acquisition at 1 kHz for 10 s). **(B)** The contours of the FA and the islands of paxillin and other FA proteins were identified by the Voronoï segmentation analysis of the images in the regions enclosed within squares shown in A (expanded and overlaid on the images). **(C)** Autocorrelation functions obtained from the dSTORM data for the FA proteins (calculated for all of the fluorescent spots localized in the detected islands; see the legend to Fig. 5 F) with their best-fit functions. Integrins β3 and β1 did not exhibit the longer autocorrelation component, whereas they exhibited short-range auto-correlation lengths ($2 \times \xi_1$ values of 70 and 66 nm, respectively) slightly greater than those for talin, FAK, vinculin, and paxillin ($2 \times \xi_1$ values of 40∼46 nm). However, these slightly greater $2 \times \xi_1$ values are likely spurious, as the mean diameters of the islands enriched in integrins β3 and β1, after localization error correction, are 57 nm in the island diameter distributions (Fig. S4 A), consistent with those of paxillin, talin, FAK, and vinculin (48–62 nm; Fig. S4 A). Therefore, we conclude that the longer short-range autocorrelation lengths of integrins β3 and β1 ($2 \times \xi_1$ values of 70 and 66 nm, respectively) are probably due to the slight mixing of the longer correlation component, which could not be resolved. **(D)** Cross-correlation functions between the dSTORM spots of Halo-linked FA proteins and the PALM spots of mEos3.2-paxillin in the detected islands (see the legend to Fig. 5 G).

likely to contain integrins since FA formation is generally initiated by integrin clustering (Changede et al., 2015). Therefore, we propose that integrins might exist in the majority of FA-protein islands, but the copy numbers of each integrin species (such as integrins β3 and β1) may often be <6 copies/island.

The mean diameters of the islands enriched in FA-proteins, including paxillin, talin, FAK, vinculin, and integrins β3 and β1, determined in this way fall within the range of 24–32 nm (Fig. 5 D and Fig. S4 A), although the overall variations of island diameters are much greater and mostly in the range of 13–100

nm. The median diameter agrees with the diameter of adhesion particles detected by cryo-electron microscopy (25 ± 5 nm; Patla et al., 2010), and is consistent with the findings of Changede et al. (2019), who reported that stable integrin nanoclusters bridge between thin extracellular matrix fibers (≤30 nm), leading to downstream effects on cell motility and growth.

Based on these results and considering the high probabilities that the copy numbers of FA-proteins in most of the islands identified by using paxillin, talin, FAK, vinculin, or integrins β3 or β1 would be quite limited (see the descriptions in the text

related to Fig. 4, G and H; Fig. 5, D–F; Fig. 6, A–C; and Fig. S4 A), we propose that the FA is composed of FA-protein islands with diameters ranging from 13 to 100 nm (with a mean diameter of ≈30 nm), where a variety of FA proteins are assembled in various stoichiometries, including cases where the copy number of certain FA proteins is zero. This proposal does not contradict the specific layered protein architecture of the FA (Kanchanawong et al., 2010; Liu et al., 2015a; Xia and Kanchanawong, 2017) since a 3D-layered structure would exist in each island, and the 3D-layered structure reported previously would represent the protein layers averaged over many FA-protein islands. Meanwhile, even outside the FA-protein islands, the area surrounding the FA should contain higher concentrations of FA proteins as compared with those in the bulk basal PM by the definition of the FA area in the PALM/dSTORM images. These proteins located outside the FA-protein islands might exist as monomers and oligomers and are probably bound to the actin filaments.

### Live-cell dSTORM reveals synchronized paxillin recruitment to the loose clusters of paxillin-enriched islands

Next, we conducted long-term (60-s) dSTORM observations of loose clusters of paxillin-enriched islands in live MEFs expressing HMSiR-labeled Halo-paxillin using a 10× lower excitation laser intensity (2.2 µW/µm²) in the sample plane. Remarkably, even in the last frame of the data acquisition, the number of detected HMSiR spots was decreased by only ≈20% (Fig. S3 C) due to slower photobleaching as well as the exchange of paxillin molecules between the FA and the cytoplasm. The dSTORM observations of the recruitment and dissociation of molecules are only possible with live cells. We employed a data acquisition rate of 250 Hz (4 ms/frame) instead of 1 kHz (1 ms/frame) due to the longer HMSiR on-period duration of 6.6 ms at the laser intensity of 2.2 µW/µm² (Fig. S3 B-a). Under these dSTORM data acquisition conditions, the number of detected photons during an on-period was 278 ± 5.3 and the localization precision for each on-event was 23 ± 0.12 nm (Fig. S3 B-b and -c).

Typical dSTORM image sequences in live and fixed MEF cells, reconstructed for the shifting time window of 10 s (a total of 2,500 frames for each image), and the contours of the FAs and paxillin-enriched islands determined by the Voronoï-based segmentation are shown in Fig. 7, A and B (see Video 2 in the live MEF, where the window width remains 10 s, but the window is shifted by 1 s; also see Video 3 in a live T24 cell for reference). Comparing the sequential images of the live and fixed cells, we observed the extensive recruitment of paxillin molecules to the FAs in live cells, consistent with the results shown in Fig. S3 C. Importantly, the paxillin recruitment appears to occur synchronously among the paxillin-enriched islands within the loose cluster in the time scale of several tens of seconds (Fig. 7 A), while its recruitment to different loose clusters is not synchronized (Fig. 7, A, C, and D). This synchronization within the loose cluster of paxillin-enriched islands is likely induced by the movement of paxillin molecules on the FA membrane (Fig. 7 E). This movement probably involves repeated transient bindings to and releases from several FA-protein islands in the loose cluster until the molecule settles down on one of the islands in the loose cluster. Compare these movement trajectories with the trajectories

in fixed cells (Fig. 7 F), which likely represent the blinking and localization errors of HMSiR-Halo-paxillin molecules. Where the paxillin-enriched island densities are lower in the FA, the paxillin molecules arriving there would leave quickly due to the lack of binding sites. These findings suggest that an individual loose cluster of FA-protein islands might function as a unit for recruiting paxillin to the FA, and perhaps other FA cytoplasmic proteins as well, and thus for the formation and disintegration of the paxillin-enriched islands. Namely, the FA-protein islands within a cluster work collaboratively rather than competitively for recruiting FA proteins. Therefore, it is concluded that the loose clusters of FA-protein islands represent the higher levels of a coordinated organization and regulation of FA formation–disintegration and function (Spiess et al., 2018).

### TfR undergoes hop diffusion inside the FA

We next investigated the organization of the interisland channels, which are predominantly composed of the fluid membrane. For this purpose, we observed the movement of the non-FA transmembrane protein TfR within the FA, using ultrafast SFMI at a frame rate of 6 kHz (0.167-ms resolution). We employed T24 cells because, as described in the companion paper, we used T24 cells to show that TfR undergoes hop diffusion in the basal PM outside the FA (bulk basal PM), and thus we intended to compare the TfR diffusion inside the FA with that outside the FA.

We found that 85% of the trajectories of TMR-labeled Halo-TfR obtained within the FA region were statistically classified into the suppressed diffusion mode (Fig. 8, A and B; Video 4; and Table 1). The hop-diffusion fitting of the plot of mean-square displacement (MSD) against the time lag ($\Delta t$) of each trajectory (see the companion paper) revealed that the median compartment size was reduced to 74 from 109 nm in the bulk basal PM (Fig. 8 C). Thus, the compartment area size within the FA is smaller than that in the bulk PM by a factor of 2.2. The dwell lifetime was 36 ± 2.8 ms in the FA, which is 1.5× longer than that in the bulk basal PM (24 ± 1.6 ms; Fig. 8 D). These results unequivocally demonstrated that the interisland fluid membrane in the FA is compartmentalized.

Given the similarity to hop diffusion in the bulk basal and apical PMs described in the companion paper, we propose that the compartmentalization in the FA is induced by the actin-based membrane skeleton meshwork. However, attempts to directly observe the effects of actin depolymerizing drugs failed because at the concentrations where their effects were detectable, the cells became round and some did not survive. Assuming that the compartmentalization of the fluid membrane part within the FA is induced by the actin meshwork bound to the PM inner surface (picket-fence model, Fig. 1 B), the smaller compartment area size (2.2×) indicates that the actin-based membrane skeleton meshwork is finer in the FA. This could be due to the presence of more actin filaments in the FA (Patla et al., 2010).

The smaller compartments found within the FA (74 vs. 109 nm in the bulk basal PM; median values) would result in shorter dwell lifetimes of TfR within a compartment (higher hop frequency) as compared with those outside the FA, if the

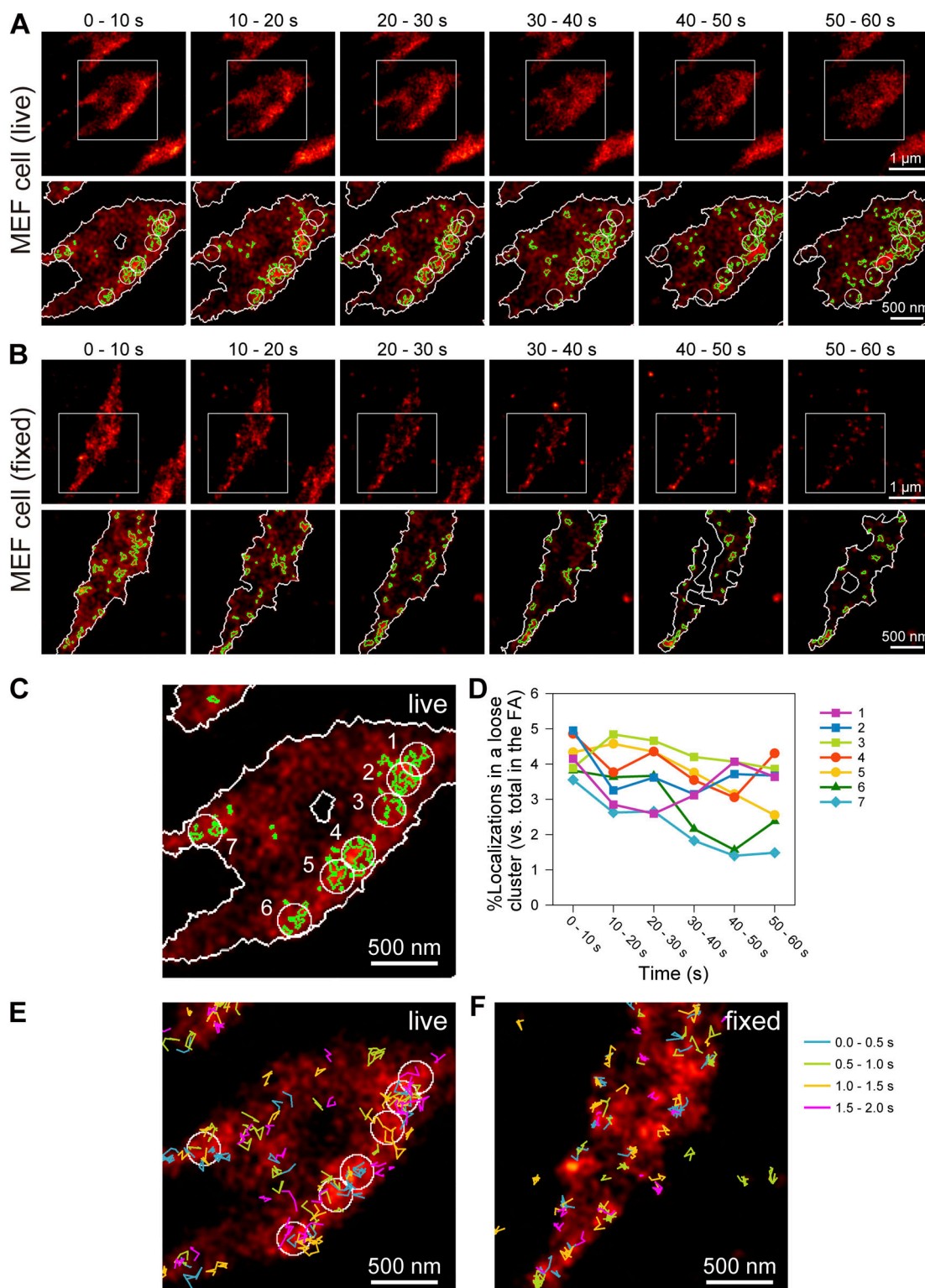

Figure 7. **Synchronized paxillin recruitment to loose paxillin-enriched island clusters detected by the live-cell dSTORM. (A and B)** Spatially non-homogeneous paxillin recruitment in time found by dSTORM of live MEF cells. Typical dSTORM image sequences of HMSiR-labeled Halo-paxillin on the basal PM of a live MEF (A) and a fixed MEF (B) are shown every 10 s (data acquisition of 2,500 frames at 250 Hz; top rows). The bottom rows show the expanded images of the regions in squares in A superimposed by the contours of FAs (white) and paxillin-enriched islands (green) determined by the Voronoï segmentation analysis. For the circles in the expanded images of the live MEF in A, see the legend of C. **(C)** The bottom-left image in A (0–10-s expanded image superimposed by the Voronoï diagram) is further expanded. Here, we placed seven 250-nm diameter circles, which are the plausible loose clusters of paxillin-enriched islands. These seven circles are included in the image sequences in the bottom row in A to show the spatially heterogeneous time-dependent changes of the local fluorescent paxillin concentration. Even outside of the detected paxillin-enriched islands, the FA is enriched in paxillin, which might exist in islands enriched in other FA proteins and/or as smaller paxillin oligomers and monomers. **(D)** Paxillin recruitment might take place synchronously in each loose cluster,

suggesting that the loose island cluster functions as the unit for recruiting paxillin. The figure shows the time-dependent changes of the percentages of paxillin localizations in each circle (loose cluster) shown in C, relative to the total paxillin localizations, during each 10-s sequence in the entire FA. The number of each graph refers to each numbered circle in C. **(E and F)** Recruited single paxillin molecules frequently undergo rapid movements on the FA membrane, which might induce synchronous paxillin recruitment to paxillin-enriched islands within the loose island cluster. Compare the results obtained in live (E) and fixed (F) MEF cells. The trajectories in the figure show those of newly detected paxillin molecules of four frames and longer (for ≥16 ms; data acquisition at 250 Hz) observed during a 2-s dSTORM data acquisition period (500 frames) to obtain the superimposed 0- to 10-s dSTORM images, which are the same as those shown in A and B (images on the extreme left). The trajectory color is changed every 0.5 s in the 2-s image frame sequence.

compartment boundary's properties in the FA are the same as those outside the FA: the hop frequency would increase by a factor of more than 2 ($[109/74]^2$) due to the greater frequency of TfR arrival at the compartment boundaries, and thus the expected dwell lifetime would be <12 ms. However, the dwell lifetime in the FA is 36 ms (Fig. 8 D), which is three times longer than expected, indicating that the ability of the actin filaments to confine TfR is much greater in the FA. It is possible that the actin meshwork on the PM might be more stable in the FA due to the finer actin meshwork in the FA and/or the binding of FA proteins and their oligomers to actin filaments, which might suppress the filament fluctuation and transient breakage of the actin meshwork due to the binding of actin severing molecules such as cofilin. These would lower the TfR hop probability across the compartment boundaries made of the actin-fence and pickets. Another potential explanation might be that the FA proteins densely bind to the actin-based meshwork, thereby sterically hindering the passage of TfR across the meshwork (compartment boundaries). Indeed, our PALM and dSTORM results suggested that FA proteins such as paxillin, talin, FAK, and vinculin are enriched in the FA even outside the FA-protein islands (Fig. 4 F; Fig. 5, B and C; and Fig. 6, A and B), perhaps by binding to the actin filaments in the FA.

Based on these observations, we propose a new architectural model for the FA, which we call the compartmentalized archipelago model of the FA-protein island clusters and oligomers (Fig. 8 E). In this model, the FA region is partitioned into 74-nm compartments by the actin-based meshwork, and the FA-protein islands are nonhomogeneously scattered over the actin-based meshwork. In addition, the monomers and oligomers of the FA proteins might bind to the actin meshwork, perhaps stabilizing it and also acting as diffusion obstacles.

In addition, we found that TfR is excluded from certain subregions in the FA, which might be the regions with the loose clusters of FA-protein islands (Fig. S5). Even if TfR molecules can enter these regions, their diffusion would be quite suppressed (Fig. 8 F; for more explanations, see the legend to Fig. S5).

### Integrin β3 becomes temporarily immobilized at the paxillin-enriched islands

We next performed simultaneous live-cell PALM of mEos3.2-paxillin to visualize the paxillin-enriched islands, and SFMI of integrin β3 at 250 Hz to observe its movement and immobilization (integrin β3's N-terminus fused to the acyl-carrier protein (ACP)-tag protein, labeled with SeTau647; <1,000 copies expressed on the cell surface, thus avoiding overexpression; Fig. 9 A-a and b; and Videos 5 and 6). For each FA that integrin β3 molecules entered, we obtained the distribution of the paxillin pixel signal intensities inside the PALM image of the FA and normalized it by using the median value of the distribution, yielding the thermographic scale depicted in Fig. 9 B. Based on this scale, the thermographic PALM image was obtained and superimposed on an integrin β3 trajectory (Fig. 9 C). When we found the immobilization of integrin β3 molecules in the FA, we measured the paxillin pixel signal intensity at the center of the immobilized position (Simson et al., 1995; Fig. 9 C and the arrow in Fig. 9 B). By conducting these measurements for 60 integrin β3 immobilization events in 55 FAs (averaged thermographic intensity distributions are presented in Fig. 9 D top), we obtained a normalized pixel signal intensity distribution at the locations where integrin β3 molecules were immobilized (Fig. 9 D bottom). Our results demonstrated that over two-thirds of the immobilization events occurred in areas where the paxillin intensity exceeded its median value, indicating that integrin β3 molecules are preferentially immobilized at places where the paxillin densities are higher, namely, on the paxillin-enriched islands. The photobleaching lifetime of Se-Tau647 bound to ACP-integrin β3 at 250 Hz was 270 frames (1.1 s), and >90% of the immobilization events occurred from the beginning of the observation until photobleaching (Fig. 9 A-b and Video 6).

Tsunoyama et al. (2018) previously found that integrins β3 and β1 undergo temporary immobilizations in the FA in the time scales of 0.66–79 and 0.5–43 s, respectively (exponential lifetimes), and by using integrin β3, we have shown here that 68% of these temporary immobilizations occur on the paxillin-enriched islands. The remaining 32% of the temporary immobilization might occur in other FA-protein islands containing <6 paxillin copies. Therefore, we conclude that the cell linkage to the extracellular matrix primarily occurs through the integrin molecules mediating the linkage of the FA-protein islands to the extracellular matrix. Although each integrin molecule might contribute to the binding for periods on the order of one to several tens of seconds, multiple integrin molecules would be dynamically and continually recruited to the FA-protein islands, exchanging with those located outside the FA-protein islands. As a result, the FA-protein islands would remain linked to the extracellular matrix for much longer durations. The dynamic linkage of integrins via FA-protein islands would facilitate the rapid control of FA formation and disintegration (Shibata et al., 2012; Rossier et al., 2012; Tsunoyama et al., 2018; Orré et al., 2021).

We then increased the frame rate from 250 Hz to 6 kHz to examine whether integrin β3 undergoes hop diffusion in the FA (Halo-integrin β3, rather than ACP-integrin β3, which was used for obtaining the results shown in Fig. 9, A–D, was used because SeTau647 photobleaches more slowly when bound to the Halo

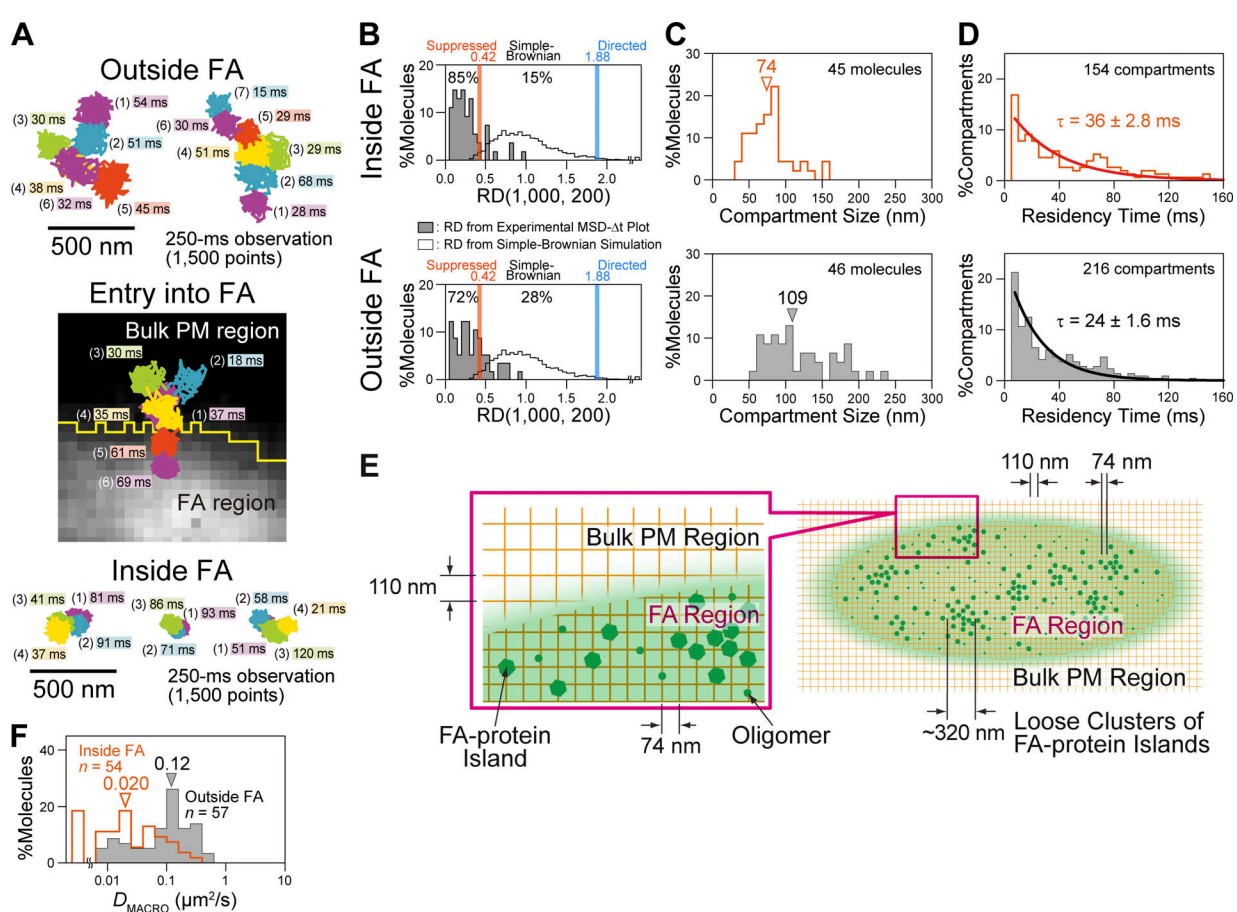

Figure 8. **TfR undergoes hop diffusion within the FA region but with a smaller median compartment size of 74 nm (vs. 110 nm in the bulk basal PM) and a longer dwell lifetime of 36 ms (vs. 24 ms in the bulk basal PM). (A)** Typical ultrafast single fluorescent-molecule trajectories (0.167-ms resolution; 6-kHz frame rate) of TMR-Halo-TfR diffusing outside the FA (top), entering the FA from the bulk PM (middle), and diffusing inside the FA (bottom). In the middle figure, the background of the trajectory is the mGFP-paxillin image, and the yellow line shows the boundary of the FA region determined by binarization, using the minimum cross entropy thresholding. **(B)** Distributions of *RD*s (relative deviation from ideal Brownian diffusion; for the definition and explanation, see Fig. 4 B in the companion paper) for the trajectories of TfR diffusing inside (top) and outside (bottom) the FA region in the basal PM (shaded histograms; 54 and 57 molecules, respectively). The open histogram (the same for both inside and outside the FA) represents the *RD* distribution for simple-Brownian trajectories generated by Monte-Carlo simulations (5,000 trajectories), with the red and blue vertical lines indicating the 2.5 percentiles from both ends of the distribution ($RD_{min}$ and $RD_{MAX}$, respectively). The trajectories exhibiting *RD* values below $RD_{min}$ are categorized into the suppressed diffusion mode. For details, see Fig. 4 B of the companion paper. The data for "Outside FA" shown here in B–D are reproduced from Fig. 7, B–D (top), in the companion paper for the direct comparison with the "Inside FA" data. **(C)** Distributions of the compartment sizes determined by the hop-diffusion fitting of the MSD-Δ*t* plot of each TfR trajectory (see Fig. 4 A, Supplemental theory 2, and related main text in the companion paper), inside (top) and outside (bottom) the FA region in the basal PM. Arrowheads indicate the median values (statistically significant difference with P = $1.7 \times 10^{-7}$, using the Brunner–Munzel test). **(D)** Distributions of the TfR residency times within a compartment, obtained for each molecule in each compartment determined by the TILD analysis (see Fig. S5 and its legend in the companion paper), with the best-fit exponential curves (providing the dwell lifetimes), inside (top) and outside (bottom) the FA region. A statistically significant difference exists between before and after stimulation with P = $4.9 \times 10^{-4}$, using the log-rank test. For the derivation of the single exponential dependence of the dwell lifetime distribution, see the subsection "Expected distribution of the residency times: development of the hop diffusion theory" in the legend to Fig. S5 in the companion paper. **(E)** Schematic model of our proposed FA architecture, based on the observations made in the present study. The FA-protein islands with various molecular stoichiometric compositions (24–32 nm in the mean diameter in MEFs; green hexagons) form loose clusters with a diameter of ≈320 nm in the compartmentalized fluid (actin meshwork schematically shown by the green lattice). Recruitment of paxillin to these islands might not occur randomly at individual islands but instead occur synchronously at islands within the same loose cluster. The fluid membrane in the inter-island channels in the FA is partitioned into 74-nm compartments (110 nm in the bulk PM). These compartment boundaries are probably composed of the actin-based membrane-skeleton mesh, which might be bound and stabilized by various FA proteins as monomers, oligomers, and islands. **(F)** Distributions of $D_{MACRO}$ for TfR determined by the hop-diffusion fitting of the MSD-Δ*t* plot for each TfR trajectory obtained at 6 kHz. Arrowheads indicate the median values (statistically significant difference with P = $9.3 \times 10^{-9}$, using the Brunner–Munzel test).

protein). Indeed, integrin β3 molecules underwent hop diffusion in the FA as well as in the bulk PM (Fig. 9 E and Video 7), whereas many integrin β3 molecules remained immobilized in the FA for the entire observation period of 250 ms (6 kHz × 1,500 frames).

## Discussion

The ultrafast camera described in the companion paper enables us to perform ultrafast PALM, ultrafast dSTORM, simultaneous two-color ultrafast PALM-dSTORM, and simultaneous two-color ultrafast PALM-SFMI observations. After establishing

**Table 1.** Motional mode, compartment size (*L*), $D_{MACRO}$, and dwell lifetime (τ) characterizing the hop diffusion of TfR, outside and inside the FA region in the basal PM of T24 cells

| | Motional Mode (%) | | Cmpt. Size (*L*, nm)[a] | $D_{MACRO}$ (µm²/s)[a] | Residency Lifetime (τ, ms)[a] | n[b] |
|---|---|---|---|---|---|---|
| | **Suppressed** | **Simple** | **Median** <br> **Mean ± SEM** <br> **Histogram Fig. No.[c]** | **Median** <br> **Mean ± SEM** <br> **Histogram Fig. No.** | **Mean ± SEM** <br> **Histogram Fig. No.** | |
| TfR (outside FA) | 72 | 28 | 109 <br> 121 ± 6.9[#1] <br> Fig. 8 C | 0.12 <br> 0.14 ± 0.016[#2] <br> Fig. 8 F | 24 ± 1.6[#3] <br> Fig. 8 D | 57 |
| TfR (inside FA) | 85 | 15 | 74 <br> 79 ± 4.2[Y1] <br> Fig. 8 C | 0.020 <br> 0.046 ± 0.0081[Y2] <br> Fig. 8 F | 36 ± 2.8[Y3] <br> Fig. 8 D | 54 |

[a]Cmpt. Size = Compartment size (*L*). *L* and $D_{MACRO}$ in the intact PM were estimated by the hop-diffusion fitting. Residency Lifetime (τ) = exponential decay lifetime (mean ± SEM; SEM was determined from the fitting error of the 68.3% confidence interval) within a compartment, based on the residency time distributions obtained by the TILD analysis.
[b]The number of inspected molecules.
[c]Histogram. Fig. No. = The figure in which the histogram (distribution) is shown.
[#], [Y], and [N] are the results of the statistical tests. The distributions selected as the basis for the comparisons are shown by the superscript [#]. Different numbers indicate different bases for the Brunner-Munzel test (1 and 2) and the log-rank test (3). The superscript Y (or N) with numbers indicates that the distribution is (or is not) significantly different from the base distribution indicated by the number after [#], with P values smaller (or greater) than 0.05 as shown: Y1: P = 1.7 × 10⁻⁷, Y2: P = 9.3 × 10⁻⁹, Y3: P = 4.9 × 10⁻⁴.

the optimal conditions for PALM using mEos3.2 and dSTORM using HMSiR (Fig. 2 and Fig. S3), we achieved a high data acquisition rate of 1 kHz with very large-view fields of up to 640 × 640 pixels (35.3 × 35.3 µm² with the pixel size of 55.1 nm, an area that can often encompass an entire live cell; Figs. 3 and 4). We showed that the optimal spatiotemporal resolution is photon-limited by fluorophore photophysics, with a data acquisition rate of 1 kHz and single-molecule localization precisions of 29 and 19 nm for mEos3.2 (PALM) and HMSiR (dSTORM), respectively. With the developed camera, the temporal resolution of SMLM is no longer limited by the instrument but rather by the availability of dyes that can emit photons faster and photobleach faster. Since the camera itself can operate at higher frame rates up to 10 kHz for the frame size of 256 × 256 pixels, 30 kHz for 128 × 112 pixels, and 45 kHz for 128 × 64 pixels, SMLM imaging at or beyond video rate would become possible using the developed ultrafast camera, contingent upon the emergence of novel fluorophores in the future.

Even now, the data acquisition of 333–10,000 frames can be accomplished within a mere 0.33–10 s, enabling live-cell SMLM for subcellular structures in the PM such as caveolae and FAs (Figs. 3, 5, 6, and 7). This duration is about 30 times shorter than the common several-minute SMLM data acquisition time.

Furthermore, simultaneous ultrafast PALM and ultrafast SFMI can be conducted using the developed camera. Thus, the developed ultrafast camera system now allows observations of subcellular structures in/on the PM with nanoscale precisions while simultaneously tracking the ultrafast behaviors and movements of many individual single molecules in these structures. Therefore, we conclude that this ultrafast camera system will serve as an extremely powerful tool for cell biology research.

By using the developed ultrafast simultaneous two-color PALM-dSTORM, we discovered that FA proteins, including

paxillin, integrins β1 and β3, talin, FAK, and vinculin, assemble into nanoclusters, some of which are ≥13 nm in diameter and contain ≥6 copies of one of the FA-protein species, which we called FA-protein islands, with a mean island diameter of 24–32 nm in MEFs (Fig. 5, D and F; Fig. 6 C; and Fig. S4 A), after the correction for single-molecule localization errors (Fig. S2, B and C). The diameter distribution is quite broad, ranging from 13 to 100 nm. These estimates are generally consistent with the diameters of FA adhesion particles found by cryo-electron microscopy (Patla et al., 2010) and integrin nanoclusters identified by super-resolution microscopy (Changede et al., 2019).

The FA-protein islands identified by the ultrafast simultaneous two-color PALM-dSTORM of different FA proteins or even the same protein (paxillin) exhibited limited colocalizations (Fig. 5 C and Fig. 6, A and B). This predominantly occurs because an FA-protein island tends to contain <6 copies of the same protein species and many smaller nanoclusters of FA proteins that were not categorized into islands might in fact be forming the FA-protein islands containing a total of ≥6 copies of various FA proteins. This limitation, due to the presence of limited copy numbers of the same molecular species in a structure, is a general phenomenon and a limitation in super-resolution methods.

It follows then that the molecular stoichiometries of individual FA-protein islands (and nanoclusters) could vary widely. In the case of paxillin, a median of ≈30 copies exists in an island (Fig. 4 H and Fig. 5 E, and related main text), but the copy number variation is large. Even disregarding the islands containing >100 paxillin copies in these distributions, as discussed in the text related to these figures (they are likely unresolved paxillin-enriched islands), the variation would still be in the range of 0–100 copies per FA-protein island (because from the discussion in the previous paragraph, some FA-protein islands might contain 0–5 paxillin copies). Such broad stoichiometries

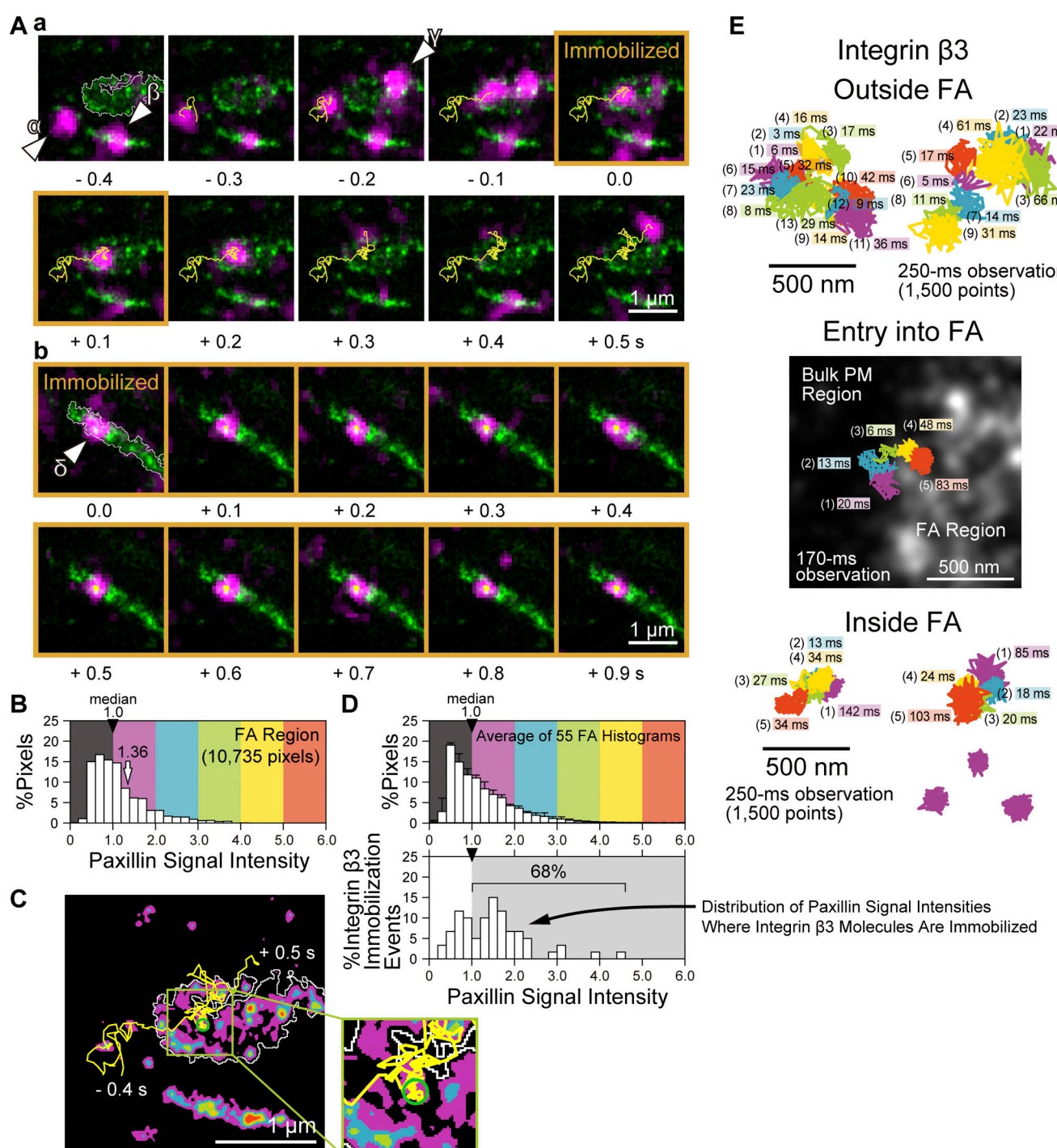

Figure 9.   **Integrin β3 molecules are temporarily immobilized at the paxillin-enriched islands and perhaps on FA-protein islands in general. (A)** Typical results of single-molecule tracking of SeTau647-ACP-integrin β3 (recorded at a 4-ms resolution; large magenta spots; images shown every 0.1 s or 25 frames) simultaneously performed with the live-cell PALM of mEos3.2-paxillin, visualizing the paxillin-enriched islands (green spots; the same PALM image is used for all of the images). Data were obtained by using T24 cells. The magenta spots representing single integrin β3 molecules appear much larger than the green spots representing the paxillin-enriched islands because the spot size in single-molecule imaging is diffraction-limited (≈250 nm for single SeTau647-ACP-integrin β3 molecule: the single-molecule localization precision was ≈21 nm, obtained from the SDs of 15-frame immobile trajectories; *n* = 30), whereas the FA islands were imaged by PALM. The FAs we consider here are indicated by the white contours in the first frames, which were determined by applying the minimum cross entropy thresholding to the PALM images. (a) Three fluorescently labeled integrin β3 molecules (marked as α, β, and γ) appeared in the FA region. The trajectory of the α molecule is shown. Time 0 is set at the time when the α molecule started exhibiting immobilization in an FA. The immobilization of this molecule lasted for 0.19 s (orange frames). The signal intensities of the magenta spots at times 0.3 and 0.4 s are lower, probably due to the stochastic fluctuation of emitted photon numbers and blinking (off-periods) shorter than 4 ms. (b) An integrin β3 molecule was immobilized from time 0 till the end of the observation, for 1.1 s (270 frames at 250 Hz). Large majorities (>90%) of integrin β3 molecules in the FA were immobile throughout the observation period of 1.1 s, consistent with the previous results reported by Tsunoyama et al. (2018). **(B)** Distribution of the pixel intensities of the mEos3.2-paxillin

signal in the FA indicated by the white contour in the first frame in A-a, normalized by its median pixel intensity. The thermographic color scale for this FA is superimposed (in the linear scale between 0× and 6× of the median intensity). **(C)** The thermographic PALM image of mEos3.2-paxillin shown in A-a, using the color scale shown in B. The immobilized area identified in the trajectory of the molecule α, as previously defined (Simson et al., 1995), is shown by a green 104-nm diameter circle along with the yellow trajectory (also see the expanded figure on the right). The paxillin pixel signal intensity at the center of the immobilized position was measured (open arrow in B), and after this was done for 60 integrin β3 immobilization events, the histogram in D bottom was generated. **(D)** Top: Averaged normalized distribution of mEos3.2-paxillin signal pixel intensities (the distributions like that shown in B for a single FA are averaged over 55 FAs). Bottom: Distribution of the normalized mEos3.2-paxillin signal pixel intensities at the centers of the immobilization circles of single integrin β3 molecules (such as the green circle in C; n = 60 events), indicating that integrin molecules are preferentially immobilized on the paxillin-enriched islands (refer to the main text). **(E)** Typical ultrafast single fluorescent-molecule trajectories of integrin β3 molecules observed at 6 kHz, diffusing outside the FA region (top), entering the FA region (middle), and diffusing and becoming temporarily immobilized inside the FA zone (bottom). The middle figure displays a typical result of simultaneous observations of ultrafast (6 kHz) single-molecule tracking of SeTau647-Halo-integrin β3 (colored trajectory) and ultrafast live-cell PALM of mEos3.2-paxillin (islands; 1 kHz, 10-s integration), showing the entry of integrin β3 from the bulk basal PM into the FA region. Consistent with the FA model of the archipelago architecture of the FA-protein islands in the compartmentalized fluid, integrin molecules continued to exhibit hop diffusion when they entered the FA region outside the paxillin-enriched islands.

would be possible because many FA proteins can interact with multiple other FA protein species (Geiger and Yamada, 2011; Horton et al., 2015). In addition, some FA proteins are capable of undergoing liquid–liquid phase separation, leading to the formation of condensates and co-condensates (Li et al., 2020 *Preprint*; Wang et al., 2021; Case et al., 2022), and consequently, they could form nanoscale condensates that recruit other client FA proteins to generate FA-protein islands. Furthermore, the diverse stoichiometry of FA proteins in each FA-protein island indicates the existence of quite different types of FA-protein islands. For example, Shroff et al. (2007) reported very low co-localization levels of paxillin and zyxin clusters, displaying interdigitated morphologies. Spiess et al. (2018) found that active and inactive integrin β1 molecules segregate into distinct nanoclusters. The concept of broad stoichiometries in the FA-protein islands is consistent with the presence of hundreds of FA proteins identified by unbiased proteomic approaches (Horton et al., 2015; Robertson et al., 2015).

The broad diversities in the sizes, protein copy numbers, protein compositions, and stoichiometries of individual FA-protein islands (and possibly nanoclusters) might be the general characteristics of the FA. This implies that for enhancing the FA's ability to respond to very broad ranges of mechanical stress levels and loading rates and perform mechanosignal transduction and regulate cell binding to various extracellular matrices, these broad characteristics of FA-protein islands might be essential.

Along the z-axis (in 3D), each FA-protein island would comprise layers of proteins with distinct molecular compositions, as previously found for the entire FA (Kanchanawong et al., 2010; Liu et al., 2015a; Xia and Kanchanawong, 2017). Hence, we propose that these layers found previously would represent the averages over the layers in FA-protein islands. Meanwhile, in the FA region, and even outside the FA-protein islands, the FA protein concentrations exceed their average concentrations in the bulk basal PM. Therefore, some fractions of each FA protein might exist as oligomers and monomers outside the FA-protein islands in the FA and are probably bound to the actin meshwork underlying the PM cytoplasmic surface in the FA region (as described in the main text related to Fig. 5 C and Fig. 6, A and B; the models are shown in Fig. 1 D-d and -e).

Since the protein tags, mEos3.2 and the Halo-tag protein, are attached to the N-termini of all the examined proteins, the island

sizes might only represent the positions of the FA proteins' N-termini. This point might not be important for spherical ∼ ellipsoidal molecules, but talin, for instance, is a highly elongated molecule with a length of ≈97 nm and is oriented at ≈15° relative to the PM surface in the FA (Liu et al., 2015a). Therefore, its length projected along the PM is ≈94 nm (the z-axis length is ≈25 nm; Liu et al., 2015a), i.e., its 2D-projected length is much greater than the mean diameter of the paxillin-enriched island (≈32 nm). This suggests that talin might be mainly located in greater islands and/or might place its N-terminal FERM domain within the central 32-nm region of the island, where it binds to integrins, paxillin, kindlin, vinculin, and other FA proteins (Grashoff et al., 2010; Yao et al., 2014), whereas its C-terminus might be extended beyond the 32-nm core region of the FA-protein island.

The FA-protein islands are not homogeneously distributed in the FA but rather form loose island clusters of ≈320 nm in diameter (Fig. 1 D-d, archipelago model of FA-protein island clusters and oligomers; Fig. 5, F–H; and Fig. 6, C and D). Our analysis using paxillin indicated that these 320-nm loose island clusters function as the units for recruiting paxillin to the FA (Fig. 7, A–D). The dynamics of the recruitment and exchanges of paxillin molecules could never have been observed without applying ultrafast dSTORM to live cells (Fig. 7, A and B).

The fluid-membrane part of the FA is compartmentalized in a similar manner to the bulk basal PM outside the FA, albeit with a smaller compartment size (74 vs. 109 nm in the bulk basal PM). TfR molecules undergo hop diffusion among these compartments with an exponential dwell lifetime within a compartment of 24 ms (vs. 36 ms in the compartment in the bulk basal PM; Fig. 8 and Fig. 1 D-e). This compartmentalization in the FA would probably be induced by the actin-based membrane skeleton, just like that in the bulk PM. The means by which the actin-based membrane skeleton is linked to both the FA-protein islands and the stress fibers would be the next key issue that requires clarification.

The archipelago architecture of the FA-protein islands based on the fluid membrane is biologically significant because it would allow for integrin's rapid recruitment to and removal from the FA-protein islands, thereby facilitating the quick regulation of FA formation, reorganization, and disintegration. We indeed visualized the diffusion of integrin β3 within the FA's fluid membrane region and its immobilization at the paxillin-

enriched islands for various durations (Fig. 9). Previously, using the video-rate SFMI, Tsunoyama et al. (2018) found that both integrins β3 and β1 undergo temporary (<80-s) immobilizations in the FA but perform distinct functions in FA formation, maintenance, and disintegration (also refer to the following publications: Roca-Cusachs et al., 2009; Rossier et al., 2012; Schiller et al., 2013). In the growing phase, integrin β1 initially exhibited longer immobilizations, while integrin β3 did so when the FA was in the mature steady phase. In the process of FA disintegration, the prolonged immobilizations of integrin β1 were reduced first, while integrin β3 continued exhibiting longer immobilizations for some time. In the present research, we unequivocally demonstrated that these immobilizations occur at the FA-protein islands, clearly demonstrating the functional importance of the FA-protein islands (Fig. 9).

Taken together, we conclude that the FA predominantly consists of the fluid membrane that is partitioned into 74-nm compartments by actin-based picket fences and the archipelago of FA-protein islands of 13~100 nm in diameter (mean diameter of ≈30 nm), which might be linked to the actin membrane-skeleton meshes. The FA-protein island distribution is non-homogeneous, forming subregions or loose clusters enriched in FA-protein islands of ≈320 in diameter, which could function as units for FA formation and disintegration. Integrin molecules diffuse in and out of the FA and are occasionally anchored to the FA-protein islands, mechanically linking the actin membrane skeleton to the extracellular matrix, and undergo mechanosignal transduction (Fig. 1 D-e).

## Materials and methods

### Cell culture
Human T24 epithelial cells (the ECV304 cell line used in the previous research, which was later identified as a subclone of T24; Dirks et al., 1999; Murase et al., 2004) were cultured in Ham's F12 medium (Sigma-Aldrich) supplemented with 10% fetal bovine serum (Sigma-Aldrich). MEFs were cultured in DMEM (Nacalai Tesque), supplemented with 10% fetal bovine serum (Gibco). The cells were grown on 12-mm diameter glass-bottom dishes (IWAKI) and used 2 d after inoculation for single-molecule microscopy observations. To culture cells expressing mGFP-paxillin or mEos3.2-paxillin, the glass surface was coated with fibronectin by incubation with 5 µg/ml fibronectin (Sigma-Aldrich) in PBS (pH 7.4) at 37°C for 3 h.

### Ultrafast live-cell PALM of caveolin-1-mEos3.2 and mEos3.2-paxillin (Figs. 2, 3, and 4)
The plasmid encoding mouse caveolin-1 (GenBank: U07645.1) fused to mEos3.2 (caveolin-1-mEos3.2) was generated by replacing the cDNA encoding the EGFP protein in the caveolin-1-EGFP plasmid (a gift from T. Fujimoto, Nagoya University, Nagoya, Japan; Kogo and Fujimoto, 2000) with the cDNA encoding mEos3.2 (Zhang et al., 2012; generated by making three point mutations, I102N, H158E, and Y189A, in the Addgene mEos2 plasmid #20341 [http://n2t.net/addgene:20341; RRID:Addgene_20341], a gift from L. Looger, Janelia Research Campus Ashburn, VA, USA; McKinney et al., 2009). The plasmid encoding mEos3.2-paxillin was generated

from the mGFP-paxillin plasmid used in the companion paper (Fujiwara et al., 2023). T24 cells were transfected with the cDNAs encoding caveolin-1-mEos3.2 and mEos3.2-paxillin using Nucleofector 2b (Lonza) according to the manufacturer's recommendations.

Ultrafast live-cell PALM data acquisitions were performed on the basal PM at 37 ± 1°C. The TIR illumination mode was employed in our home-built TIRF microscope, based on a Nikon Ti-E inverted microscope, equipped with two ultrafast camera systems developed in this project and a high NA oil immersion objective lens (CFI Apo TIRF 100×, NA = 1.49, Nikon). Photoconversion of mEos3.2 was induced by a 405-nm diode laser (PhoxX, 120 mW, Omicron) with its intensity exponentially increasing from 0.014 to 0.036 µW/µm² during the 10-s PALM data acquisition period with an e-folding time $\tau$ of 10.6 s (i.e., the photoconversion laser intensity at time $t$ = 0.014 µW/µm² × exp [t/10.6], and thus at the end of the 10-s data accumulation time = 0.036 µW/µm²), to keep the number densities of mEos3.2 spots in the FA region similar during the e-folding period. The photoconverted mEos3.2-paxillin was excited using a 561-nm laser (Jive, 500 mW; Cobolt) at 30 µW/µm², and single-molecule spots were recorded at a frame rate of 1 kHz (with an integration time of 1 ms).

The fluorescence image isolated by a bandpass filter of 572–642 nm (FF01-607/70; Semrock) was projected onto the photocathode of the image intensifier of the developed ultrafast camera system, and the reconstructions of PALM images with a pixel size of 10 nm were performed using the ThunderSTORM plugin for ImageJ (Ovesny et al., 2014) installed in the Fiji package (Schindelin et al., 2012). The 1-frame gap closing was applied to sequences of captured images using the "Merging" postprocessing function, with the maximum off frame of 1 and the maximum search distance of $\sqrt{2}$ × 2 × (mean localization precision for mEos3.2 [29 nm; Fig. 2 A-d]) = 82 nm. Gaussian rendering was performed with a localization precision of 29 nm.

### Spatial resolutions of the ultrafast PALM images of caveolin-1-mEos3.2 (Figs. 2 and 3)
The spatial resolutions of the typical images shown in Fig. 3, A-a and B-a, were evaluated by a parameter-free decorrelation analysis (Descloux et al., 2019) and were 75 and 77 nm, respectively. However, note that these values do not simply represent the instrumental specs. These resolution values are strongly influenced not only by the instrumental parameters but also by the number densities of the fluorescent probes in the observed structure and the structure's shape (Lelek et al., 2021). Meanwhile, if conventional sCMOS had been used for the fixed cells (although the data acquisition duration would have become 30 times slower and the view-field size would have been much smaller), it might have provided a spatial resolution of 59 (77/1.3) nm (for the factor of 1.3, see the main text related to Fig. 2 A-d).

### Detection and characterization of FA-protein islands using Voronoï-based segmentation of the PALM images (Fig. 4 and Fig. S2)
The SR-Tesseler software based on Voronoï polygons (Levet et al., 2015) was applied to the locations of mEos3.2-paxillin to

automatically segment the PALM data. FA contour detection was performed by using a thresholding Voronoï polygon density factor of 1.45 (within the region of interest), with a minimum area corresponding to a circle of 178 nm in diameter. The contour of the islands of FA proteins was determined in basically the same way, by using a thresholding Voronoï polygon density factor of 1.45 (within the FA contour), with a minimum area corresponding to a circle of 13 nm in diameter and a minimum molecular localization of six copies. The FA-protein clusters containing ≤5 copies were not included as islands because these smaller clusters abundantly exist outside the FAs. The diameter of each island was defined as $2 \times \sqrt{polygon\ area\ for\ the\ island/\pi}$.

**Expression of Halo-tagged FA proteins in the cell and HMSiR labeling for ultrafast live-cell dSTORM observations (Fig. S3 A and Figs. 5 and 6)**
A stable MEF line expressing mEos3.2-paxillin without endogenous paxillin was generated by incubating the paxillin-null MEFs (Sero et al., 2011) with the cDNA encoding mEos3.2-paxillin subcloned into the lentiviral transfer vector pCDH-EF1-IRES-Blast (a gift from N. Kioka, Kyoto University, Kyoto, Japan; Hino et al., 2019), followed by selection with 10 µg/ml blasticidin (InvivoGen). The resultant cells expressed mEos3.2-paxillin at 0.64× the level of the endogenous paxillin level in the parental MEF line (a gift from N. Kioka, Kyoto University, Kyoto, Japan; Hino et al., 2019). This was determined by Western blotting in the following way. The MEFs cultured to ≈80% confluency in a 60-mm plastic dish were extracted on ice for 10 min with 0.14 ml of ice-cold PBS (pH 7.4) containing 1% SDS. The extract was incubated at 98°C for 20 min and then centrifuged at 13,000 rpm for 10 min at room temperature. The supernatant was collected and the total protein concentration was determined by the BCA assay (Thermo Fisher Scientific). The proteins in the extract were separated by SDS-PAGE and then Western blotting was performed using mouse anti-paxillin (1:1,000; clone 5H11; Invitrogen) and mouse anti-GAPDH (clone 1E6D9; ProteinTech) primary antibodies, and HRP-conjugated goat anti-mouse IgG secondary antibodies (Jackson). The protein bands were detected by the chemiluminescence (Chemi-Lumi L, nacalai tesque) and quantitative analysis of the band intensities was performed by the ImageJ software. The paxillin signal intensities were normalized to the GAPDH intensities.

The cloned mEos3.2-paxillin–rescued MEFs were transfected with the cDNA encoding Halo-paxillin, Halo-talin, Halo-FAK, Halo-vinculin, Halo-integrin β1, or Halo-integrin β3 using a Nucleofector 2b device and cultured for 8 h. The expressed proteins were labeled with the spontaneously blinking dye HMSiR (Uno et al., 2014). Briefly, the cells were incubated in culture medium containing 100 nM HMSiR-conjugated Halo-ligand (HaloTag SaraFluor 650B Ligand; Goryo Chemical) for 15 h, washed three times with the cell culture medium, removed from the cell culture dish by trypsinization, and then replated and cultured on 12-mm-diameter glass-bottom dishes coated with fibronectin for at least 3 h before dSTORM observations.

The cDNAs encoding Halo-paxillin, Halo-talin (human talin 1, Kazusa DNA Research Institute: ORK01622), Halo-FAK (mouse FAK isoform 3, a gift from C.H. Damsky, University of California,

San Francisco, San Francisco, CA, USA), and Halo-vinculin (human vinculin, Kazusa DNA Research Institute: FXC01835) with a 15 amino-acid SGGGG ×3 linker sequence were generated. The cDNA encoding Halo-integrin β3 (human integrin β3, NCBI reference sequence: NM_000212.2; a gift from J.C. Jones, Northwestern University, Evanston, IL, USA; Tsuruta et al., 2002) was subcloned into the pOsTet15T3 vector, with the IL6 signal peptide before Halo and a 45-base linker (15 amino acids, with the sequence SGGGG ×3) between Halo and integrin β3. The cDNA encoding Halo-integrin β1 was produced by replacing the cDNA encoding integrin β3 in the Halo-integrin β3 plasmid with the cDNA encoding integrin β1 (human integrin β1, NITE Biological Resource Center: AK291697).

**Calculations for obtaining the paxillin copy number/detected islands from the PALM and dSTORM data (Fig. 5 E)**
The median copy number of paxillin molecules located in a paxillin-enriched island in a non-transfected parental MEF was estimated to be 26 copies/island from the PALM data in the following way. The median number of mEos3.2-paxillin detections (localizations = total number of fluorescent spots that appeared in the raw PALM image) per a detected island obtained from the PALM data was 14 (Fig. 5 E bottom). Using the mean number of on-events (detections) per mEos3.2 molecule (overcounting) = 1.4 (Fig. 2 A-e), the fraction of fluorescent mEos3.2 (vs. total mEos3.2) = 0.60 (Baldering et al., 2019) and the expression level of endogenous paxillin in a parental MEF = [1/0.64]x of mEos3.2-paxillin in an mEos3.2-paxillin–rescued MEF, we calculated it as 14/0.60/1.4/0.64 (which gives 26.0).

The median copy number of paxillin molecules located in a paxillin-enriched island in a non-transfected parental MEF was estimated to be 33 copies/island from the dSTORM data. This was evaluated in the following way. The median number of HMSiR-Halo-paxillin detections (localizations = total number of fluorescent spots that appeared in the raw dSTORM image) per detected island obtained from the dSTORM data was 13 (Fig. 5 E top). Using the mean number of on-events (detections) per HMSiR molecule (overcounting) = 2.7 (Fig. S3 B-d), the fraction of fluorescent Halo-paxillin (vs. total Halo-paxillin; i.e., the labeling efficiency) = 0.90 (Morise et al., 2019), and the expression level of endogenous paxillin in the parental MEF = [1/0.16]x of Halo-paxillin in tagged-paxillin-expressing paxillin-KO MEF, we calculated it as 13/0.90/2.7/0.16 (which gives 33.4).

**Ultrafast live-cell dSTORM of HMSiR-labeled FA proteins and simultaneous ultrafast PALM of mEos3.2-paxillin for characterizing the FA-protein islands (Figs. 5, 6, and 7; and Figs. S3 and S4)**
The data acquisition for ultrafast live-cell dSTORM using the HMSiR probe and the simultaneous data acquisitions for ultrafast live-cell dSTORM and ultrafast live-cell PALM using mEos3.2 were performed on the basal PM at 37 ± 1°C. The ultrafast dSTORM observations of the HMSiR-labeled FA proteins were performed at a frame rate of 1 kHz for 10 s (the same rate and acquisition time for PALM) using a 660-nm laser (Ventus, 750 mW; Laser Quantum) at 23 µW/µm². In the excitation arm, a multiband mirror (ZT405/488/561/647rpc, Chroma)

was employed. The leakage of the intense 561-nm excitation laser beam into the emission arm was blocked by a notch filter (NF03-561E; Semrock) placed right before the entrance into the detection arm. The fluorescence images of mEos3.2 and HMSiR were separated by a dichroic mirror (ZT647rdc, Chroma) into two detection arms with bandpass filters of 572–642 nm for mEos3.2 (FF01-607/70; Semrock) and 672–800 nm for HMSiR (FF01-736/128; Semrock). Each detection arm was equipped with the developed ultrafast camera system and the images were projected onto the photocathode of the camera system (more specifically, the photocathode of the image intensifier). The images from the two cameras were spatially corrected and superimposed with subpixel precisions, as described previously (Koyama-Honda et al., 2005). The reconstructions of a dSTORM image with a pixel size of 10 nm were performed using the ThunderSTORM plugin with the 1-frame gap closing by the "Merging" post-processing function, with the maximum off-frame of 1 and the maximum search distance of $\sqrt{2} \times 3 \times$ (mean localization precision for HMSiR [19 nm; Fig. S3 A–c]) = 81 nm. Gaussian rendering was performed with a localization precision of 19 nm. In longer dSTORM observations, such as for 60 s using HMSiR-labeled Halo-paxillin, a frame rate of 250 Hz and a 660-nm excitation laser intensity of 2.2 µW/µm² were employed.

### Autocorrelation and crosscorrelation analyses for characterizing the spatial organization of the FA-protein islands (Figs. 5 and 6)

To examine the spatial organization of the FA protein islands, in addition to the Voronoï tessellation analysis, we employed the spatial autocorrelation and crosscorrelation analysis for the fluorescent spots located in the FA-protein islands (Voronoï's polygons) in each FA. The autocorrelation function, $g(r)$, represents the probability of finding a second molecule of the same molecular species located at a distance "r" away from a given molecular localization and thus can address the distribution of single molecules. The crosscorrelation function, $c(r)$, represents the probability of finding a second molecular species located at a distance "r" away from a given localization of the first molecular species and thus can address the codistribution of the two molecules. They were computed using Fast Fourier Transforms, which can account for complex boundary shapes without additional assumptions, as described by Veatch et al. (2012).

### Fluorescence labeling of TfR and integrin β3 for observing diffusion in the FA region: cDNA construction, expression, and labeling in T24 cells

The cDNA encoding human TfR (GenBank: M11507.1) fused to the Halo-tag protein at the TfR's N-terminus (Halo-TfR) was generated by replacing the cDNA encoding the EGFP protein in the EGFP-TfR plasmid with that of the Halo-tag protein (Promega), with the insertion of a 45-base linker (15 amino acids, with the sequence SGGGG ×3) between Halo and TfR. The cDNA encoding human integrin β3 tagged with ACP (ACP-integrin β3) was subcloned into the pOsTet15T3 vector, with the CD8 signal peptide before ACP and a 15-base linker (five amino acids, with the sequence SGGGG) between ACP and integrin β3.

The leaky expression without doxycycline induction was useful for single-molecule imaging and tracking, and avoiding overexpression. The sequences of all newly generated constructs were confirmed by DNA sequencing.

T24 cells were transfected with the cDNA encoding mGFP-paxillin using the Lipofectamine LTX and Plus reagents (Invitrogen) and with other cDNAs by using Nucleofector 2b (Lonza), following the manufacturers' recommendations. To covalently link TMR to Halo-TfR, T24 cells coexpressing Halo-TfR and mGFP-paxillin were incubated in Hanks' balanced salt solution, buffered with 2 mM N-Tris(hydroxymethyl)methyl-2-aminoethanesulfonic acid (TES; Dojindo) at pH 7.4 (HT medium), containing 10 nM TMR-conjugated Halo-ligand (Promega) at 37°C for 1 h, and then washed three times with HT medium. The remaining unbound ligand in the cytoplasm was removed by incubating the cells in HT medium for 30 min and then washing the cells three times with HT medium. To form the covalent complex between SeTau647 (SETA BioMedicals) and ACP-integrin β3, T24 cells coexpressing ACP-integrin β3 and mEos3.2-paxillin were incubated in cell culture medium (Ham's F12 medium supplemented with 10% fetal bovine serum) containing 100 nM SeTau647-CoA (Shinsei Chemical Company), 1 µM ACP synthase (New England Biolabs), and 10 mM MgCl₂ for 30 min at 37°C, and then washed three times with HT medium. To form the covalent complex between SeTau647 and Halo-integrin β3, T24 cells coexpressing Halo-integrin β3 and mEos3.2-paxillin were incubated in a cell culture medium containing 100 nM SeTau647-conjugated Halo-ligand (Shinsei Chemical Company) at 37°C for 1 h and then washed three times with HT medium. We previously found that under these conditions, >90% of ACP and Halo could be complexed with their fluorescent ligands (Morise et al., 2019). ACP-integrin β3 was used for observing temporary immobilizations at a frame rate of 250 Hz due to the smaller size of the ACP-tag protein as compared with the Halo-tag protein, for a possibly smaller effect on the integrin β3 binding to FA-protein islands. Halo-integrin β3 was used for observing hop diffusion at 6 kHz due to the better stability of the SeTau647 dye bound to the Halo-tag protein at higher laser intensities, as compared with the dye bound to the ACP-tag protein.

### Ultra-high-speed imaging of single Halo-TfR labeled with TMR (Fig. 8 and Fig. S4)

Individual TMR-Halo-TfR molecules located on the basal PM were observed at 37 ± 1°C using the TIR illumination mode of a home-built objective lens-type TIRF microscope (based on an Olympus IX70 inverted microscope), which was modified and optimized for the camera system developed in the companion paper. A 532-nm laser (Millennia Pro D2S-W, 2W, Spectra-Physics) was attenuated with neutral density filters, circularly polarized, and then steered into the edge of a high numerical aperture (NA) oil immersion objective lens (UAPON 150XOTIRF, NA = 1.45, Olympus), focused on the back-focal plane of the objective lens. The TIR illumination intensities at the sample plane were 14 and 0.16 µW/µm² for the camera frame rates of 6 kHz (Fig. 8) and 60 Hz (Fig. S4), respectively. Right before recording the images of single TMR-Halo-TfR molecules, the

mGFP-paxillin image was obtained in the same view field by using the TIR illumination (0.063 µW/µm² at the specimen using a Spectra-Physics Cyan-PC5W 488-nm laser using a frame rate of 60 Hz, averaged over 10 s).

### Simultaneous live-cell PALM of mEos3.2-paxillin to visualize the paxillin-enriched islands and high-speed imaging of SeTau647-ACP (or Halo)-integrin β3 to track their single-molecule movements in the FA archipelago (Fig. 9)

Simultaneous data acquisitions of ultrafast live-cell PALM of mEos3.2-paxillin and ultrafast SFMI of ACP-integrin β3 or Halo-integrin β3 labeled with SeTau647 were performed on the basal PM at 37 ± 1°C using the same microscope setup employed for simultaneous ultrafast PALM for mEos3.2 and ultrafast dSTORM for the HMSiR probe. Single-molecule imaging of SeTau647 bound to integrin β3 was performed (simultaneously with the ultrafast live-cell PALM data acquisition) at a frame rate of 250 Hz for visualizing the integrin β3 binding to the paxillin-enriched islands (using ACP-integrin β3) or 6 kHz for visualizing the integrin β3 hop diffusion inside the FA (using Halo-integrin β3), using a 660-nm laser at 0.84 or 8.4 µW/µm², respectively.

### Online supplemental material

Fig. S1 illustrates that the single-molecule localization precision is minimally affected by the PRNU of the developed camera system. Fig. S2 shows the expression level of mEos3.2-paxillin in the clonal T24 cells used in this study, the determination of the appropriate thresholding density factor for the Voronoï tessellation analysis, and the effects of single-molecule localization precisions in the PALM and dSTORM data acquisition on the FA-protein island diameters determined by the tessellation analysis. Fig. S3 demonstrates the optimization of the live-cell dSTORM data acquisition conditions for HMSiR-labeled FA molecules. Fig. S4 exhibits the diameter distributions of the islands of various FA-proteins in mEos3.2-paxillin–rescued MEFs obtained by the tessellation analysis of dSTORM and PALM images and displays the representative T24-cell dSTORM image sequences for 60 s, demonstrating that synchronized paxillin recruitment to loose paxillin-enriched island clusters occurs in T24 cells as well as in MEFs (Fig. 7, A and B). Fig. S5 shows that TfR molecules can diffuse within the FA, but in certain subregions, their diffusibility is limited. Video 1 shows typical ultrafast PALM data acquisition and reconstruction with mEos3.2-paxillin expressed in a live T24 cell. Almost the entire basal PM (35.3 × 35.3 µm²) can be imaged with the data acquisition rate of 1 kHz for a period of 10 s (10,000 frames). Video 2 and Video 3 show typical time-dependent changes of HMSiR-labeled Halo-paxillin in a MEF cell and a T24 cell, respectively, every 1 s for 60 s (250-Hz data acquisition for 15,000 frames; the ultrafast live-cell dSTORM images were reconstructed with a shifting time window of 2,500 frames or 10 s, and the time is shifted every 1 s to create the video sequence). Video 4 displays the typical hop diffusion of single Halo-TMR-TfR molecules occurring both outside and inside the FA regions in the basal PM of T24 cells, observed at 6 kHz (every 0.167 ms). Videos 5 and 6 show representative behaviors of single integrin β3 molecules (ACP-SeTau647) captured at 250 Hz (every 4 ms), which exhibited temporary

binding (Video 5) and longer binding (longer than the entire observation period of ≈1 s; Video 6) to the paxillin-enriched islands, visualized by the ultrafast live-cell PALM of mEos3.2-paxillin. Video 7 shows a typical recording of an integrin β3 molecule (Halo-SeTau647) at 6 kHz (every 0.167 ms), exhibiting hop diffusion in the bulk basal PM and continuing to undergo hop diffusion after entering the FA region, visualized by the live-cell PALM of mEos3.2-paxillin.

### Data availability

Data supporting the findings of this study are available from the corresponding author upon reasonable request. The code is available from the corresponding author upon reasonable request.

### Acknowledgments

We thank Profs. Y. Miwa of the University of Tsukuba (Tsukuba, Japan), J.C. Jones of Northwestern University (Evanston, IL, USA), I.R. Nabi of the University of British Columbia (Vancouver, Canada), L. Looger of the Janelia Research Campus (Ashburn, VA, USA), D.E. Ingber of Harvard University (Cambridge, MA, USA), and C.H. Damsky of the University of California, San Francisco (San Francisco, CA, USA) for their kind gifts of the pOsTet15T3 vector, the cDNA encoding human integrin β3, the human EGFR-YFP plasmid, the mEos2 plasmid, the paxillin-null MEFs, and the cDNA encoding mouse FAK isoform 3, respectively. We thank Profs. T. Ichikawa and N. Kioka of Kyoto University (Kyoto, Japan) for their kind gifts of immortalized wild-type MEFs and the lentiviral transfer vector, and their support in vector generation and infection. We also thank Mr. K. Hanaka and Prof. M. Kengaku of Kyoto University (Kyoto, Japan) for their enthusiastic encouragement of this research, Mss. M. Yahara, A. Chadda, and H. Hijikata for constructing various cDNAs, Ms. J. Kondo-Fujiwara and Mr. K. Kanemasa for preparing the figures, and the members of the Kusumi laboratory for helpful discussions. We are grateful to Prof. P.T. Kanchanawong of the National University of Singapore and to the late Prof. K. Jacobson of the University of North Carolina (Chapel Hill, NC, USA) for their critical reading of the manuscript and constructive comments.

This work was supported in part by Grants-in-Aid for Scientific Research from the Japan Society for the Promotion of Science (Kiban B to T.K. Fujiwara [16H04775, 20H02585], Kiban B to K.G.N. Suzuki [18H02401, 21H02424], and Kiban S and A to A. Kusumi [16H06386 and 21H04772, respectively]), Grants-in-Aid for Challenging Research (Exploratory) from Japan Society for the Promotion of Science to T.K. Fujiwara (18K19001) and A. Kusumi (22K19334), and Grants-in-Aid from the Ministry of Education, Culture, Sports, Science and Technology of Japan for Transformative Research Areas (A) to T.K. Fujiwara (21H05252) and for Innovative Areas to K.G.N. Suzuki (18H04671), a Japan Science and Technology Agency grant in the program of the Core Research for Evolutional Science and Technology in the field of "Biodynamics" to A. Kusumi and in the field of "Extracellular Fine Particles" to K.G.N. Suzuki (JPMJCR18H2), a Japan Science and Technology Agency grant in the program of

"Development of Advanced Measurements and Analysis Systems" to A. Kusumi and T.K. Fujiwara, and by a grant from the Takeda Foundation to K.G.N. Suzuki. Institute for Integrated Cell-Material Sciences of Kyoto University is supported by the World Premiere Research Center Initiative of Ministry of Education, Culture, Sports, Science and Technology of Japan.

Author contributions: T.K. Fujiwara and A. Kusumi conceived and formulated the project. T.K. Fujiwara, T.A. Tsunoyama, S. Takeuchi, Y. Nagai, K. Iwasawa, and A. Kusumi developed the microscope station for ultrafast SFMI, ultrafast PALM, and ultrafast dSTORM. T.K. Fujiwara, T.A. Tsunoyama, Y.L. Nemoto, L.H. Chen, A.C.E. Shibata, K.G.N. Suzuki, and A. Kusumi designed the biological experiments and participated in discussions. T.K. Fujiwara performed virtually all of the ultrafast SFMI, ultrafast PALM, and dSTORM experiments and the data analysis. T.K. Fujiwara, Z. Kalay, T. Kalkbrenner, K.P. Ritchie, and A. Kusumi evaluated the data. T.K. Fujiwara, Z. Kalay, and A. Kusumi wrote the manuscript, and all authors discussed the results and participated in revising the manuscript.

Disclosures: The authors declare no competing interests exist.

Submitted: 28 October 2021

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

# Supplemental material

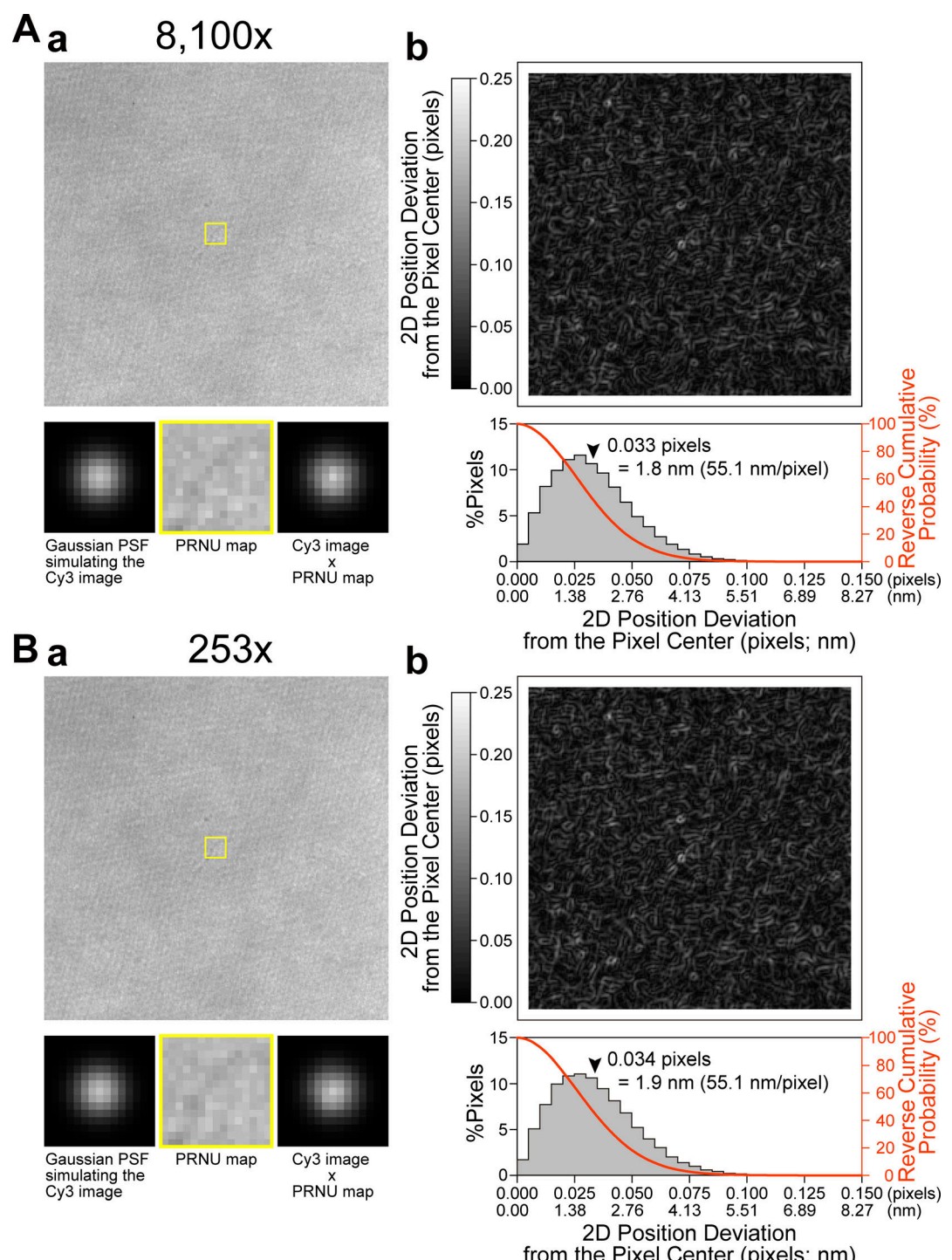

**Figure S1. The PRNU of the developed camera system scarcely affects the single-molecule localization precision, examined at 10 kHz. Two image intensifier amplifications were employed: 8,100× (A) and 253× (B). (A and B)** (a) Images used for evaluating the PRNU effects of the developed camera system on the localization precision of a single Cy3 molecule. Top: 256 × 256–pixel images representing the PRNU of the developed camera system, obtained by averaging images over 40,000 consecutive frames recorded at 10 kHz under uniform illumination, so that the mean pixel intensity counts became 515 ± 34 (A) and 513 ± 32 (B; SD for 256 × 256 pixels), which are approximately half of the maximum intensity count of 10 bits. The uniform illumination was generated by Köhler illumination, using the halogen lamp of the microscope and a 572- to 642-nm bandpass filter (FF01-607/70, Semrock). Bottom: Modulation of the image of a single Cy3 molecule by PRNU, evaluated by calculation. Left: The Cy3 image was approximated by an ideal two-dimensional Gaussian point spread function (PSF) in the 15 × 15–pixel region, based on an experimentally determined SD of 2.2 pixels for 50 Cy3 molecules immobilized on the glass, obtained by the TIR illumination at 79 μW/μm² and a peak intensity count of 511 (half of the maximum intensity count of 10 bits). In the actual imaging experiments, we employed 55.1 nm/pixel: 2.2 pixels = 123 nm. The PSF peak was placed at the center of the 15 × 15–pixel region. Middle: The 15 × 15–pixel yellow regions shown in the top images are magnified. Right: Images on the left and middle were multiplied pixel by pixel and normalized, generating the PRNU-modulated images of a single Cy3 molecule. **(b)** The effect of PRNU on the single-molecule localization precision is quite limited. Top: Maps of the 2D position deviation from the pixel center

(coded on the gray scale). These maps were generated by moving the 15 × 15–pixel image of an ideal Gaussian PSF simulating the Cy3 image (a, bottom left), scanning over the 256 × 256–pixel PRNU images (a, top) pixel-by-pixel, and calculating the 2D position deviation at every position in the scan. Bottom: Distributions of the 2D position deviations, showing that the mean deviations are 0.033 and 0.034 pixels (= 1.8 and 1.9 nm at 55.1 nm/pixel; $n$ = 57,600 pixels; arrowheads) for A and B, respectively, which are comparable to the typical PRNU effect found with the EM-CCD camera (Pertsinidis et al., 2010). Furthermore, the reverse cumulative distributions (shown in red) indicate that 95% of the 2D position deviations are within the range of 0.066 pixels (= 3.6 nm) and 0.068 pixels (= 3.7 nm) for A and B, respectively. These results suggest that the effects of PRNU on the single-molecule localization precision are limited and almost identical at amplifications of 8,100× (A) and 253× (B).

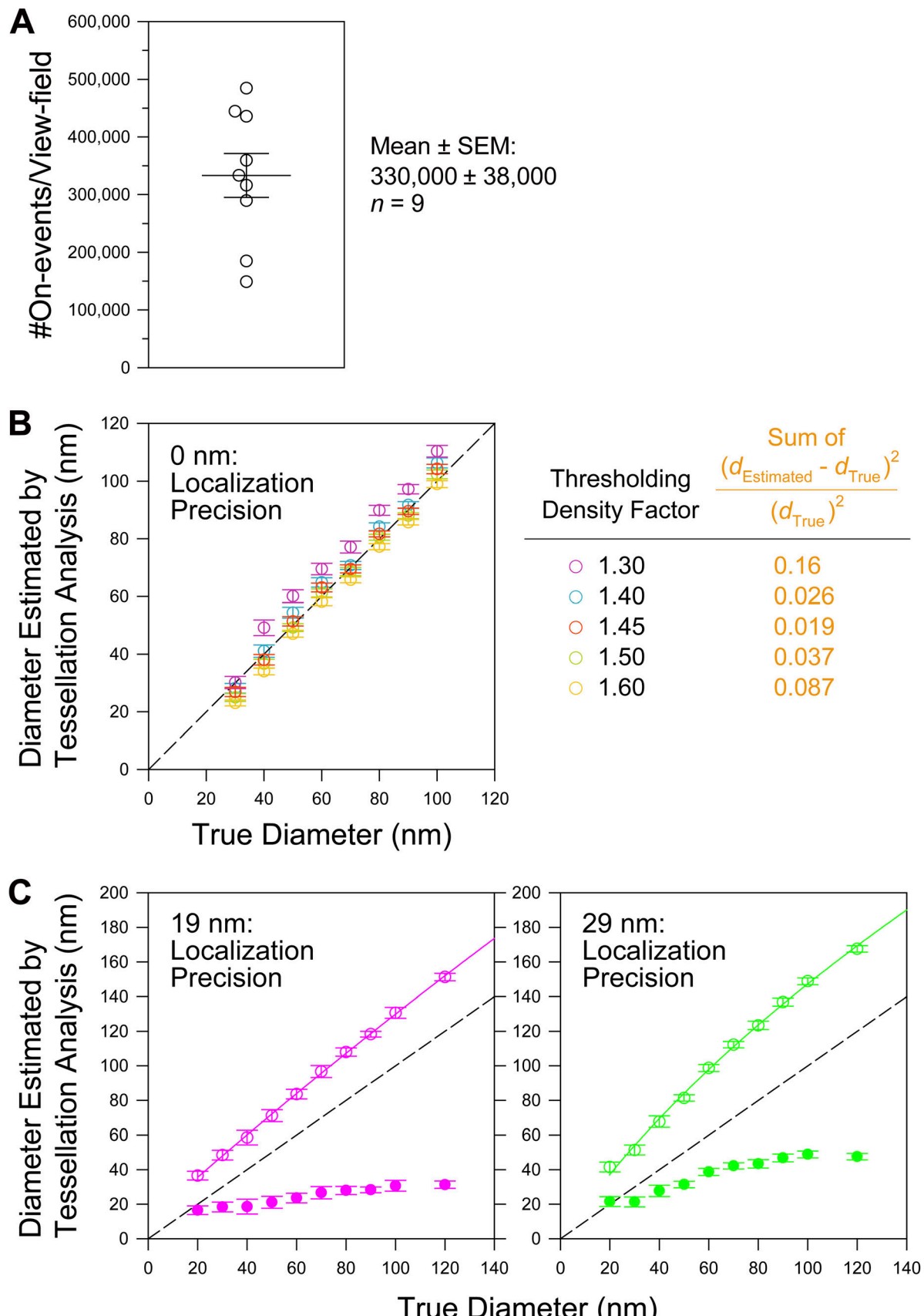

Figure S2.   **Determination of the expression level of mEos3.2-paxillin in the clonal T24 cells stably expressing mEos3.2-paxillin used in this study (0.9× the endogenous paxillin; i.e., total paxillin ≈1.9× of the endogenous paxillin), determination of the thresholding density factor for the Voronoï**

**segmentation analysis, and the correction method in the estimate of the FA-protein island diameters for limited single-molecule localization precisions of PALM and dSTORM images (using Voronoï segmentation analysis). (A)** The mEos3.2-paxillin–expressing T24 cell clone used in this work expresses mEos3.2-paxillin at ≈0.9× of the endogenous paxillin in non-transfected cells, and thus the total paxillin amount in these cells will be ≈1.9× the endogenous paxillin without transfection, assuming that the endogenous paxillin expression level is unchanged after the mEos3.2-paxillin expression. The figure shows that the number of mEos3.2 on-events (the number of fluorescent spots in the raw PALM image = the number of localizations) detected in the entire view-field was 330,000 ± 38,000 ($n$ = 9 cells). Based on this value, the amount of expressed mEos3.2-paxillin was roughly estimated in the following way. Assuming that the view-fields employed here represent about two-thirds of the entire basal PM (including the FAs) and that 70∼90% of expressed mEos3.2-paxillin molecules are recruited to the basal PM, and since the mean number of on-events per each mEos3.2 molecule = 1.4 (Fig. 2 A-e) and only ≈60% of mEos3.2 is fluorescent (Baldering et al., 2019), the number of expressed mEos3.2-paxillin molecules is estimated to be 655,000–842,000 copies/cell (3,300,00×[3/2]/[0.7–0.9]/1.4/0.60). The copy number of endogenous paxillin expressed in a T24 cell is unknown, but since 920,000 copies of zyxin and 810,000 copies of VASP, which are both essential components of FAs, are expressed in a T24 cell (Tsunoyama et al., 2021 *Preprint*), it would not be unreasonable to assume that an approximately similar number of endogenous paxillin copies exists in a T24 cell (say, ≈850,000 copies). Therefore, after the expression of mEos3.2-paxillin, the total paxillin might be over-expressed by factors of 1.8–2.0 ([655 k + 850 k]/850 k∼[842 k + 850 k]/850 k; assuming that endogenous paxillin expression was not decreased due to the overexpression of mEos3.2). Therefore, we will employ the copy number at which the expressed mEos3.2-paxillin is 0.9× the level of endogenous paxillin and the total number of paxillin molecules is 1.9× the level of endogenous paxillin in non-transfected T24 cells. **(B and C)** Determinations of the proper Voronoï polygon thresholding density factor (B) and the effects of single-molecule localization precisions (19 and 29 nm) on the island diameters evaluated by the Voronoï tessellation analysis (Levet et al., 2015) for the PALM (29-nm precision) and dSTORM (19-nm precision) images (the relationship of the evaluated diameters with true diameters; C), using Monte Carlo simulations of the PALM and dSTORM images of the paxillin islands. Details of simulation and analysis: A circular paxillin island with a given diameter between 20 and 120 nm was placed at the center of a square with a side length 10 times the circle diameter (except for the 20-nm diameter circle where a side length of 20 times the diameter was employed). Single molecule localization errors of 0 nm (B) and 19 and 29 nm (C; for dSTORM of HMSiR and PALM of mEos3.2) were employed. The positions of the fluorescent spots in the PALM/STORM raw images outside the circle were randomly placed at a number density of 0.002/nm² (a typical number of on-events [fluorescent spots in the PALM/STORM raw images] inside the FA in the experimental images), with added Gaussian noise to account for the localization errors (0, 19, and 29 nm). The locations of the fluorescent spots in the PALM/STORM raw images inside the circle were generated in the same way, with the exception that the number density was increased to 0.02/nm², 10× greater than that of the outside density, which is typical of the number density found in experimentally observed islands (0.019/nm² for the identified islands with diameters in the range of 30∼100 nm). The average number densities of fluorescent molecules will be [1/1.4]x and [1/2.7]x of these on-event densities (the densities of the fluorescent spots) for mEos3.2 and HMSiR, respectively (Fig. 2 A-e and Fig. S3 A-d), but since the actual blinking number for each molecule will follow the probability distribution described by the geometric function (Hummer et al., 2016), we included this effect in the simulation. 30 images were generated for each condition, and the islands in each image were detected by the Voronoï tessellation analysis with a minimum diameter of 13 nm, and the diameters of the detected circles were determined (see the main text). The diameters evaluated in this manner (mean ± SEM) are plotted against the true diameters used in the simulation. **(B)** Determination of the optimal Voronoï polygon thresholding density factor based on the best estimate of island diameters, showing that a thresholding density factor of 1.45 provides the most accurate estimation of the island diameters. The figure shows the evaluated diameters plotted against the true diameters (diameters used for simulation) for various Voronoï polygon thresholding density factors. The single-molecule localization error was set at 0 nm for this determination. From these plots, the closeness of the estimated diameter ($d_{\text{Estimated}}$) to the true diameter ($d_{\text{True}}$; dashed line represents $d_{\text{Estimated}} = d_{\text{True}}$) was estimated as the sum of ([$d_{\text{Estimated}} - d_{\text{True}}$]²/$d_{\text{True}}^2$) determined every 10 nm in the range of 30–100 nm (orange text). See the table on the right. **(C)** The relationship of the evaluated island diameters with the true diameters (open circles). The curves represent the best-fit quadratic functions for the plots of open circles in the range of $d_{\text{True}}$ ≥20 nm, which were used to estimate the true mean diameters of the FA-protein islands in Fig. 4 G, Fig. 5 D, and Fig. S4 A and B. The dashed linear lines indicate the ideal case of $d_{\text{Estimated}} = d_{\text{True}}$. The filled circles show $d_{\text{Estimated}} - d_{\text{True}}$ (mean ± SEM; for both open and filled circles). The differences increase with an increase of the true diameter, but appear to level off from around $d_{\text{True}}$ ≈100 nm.

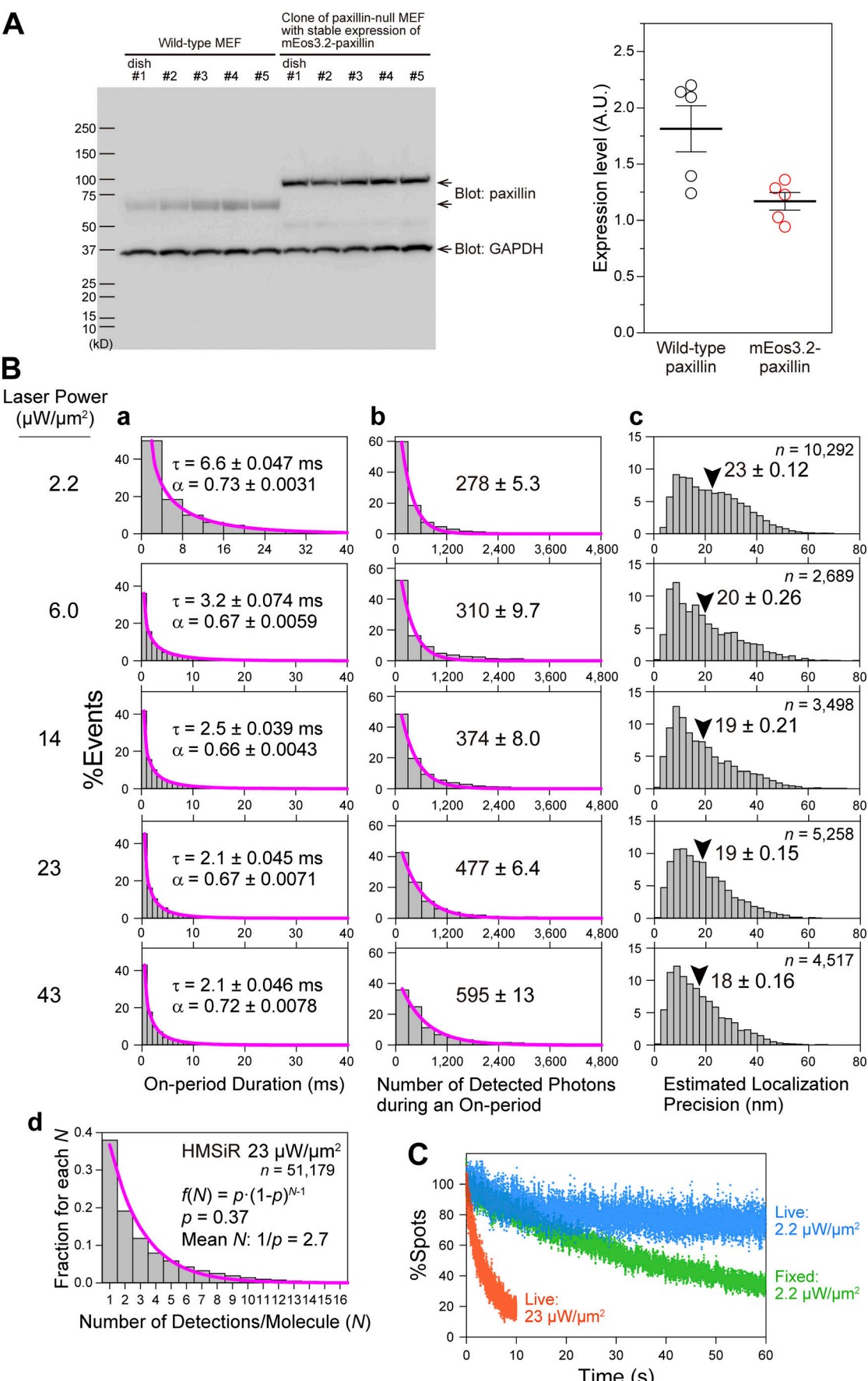

Figure S3.   **Live-cell dSTORM data acquisition conditions established for HMSiR-labeled FA molecules. (A)** The MEF cells used in this study stably express mEos3.2-paxillin at the level of 0.64× of the endogenous paxillin in the parental MEFs. The figure shows the paxillin Western blot membrane and the plots of paxillin band intensities of the parental MEFs and paxillin-null MEFs (Sero et al., 2011) rescued by the stable expression of mEos3.2-paxillin (cloned), showing that mEos3.2-paxillin–rescued paxillin-null cells (mEos3.2-paxillin–rescued MEFs) express 0.64× the endogenous paxillin in the parental MEF cells. For the dSTORM experiments, the mEos3.2-paxillin–rescued MEFs were further transfected with the Halo-paxillin cDNA for transient expression, and the expressed Halo-paxillin was labeled with HMSiR at ≈ 90% efficiency (Morise et al., 2019). dSTORM observations were performed using live cells exhibiting similar levels of the HMSiR-labeled Halo-paxillin signal. These cells were found to express Halo-paxillin at the level of 0.16× the endogenous paxillin in the parental MEF line, based on the following observations and calculations (and thus these cells express a total of 0.8× the endogenous paxillin; i.e., 0.64× for mEos3.2-paxillin and 0.16× for Halo-paxillin). The ratio of detected spot densities of mEos3.2-paxillin vs. those of HMSiR-Halo-paxillin in the FA was ≈1.4:1. Considering that (1) the number of on-events for a single HMSiR molecule is 2.7 (B d) and that for mEos3.2-paxillin is 1.4 (Fig. 2 A-e), and (2) 90% of HMSiR and 60% of mEos3.2 are fluorescent, the copy number ratio of mEos3.2-paxillin vs. Halo-paxillin including non-fluorescent molecules is estimated to be ≈4.1:1 (1.4/1.4/0.6: 1/2.7/0.9 = 1.667:0.411). Therefore, the amount of Halo-paxillin in these MEF cells is 0.64x/4.1 = 0.16x (0.64x is the ratio of mEos3.2-paxillin vs. endogenous paxillin found in parental MEF cells here) of the amount of endogenous paxillin in the parental cell line. **(B and C)** First, we will give an overall explanation, and the detailed legends will be presented later. For establishing optimal dSTORM data acquisition conditions, we first examined the on-period durations of HMSiR because this will limit the data acquisition frame rate for dSTORM. With an increase of the TIR excitation laser illumination intensity at a wavelength of 660 nm from 2.2 to 43 $\mu$W/$\mu$m$^2$ in the sample plane, the on-period durations gradually decreased and plateaued at 2.1 ms at a laser intensity of 23 $\mu$W/$\mu$m$^2$ (Fig. S3 B-a), showing that further increases of the laser intensity will not improve the data acquisition frame rate. Therefore, we decided to use a camera frame rate of 1 kHz for dSTORM data acquisition, which is the same rate as that employed for PALM data acquisition using mEos3.2. Further increasing of the illumination laser intensity beyond 23 $\mu$W/$\mu$m$^2$ continued to increase the numbers of detected photons during the on-period of a single HMSiR molecule, with a concomitant improvement of the localization precision of a single dye molecule for a single on-event. However, the extent of improvement was quite limited (Fig. S3 B-b and -c), and thus increasing the laser intensity beyond 23 $\mu$W/$\mu$m$^2$ was deemed not worthwhile, due to the increased probability of photo-damage to live cells. Since the illumination by a 561-nm excitation laser intensity at 23 $\mu$W/$\mu$m$^2$ for 1 min had minimal impact on cell viability (Fig. 2, D and E, in the companion paper), and since this laser intensity is about optimal for the 1 kHz data acquisition rate for HMSiR (on-duration of 2.1 ms; Fig. S3 B-a), we chose to use the 660-nm laser intensity of 23 $\mu$W/$\mu$m$^2$ for the dSTORM experiments. At this laser excitation power density, the on-period reached the plateau at 2.1 ms, providing 477 ± 6.4 photons and a single-molecule localization precision of 19 ± 0.15 nm per on-event (Fig. S3 B-b and -c), and the mean number of on-events per HMSiR molecule was 2.7 (Fig. S3 B-d). At a laser intensity of 23 $\mu$W/$\mu$m$^2$, after the illumination for 10 s (10,000 frames at 1 kHz), ≈80% of HMSiR was photobleached (Fig. S3 C), indicating that the data acquisition for 10 s is close to the optimal conditions for the photon usage. In order to ensure single-molecule detection conditions for all experiments while employing a 660-nm laser intensity of 23 $\mu$W/$\mu$m$^2$ for all ultrafast dSTORM data acquisitions, we adjusted the expression levels of Halo-tagged proteins and/or HMSiR labeling efficiencies. **(B)** The spontaneous blinking characteristics of HMSiR bound to Halo-paxillin located on the basal PM of live MEF cells observed at 1 kHz (1-ms frame time) with 660-nm excitation laser intensities of 2.2, 6.0, 14, 23, and 43 $\mu$W/$\mu$m$^2$. (a) Duration of on-periods. The histograms show the distributions of consecutive fluorescent on-periods (with a gap closing of 1 frame). They could be fitted by stretched exponential functions $\varphi(t) = \varphi_0 e^{-(t/\tau)^{\alpha}}$, where $\varphi_0$ is the prefactor, $\alpha$ is the stretching exponent, and $\tau$ is the time constant (Morimatsu et al., 2007; mean ± SEM; SEM was determined as a 68.3% confidence limit for the fitting; the number of on-events observed [$n$] is given in c). (b) Distributions of the numbers of detected photons from a single molecule during a single on-period. The histograms could be fitted with single exponential decay functions, with the decay constants providing the mean numbers of detected photons during an on-period (the SEM was given as a 68.3% confidence limit for the fitting). (c) Histograms of localization precisions for individual on-events of single molecules, which were estimated from the numbers of detected photons during a single on-period using the theoretical equation derived by (Mortensen et al., 2010) with an "excess noise" factor ($F$) of 1.2 determined for the developed camera system (see Fig. S2 of the companion paper). (d) The distribution of the number of detections (on events) for a single HMSiR molecule ($N$) bound to Halo-paxillin and observed in the bottom PM of a chemically fixed MEF at the laser power density of 23 $\mu$W/$\mu$m$^2$. Here, the fixed cell was used to exclude the effect of the continuous paxillin exchange between the FA and the cytoplasm. Each detection was found by examining the proximity of the spots recorded at different frames (with a cutoff time of 10 s, which is a typical dSTORM data acquisition period for the observations at 1 kHz employed for this work) with a cutoff distance of $\sqrt{2} \times 3 \times$ (mean localization precision for HMSiR [19 nm]) = 81 nm. The histogram could be fitted well with the geometric function $f(N) = p\bullet(1-p)^{N-1}$ based on the model for a monomeric blinking fluorophore by Hummer et al. (2016) (magenta curve), providing the P value (fluorophore bleaching probability) = 0.37 and the mean number of detections (on events)/molecule (1/$p$) = 2.7. **(C)** Time-dependent reductions in the numbers of fluorescent spots of HMSiR plotted against the elapsed time after starting the continuous illumination by the 660-nm laser at 2.2 and 23 $\mu$W/$\mu$m$^2$ (frame rates of every 4 and 1 ms, respectively). HMSiR bound to the Halo-paxillin located on the MEF's bottom PM was detected. The plots represent the sum of the spot numbers in five cells for each condition, normalized to 100% at time 0. The reduction was slower in live cells (blue) than in chemically fixed cells (green), indicating that paxillin in the FA is continuously exchanging with that in the cytoplasm in the time scale of a few tens of seconds, consistent with the previous FRAP data (Legerstee et al., 2019). For longer observations (like 60 s; Fig. 7) we employed an excitation laser intensity of 2.2 $\mu$W/$\mu$m$^2$, whereas 23 $\mu$W/$\mu$m$^2$ was used for shorter observations (like 10 s; Figs. 5 and 6). The total number of counted spots: 1,733,927 in 15,000 frames (blue), 2,476,668 in 15,000 frames (green), and 701,702 in 10,000 frames (red).

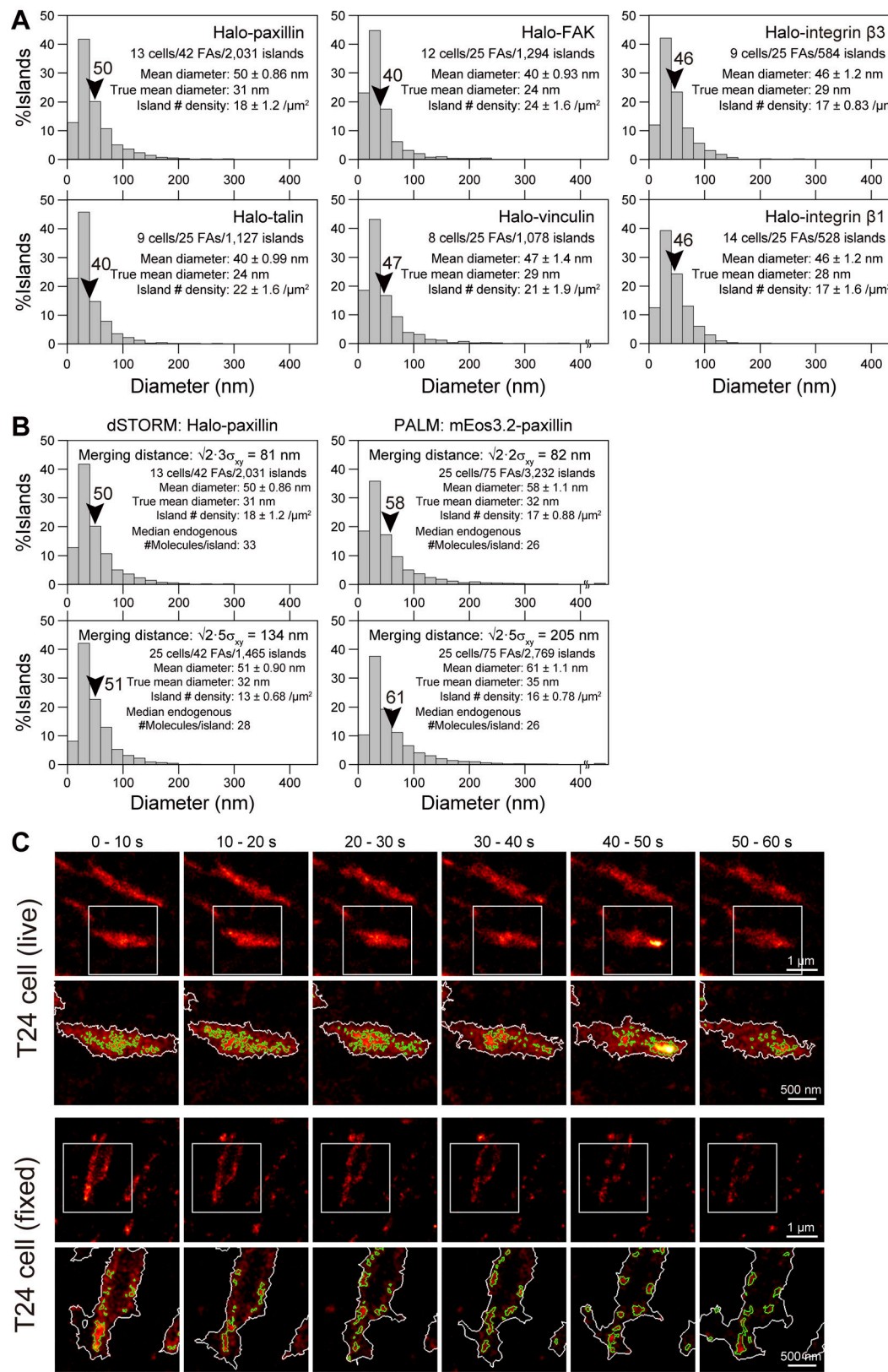

**Figure S4.** **Diameter distributions of the islands of various FA-proteins in the MEFs, obtained by the tessellation analysis of the dSTORM images, the effect of elongating the spot merging distance in the dSTORM and PALM image processing on the paxillin island diameter distribution (MEF cells), and representative dSTORM image sequences of HMSiR-labeled Halo-paxillin on the basal PM of T24 cells (similar representative image sequences in MEFs are shown in** Fig. 7**). (A)** Diameter distributions of various FA-protein islands. All FA-proteins were Halo-tagged at their N-termini and labeled with HMSiR. The arrowheads indicate the mean values. After the correction for single-molecule localization precisions of 19 nm for dSTORM images of

HMSiR (Fig. S3), the true mean diameters of the islands were estimated to be 24–31 nm. **(B)** The effects of the merging distances (the threshold distances to identify the detected spots as those representing the same molecule) on the characteristics of the paxillin islands. We generally use $\sqrt{2}\cdot3$ σ and $\sqrt{2}\cdot2$ σ (σ = single-molecule localization error) for dSTORM (81 nm) and PALM images (82 nm), respectively (bottom row, which are reproduced from Fig. 5 D). In the bottom row, the results using the extended merging distances of $\sqrt{2}\cdot5$ σ = 134 nm for dSTORM and 205 nm for PALM are shown. **(C)** Typical dSTORM image sequences of HMSiR-labeled Halo-paxillin on the basal PM of live (top panel) and fixed (bottom panel) T24 cells, shown every 10 s (data acquisition of 2,500 frames) (top rows) and those of the expanded square regions superimposed by the contours of FAs (white) and paxillin islands (green) determined by the Voronoï segmentation analysis (bottom rows). The results obtained in T24 cells are similar to those obtained by using MEFs (Fig. 7 A and B).

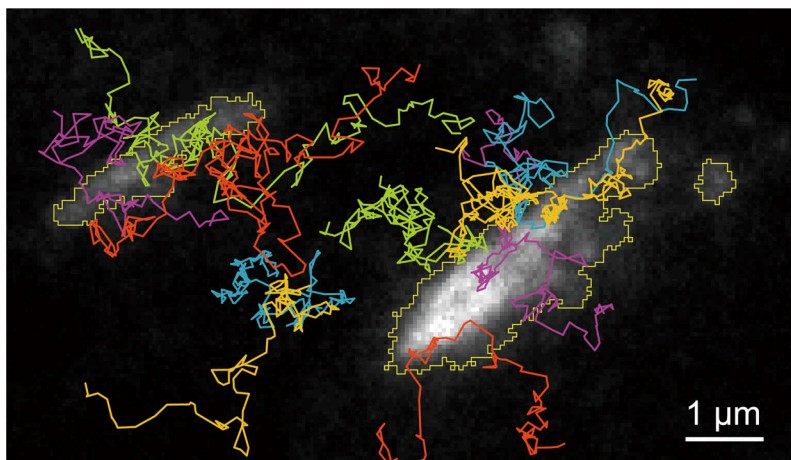

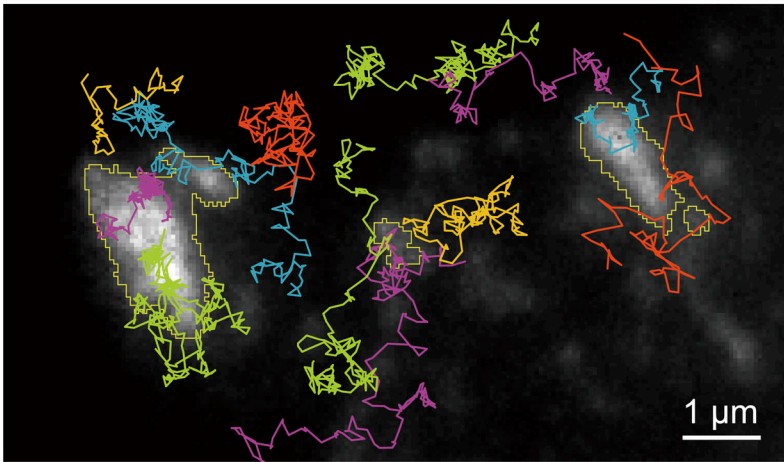

**Figure S5.** **TfR molecules diffuse within the FA, but in certain subregions, TfR is excluded or has limited diffusibility.** Typical trajectories of single TMR-labeled Halo-TfR obtained at a frame rate of 60 Hz (durations of 1.0–5.5 s) are overlaid on the simultaneously recorded TIRF image of the mGFP-paxillin (not a super-resolution image). Trajectories are colored to aid discernment where the trajectories are crowded (the color was not changed in a single trajectory). The contours of the FA regions (yellow lines) were determined by binarization, using the minimum cross entropy thresholding. Previously, our observations at slower frame rates (30–250 Hz) revealed that non-FA protein TfR molecules entered the FA region and diffused more or less freely in the fluid membrane region inside the FA (Shibata et al., 2012; Shibata et al., 2013; Tsunoyama et al., 2018). However, the presence of loose clusters of FA-protein islands shown in Figs. 6 and 7 raises the possibility that TfR molecules might not enter these island cluster domains, or if they do enter, they may not undergo free diffusion there. To address this question, we examined the movements of TMR-labeled Halo-TfR molecules located in and near the FAs at a frame rate of 60 Hz, to determine the long-range, long-time movements, such as those occurring over a time frame of 1.0–5.5 s (Fig. S5). The typical trajectories overlaid on the fluorescent paxillin images (not super-resolution) suggest that these trajectories probably represent the smeared-out (indistinct) hop diffusion of TfR, due to the slow rate of observation (60 Hz; see Fig. 8). The superimposed image in Fig. S5 implies that TfR is somewhat excluded from the areas with elevated paxillin concentrations, and when TfR molecules occasionally entered these areas, their diffusion is slowed and confined, although a quantitative analysis was beyond the scope of this study. These results indicate that the molecular diffusion within the FA is spatially quite heterogeneous, which aligns with the presence of loose FA-protein island clusters. Meanwhile, by applying the hop-diffusion fitting to 250-ms-long TfR trajectories recorded at 6 kHz in the FA (Fig. 8, A–D), we obtained the distribution of the macroscopic diffusion coefficients of individual molecules (representing the diffusion rate over several compartments rather than that within a compartment; $D_{MACRO}$; Fig. 8 F). It shows that ≈19% of TfR molecules are almost immobile in the FA, with $D_{MACRO}$ values <0.0063 µm²/s (Fig. 8 F). This result provides clear evidence for the existence of FA subdomains where TfR diffusion is suppressed. This is likely to occur in the FA subdomains where the FA-protein islands, which act as diffusion obstacles, exist at higher number densities, such as the loose clusters of FA-protein islands.

Video 1. **PALM imaging process using ultrafast data acquisition of mEos3.2-paxillin and reconstructed images, which are shown in Fig. 4, A and C.** A data acquisition rate of 1 kHz for a period of 10 s (10,000 frames) was employed for 640 × 640 pixels (35.3 × 35.3 µm²), covering most of the basal PM area, including many FAs. Fluorescent spot clusters found in the initial 30 frames (0.03 s) of the data acquisition sequence are due to the presence of mEos3.2 pre-photoconverted during the search for the cells suitable for PALM imaging, and thus they were excluded from the image reconstruction process.

Video 2. **Live-cell dSTORM of HMSiR-labeled Halo-paxillin in the FA of a MEF.** Reconstruction with sliding windows of 10 s (2,500 frames recorded at 250 Hz) every 1 s for a total period of 60 s (10×-faster replay). See Fig. 7 A.

Video 3. **The same as** Video 2 **but observed in a T24 cell.** See Fig. S4 C top.

Video 4. **Single TMR-Halo-TfR molecules diffusing in the basal PM of T24 cells and observed at a time resolution of 0.167 ms (a frame rate of 6 kHz) exhibited hop diffusion both outside and inside the FA.** Movies on the left show larger view-fields: green areas represent FAs marked by mGFP-paxillin; magenta spots represent single TfR molecules. The regions in the yellow squares are enlarged in the movies on the right, showing single-molecule TfR movements outside (top) and inside (bottom) the FA. Within the FA region, the compartment area size is smaller by a factor of ≈2 and the dwell lifetime within a compartment is ≈1.5× longer, as compared with those outside the FA region. Total observation period of 1,500 frames = 250 ms. Replayed at a 50×-slowed rate. Refer to Fig. 8.

Video 5. **A single molecule of Setau647-ACP-integrin β3 (magenta spot with a yellow trajectory) diffused into an FA region and became temporarily immobilized (arrowhead) on an FA-protein island, as identified by the live-cell PALM of mEos3.2-paxillin (green), simultaneously performed with the single-molecule integrin tracking (4-ms resolution for 1 s; replay, 8.3× slowed from real time).** See Fig. 9 A-a.

Video 6. **Same as** Video 5**, but it shows a single integrin β3 molecule immobilized on a paxillin island from time 0 till the end of the observation.** See Fig. 9 A-b.

Video 7. **An integrin β3 molecule undergoing hop diffusion in the bulk basal PM entered the channel of the archipelago of FA-protein islands in an FA, and continued hop diffusion there.** The FA-protein islands were identified by the simultaneously performed live-cell PALM of mEos3.2-paxillin (green regions; still image due to a 10-s data acquisition time = 1-ms integration time/frame × 10,000 frames). Ultrafast single-molecule imaging of a single SeTau647-Halo-integrin β3 molecule was performed at a 0.167-ms time resolution (magenta spot with a color-coded trajectory; replay, 50× slowed from real time) with a total observation period of 170 ms (1,024 frames). See Fig. 9 E and its legend.

