## [Peer Review File · The Journal of Cell Biology]

Ultrafast single-molecule imaging reveals focal adhesion nano-architecture and molecular dynamics

Takahiro Fujiwara, Taka Tsunoyama, Shinji Takeuchi, Ziya Kalay, Yosuke Nagai, Thomas Kalkbrenner, Yuri Nemoto, Limin Chen, Akihiro Shibata, Kokoro Iwasawa, Ken Ritchie, Kenichi Suzuki, and Akihiro Kusumi

Corresponding Author(s): Akihiro Kusumi, Okinawa Institute of Science and Technology Graduate University

Review Timeline:

Submission Date:	2021-10-28
Editorial Decision:	2022-01-05
Revision Received:	2023-03-25
Editorial Decision:	2023-04-27
Revision Received:	2023-05-02

Monitoring Editor: Joerg Bewersdorf

Scientific Editor: Andrea Marat

Transaction Report:

DOI: <https://doi.org/10.1083/jcb.202110162>

January 5, 2022

Re: JCB manuscript #202110162

Prof. Akihiro Kusumi
Okinawa Institute of Science and Technology
Membrane Cooperativity Unit
Onna-son
Okinawa 904-0495
Japan

Dear Prof. Kusumi,

Thank you for submitting your manuscript entitled "Focal adhesion membrane is dotted with protein islands and partitioned for molecular hop diffusion". The manuscript has been evaluated by expert reviewers, whose reports are appended below. Unfortunately, after an assessment of the reviewer feedback, our editorial decision is against publication in JCB.

While the manuscript clearly represents an impressive technical tour-de-force featuring well-executed experiments accompanied by sound theories, we feel that the level of novel mechanistic insights described in the manuscript does not meet the bar for publication in JCB. We therefore regret to inform you that we have to reject the manuscript in its current stage. However, we would be very interested in hearing your thoughts about a possible revision plan along the lines of the suggestions provided by reviewer 2 and would be happy to provide feedback whether these revisions would likely make a substantially revised manuscript acceptable for JCB. If you would like to resubmit this work to JCB, please contact the journal office to discuss an appeal of this decision or you may submit an appeal directly through our manuscript submission system. Please note that priority and novelty would be reassessed at resubmission and a revised study would be subject to further peer-review.

We understand however that the points raised by the reviewers may be more substantial than can be addressed in a typical revision period. Therefore, if you wish to expedite publication of the current data, it may be best to pursue publication at another journal. Our journal office will transfer the reviews for consideration elsewhere upon request, including to our partner journals Life Science Alliance, Journal of Cell Science, or Molecular Biology of the Cell.

Regardless of how you choose to proceed, we hope that the comments below will prove constructive as your work progresses. We would be happy to discuss the reviewer comments further once you've had a chance to consider the points raised in this letter. You can contact the journal office with any questions, cellbio@rockefeller.edu or call (212) 327-8588.

Thank you for thinking of JCB as an appropriate place to publish your work.

Sincerely,

Joerg Bewersdorf, PhD
Monitoring Editor

Andrea L. Marat, PhD
Senior Scientific Editor

Journal of Cell Biology

Reviewer #1 (Comments to the Authors (Required)):

As this is a dual submission of closely related manuscripts, I will review both together in this report. In the first manuscript, the authors developed an ultrafast digital camera module, which is then incorporated into a single-molecule and super-resolution microscopy platform that are capable of extremely high-speed single-molecule tracking, and high-frame rate PALM super-resolution microscopy imaging. The design, rationale, theoretical foundation, experimental parametrization, and characterization of the camera are described in exhaustive details. The camera system is based on an image intensifier that is fiber-coupled to a high-speed CMOS camera. Cy3 fluorophores was found to be most suitable fluorophore for high-speed single-molecule imaging but the camera speed is such that the photon emission cycle of Cy3 is slower than the capability of the camera, thus the fluorophore photophysics is claimed to be the rate limiting step, with this camera technology.

The authors then investigate diffusion characteristics of Cy3-DOPE and TfR membrane protein in plasma membrane of living cells with up to 10 kHz speed. These measurements yield finely sampled trajectories with large number of small intervals (100s-

1000s), that reveal the characteristics of hop diffusion. By perturbing actin cytoskeleton and observing the blebbing membrane, the authors showed that Hop diffusion appears to be dependent on intact actin-based membrane cytoskeleton. In conjunction, the authors also present detailed theoretical frameworks for trajectory analysis to detect hop diffusion. These results appear to be in good agreement with previous study by the authors from ~2 decades ago (also in JCB), which use gold nanoparticle and brightfield imaging. Thus, the current study corroborate the model of plasma membrane/membrane cytoskeleton organization/compartmentalization, while describing a more generalizable experimental platform and a less perturbative tagging (fluorophores vs 40-nm nanoparticles).

Next, the authors describe how the high-speed camera can also be used for Live-cell PALM, which was then applied to image caveolae and focal adhesions live cells.

In the 2nd manuscript, the author used the imaging system thus developed for a more cell biological applications, with the goals of probing membrane organization in the ventral plasma membrane as well as in focal adhesions. Using Transferring Receptor and Cy3-DOPE, the authors showed that these labelled lipid and membrane proteins exhibit similar diffusion characteristics between dorsal and ventral plasma membrane. Similar kinetics is also observed for EGFR, suggesting that these kinetics are dependent on plasma membrane compartmentalization. Subsequently, live-cell PALM is applied to image focal adhesions revealing heterogeneous nanocluster organization of paxillin (so-called 'islands'), while TfR high-speed SPT is performed to probe membrane organization in focal adhesions. Analysis of TfR trajectory indicates that plasma membrane in focal adhesion is also compartmentalized but with smaller compartment and longer dwell time. High-speed SPT of integrin beta 3 was performed which reveal complex trajectory that can periodically be immobilized at paxillin-based 'islands'.

In summary, this body of work is a technical tour de force study by established investigators in the field of single-molecule imaging/membrane biophysics that advance the state-of-the-art in fluorescence imaging capability by orders of magnitude. The experimental execution and supporting theories are sound and rigorous, while the technological advances presented should be generally useful and readily adaptable to other bioimaging modalities. The high-speed capability is likely to be game-changer in addressing a number of key biological/biophysical questions. Thus, I am in support of these works being eventually accepted for publication in JCB. That said, with the current structure of these two manuscripts, though the text itself is well-written, it is still quite challenging to digest such a large amount of information and a revision is strongly advised.

1. Of the two manuscripts, in the current organization, the first manuscript is clearly the strongest as it describes all of the novel results and major technical advances. In contrast, the key results on integrin diffusion in the 2nd manuscript is perhaps somewhat overshadowed by the authors's own previous work in 2018 Nature Chemical Biology Tsunoyama et al., that also describe similar trajectory characteristics of integrin.

As it seems that all the ground-breaking exciting results are all contained in 1st manuscript this has the side effect of depriving the 2nd manuscript of key results. At the same time, this also makes the 1st manuscript quite dense and challenging to digest.

My suggestion is to revise the 1st manuscript to focus on the camera, SPT, hop diffusion, and membrane organization. Then, the live-cell PALM sections can be moved from the 1st to the 2nd manuscript. This way, the first manuscript will be more nicely packaged as an ultrafast SPT & more biophysical membrane organization study, while the 2nd manuscript will be on fast live PALM super-resolution, their characterization, and more cell biological study. The content in both manuscripts may be better balanced and more digestible this way.

2. Some of the figure panels would benefit from more clear captions or sub-titles, so that readers do not need to go to the figure description for every panels.

3. While MINFLUX is another technique capable of high-speed fast single-molecule tracking and thus the authors may feel the need to differentiate their approach from MINFLUX, this reviewer is of the view that MINFLUX is much more of a niche technique compared to the more generalizable and modular capability presented by the fast camera in this study. Perhaps the MINFLUX discussion can be included as supplementary note instead so as not to detract from the main text.

4. One of the main weaknesses of 2nd manuscript starts with the abstract which is mostly descriptive. It is not clear from reading the abstract what is the key 'take-home' biological findings. As the authors appear to intend both manuscripts as a technological demonstration rather than a full-fledged mechanistic dissection, I would suggest to revise according to #1 above and also rewrite the abstract accordingly. Alternatively, additional biological perturbations may be needed for 2nd manuscript to dissect what factors are regulating membrane partitioning in focal adhesions. However, given that there is a vast amount of data in these two manuscripts already, this reviewer is of the view that a revision as in #1 is probably more advisable.

Reviewer #2 (Comments to the Authors (Required)):

This well-documented manuscript provides a significant extension of prior research in the fields of single-molecule membrane motility patterns and focal adhesion research. It provides further support for the archipelago model of focal adhesion organization as reported by paxillin localization, as well as confirming that apical and basal plasma membranes show similar patterns of single-molecule trajectories, including 103-109 nm compartment size with hop diffusion of both protein and lipid. The new findings include evidence that EGFR occupancy-dimerization leads to longer confinement, that transferrin receptor hop-diffusion within focal adhesions is altered with a 2-fold reduction in compartment area and enhanced dwell lifetime, suggesting

that the putative actin picket fence meshwork is finer/smaller, apparently due to fluid membrane compartmentalization rather than interference by the paxillin islands. Also interesting is quantification of beta-3 integrin immobilization, not surprisingly at the paxillin islands. The advances presented in this paper appear quite solid with impressive single-molecule tracking, but arguably often not surprising. Overall, it is not clear at present that this paper has enough truly novel content for JCB, i.e., with its present content, it appears borderline.

1. Although these findings add significantly to existing knowledge concerning membrane dynamics and focal adhesion structure, the amount of new knowledge presented in this paper seems somewhat borderline for JCB. Considering first the evaluation of dynamics at the apical versus the basal surfaces of these cells, even though the authors state at least four times that their findings were "surprising," this reviewer would have predicted that finding considering that there may be no evidence that these T4 epithelial cells have strong apical-basal polarity that might affect membrane organization. The authors should explain why they feel the results were surprising. This reviewer feels that unless the authors could use an epithelial cell with high polarity and distinct differences between apical and basal plasma membrane content, the results seem to be what would be predicted for a non-strongly polarized epithelial or fibroblastic cell in which apical and ventral membranes may not differ except at sites of focal adhesions.

2. Considering next the characterization of the fluid membrane in focal adhesions, the findings do provide this new information. However, it is not clear why the compartment size is changed. Do the authors see differences in membrane-associated actin (which might be obscured by the actin bundles inserting into focal adhesions, and thus might require TIRF microscopy)? The characterization of paxillin islands by ultrafast PALM is interesting, with a useful quantification of island diameter. Although not absolutely essential, readers in the field will wonder whether these findings are specific for just this immortalized retinal epithelial cell line and not for other more-studied cell-type focal adhesions in the literature.

3. The demonstration of integrin hop diffusion is not surprising, nor is the immobilization at sites of integrin-based adhesion in focal adhesions. What would be useful to the field is a clarification of the nature, extent, and duration of the immobilization events, which seem to involve a range of effects. Immobilization for only 0.19 s in the example shown seems quite brief, and one wonders what it signifies. An in-depth consideration of findings for integrins seems needed, even if there cannot yet be any direct correlation with paxillin due to the lack of two cameras for tracking two fluorescent labels. Basically, the integrin data seem too limited for a JCB paper, since it has been known for decades by FRAP that integrins have immobilization in focal adhesions. A more in-depth analysis might enhance interest. A specific question is whether all integrin molecules studies have a short immobilization duration in the order of seconds, or whether some key anchoring integrin molecules have considerably longer periods, which should be possible to quantify. If long-lived integrin adhesions exist, would they contribute to altering compartment size and the behavior of adjacent molecules?

March 25, 2023

Joerg Bewersdorf, Ph.D.
Monitoring Editor

Andrea L. Marat, Ph.D.
Senior Scientific Editor

Re: JCB manuscript #202110162, retitled as,
"Ultrafast single-molecule imaging reveals focal adhesion nano-architecture and molecular dynamics"

Dear Joerg and Andrea,

Thank you very much for critically reading and assessing our manuscript for publication as an Article in JCB. We would also like to thank you for obtaining the opinions of the two referees. Attached please find our revised manuscript. As we discussed with you, following your instructions, we have added significant amounts of new data and virtually rewrote this manuscript, and hereby submit a new manuscript. As we requested in our first submission, we hope that this paper can be published back-to-back with its companion Tools paper (#202110160, slightly retitled as "Development of ultrafast camera-based single fluorescent-molecule imaging for cell biology"). We are submitting these two companion manuscripts at the same time.

We have addressed all of the points raised by you and your referees in the revised manuscript, and have basically complied with all of the recommendations.

We have made large organizational changes to these two manuscripts, as recommended by Reviewer 1. Specifically, we have moved the sections describing the application of the ultrafast camera to ultrafast PALM imaging, from the revised Tools paper (companion manuscript) to the revised Article paper (this manuscript). At the same time, we have moved the sections reporting the compartmentalization of the basal plasma membrane (PM), its characteristics, and the effect of compartmentalization on EGFR diffusion before and after the ligand EGF addition, which were previously described in this Article manuscript, to the companion Tools manuscript. Instead, as recommended by Reviewer 1, this Article manuscript now includes the application of the ultrafast camera to ultrafast PALM (moved from the original Tools manuscript). Furthermore, this revised Article manuscript now includes new important experimental results about the applications of the ultrafast camera to ultrafast dSTORM and simultaneous two-color ultrafast PALM-dSTORM, as well as the results about the nano-scale architecture and dynamics of focal adhesions (FAs) obtained by these new methods. This was also motivated by the recommendations from Reviewer 2, who encouraged us to extend our studies of the focal adhesion.

As a result, we believe that the manuscript has been considerably strengthened. We would like to thank you and your reviewers again for critically reading our manuscript and providing constructive comments and recommendations. We hope that this manuscript is now acceptable for publication in *The Journal of Cell Biology*.

The main text has been comprehensively rewritten and, therefore, no highlighting was done.

Figs. 2, 3, and 4 A-F are the figures moved from the original Tools manuscript.

They are surrounded by blue rectangles. The titles of their captions are highlighted in cyan.

Newly produced figures (mostly by performing more experiments) are indicated by **green rectangles**. They are **Figs. 1 D, 4 H, 5, 6, 7, and 9 A-b and Figs. S2 - S4**, and in their captions, only their titles are **highlighted in green**. **Videos 1, 2, and 5** are new, and in their captions, their titles are **highlighted in green**.

Fig. 4 G is a revised figure using the newly-developed method for correcting for the effect of the single-molecule localization error on the diameter estimation of paxillin-enriched islands, and thus shown by an **orange rectangle**.

Fig. 1's caption had to be rewritten entirely due to the organizational changes of these two manuscripts, and so it is **highlighted in yellow**.

Figs. 1, 8, and 9 are basically the same figures used in the original Article manuscript, with small modifications indicated by **green rectangles**. The captions have been revised accordingly and are indicated by **yellow highlighting**.

Our point-by-point responses to your reviewers' comments are provided on the following pages.

Sincerely yours,

Aki (Akihiro Kusumi)

Professor

Membrane Cooperativity Unit

Okinawa Institute of Science and Technology Graduate University (OIST)

e-mail: akihiro.kusumi@oist.jp

Reviewer #1 (Comments to the Authors (Required)):

Thank you very much for carefully reading our manuscripts and for your kind and constructive comments. We addressed all your comments in our rebuttal for the revised Tools manuscript, and thus will refrain from repeating them here.

Please note the following ways in which we highlighted the new results and the results moved from the previous Tools manuscript to this revised Article manuscript.

The main text has been comprehensively rewritten and, therefore, no highlighting was done.

Figs. 2, 3, and 4 A-F are the figures moved from the original Tools manuscript.

They are surrounded by **blue rectangles**. The titles of their captions are **highlighted in cyan**.

Newly produced figures (mostly by performing more experiments) are indicated by **green rectangles**. They are **Figs. 1 D, 4 H, 5, 6, 7, and 9 A-b and Figs. S2 - S4**, and in their captions, only their titles are **highlighted in green**. **Videos 1, 2, and 5** are new, and in their captions, their titles are **highlighted in green**.

Fig. 4 G is a revised figure using the newly-developed method for correcting for the effect of the single-molecule localization error on the diameter estimation of paxillin-enriched islands, and thus shown by an **orange rectangle**.

Fig. 1's caption had to be rewritten entirely due to the organizational changes of these two manuscripts, and so it is **highlighted in yellow**.

Figs. 1, 8, and 9 are basically the same figures used in the original Article manuscript, with small modifications indicated by **green rectangles**. The captions have been revised accordingly and are indicated by **yellow highlighting**.

Reviewer #2 (Comments to the Authors (Required)):

This well-documented manuscript provides a significant extension of prior research in the fields of single-molecule membrane motility patterns and focal adhesion research. It provides further support for the archipelago model of focal adhesion organization as reported by paxillin localization, as well as confirming that apical and basal plasma membranes show similar patterns of single-molecule trajectories, including 103-109 nm compartment size with hop diffusion of both protein and lipid. The new findings include evidence that EGFR occupancy-dimerization leads to longer confinement, that transferrin receptor hop-diffusion within focal adhesions is altered with a 2-fold reduction in compartment area and enhanced dwell lifetime, suggesting that the putative actin picket fence meshwork is finer/smaller, apparently due to fluid membrane compartmentalization rather than interference by the paxillin islands. Also interesting is quantification of beta-3 integrin immobilization, not surprisingly at the paxillin islands. The advances presented in this paper appear quite solid with impressive single-molecule tracking, but arguably often not surprising. Overall, it is not clear at present that this paper has enough truly novel content for JCB, i.e., with its present content, it appears borderline.

Thank you very much for critically reading our manuscript and for your kind and constructive feedback.

For strengthening this Article manuscript, we developed new methodologies for ultrafast live-cell dSTORM and simultaneous two-color ultrafast live-cell PALM and dSTORM, based on the developed ultrafast camera

(**Fig. S3**), and applied these techniques to reveal the molecular organization and dynamics of the focal adhesion (FA) (**Figs. 4 G, H, 5, 6, 7, and 9 A-b and Figs. S2, S4, and S5** and related text).

Following the recommendations by Reviewer 2, we extensively applied the developed ultrafast techniques, especially the simultaneous two-color PALM-dSTORM imaging, to further investigations of the FA, and found that FA proteins, including paxillin, integrins $\beta 1$ and $\beta 3$, talin, FAK, and vinculin, often assemble into nano-clusters. These clusters are frequently ≥ 13 nm in diameter and contain ≥ 6 copies of one of the FA-protein species, and we refer to these greater nano-clusters as FA-protein islands. The mean island diameter is 29~32 nm in mouse embryonic fibroblasts (MEFs) (**Figs. 5 D, F, and 6 C; Figs. S2 and S4 A**), which are the values obtained after the correction for single-molecule localization accuracies (**Fig. S2**). This correction has never been conducted previously. These estimates are generally consistent with the diameters of adhesion particles found by cryo-electron microscopy (Patla et al., 2010) and integrin nanoclusters identified by super-resolution microscopy (Changede et al., 2019). However, the broad distribution of the FA-protein-island diameters, ranging from 13 to 100 nm, is quite new and we believe this is a characteristic feature of the FA-protein islands. The broad diversities were not only found in the sizes but also implied in the protein compositions and molecular stoichiometries of the FA-protein islands. Such diversities might be important for the functions of FAs, as they must respond to various force levels, force loading rates, and the diverse characteristics of the extracellular matrix.

Furthermore, we discovered that the FA-protein islands are not distributed homogeneously within the FA, but rather form loose island clusters of ≈ 320 nm in diameter (**Fig. 1 D-d, archipelago model of FA-protein island clusters and oligomers; Figs. 5 F-H, and 6 C, D**). Our analysis, using paxillin, indicated that these 320-nm loose island clusters function as the units for recruiting paxillin to the FA (**Fig. 7 A-D**). The dynamics of the recruitment and exchange of paxillin molecules could never have been observed without applying ultrafast dSTORM to live cells (**Fig. 7 A, B**).

Please note that, as per the recommendations by Reviewer 1, we have moved the sections describing the compartmentalization of the *basal* plasma membrane (PM) and the hop diffusion of Cy3-DOPE, transferrin receptor, and EGF receptor before and after the addition of the ligand EGF, from the Article manuscript (this manuscript) to the Tools manuscript. Now, all of the sections including PALM observations have been moved from the original Tools manuscript to this revised Article manuscript.

The compartmentalization of the FA's fluid membrane region and the hop diffusion of transferrin and integrin molecules there remain in the Article manuscript. Thus, virtually all the sections related to FA organization and molecular dynamics are now included in this Article manuscript.

1. Although these findings add significantly to existing knowledge concerning membrane dynamics and focal adhesion structure, the amount of new knowledge presented in this paper seems somewhat borderline for JCB. Considering first the evaluation of dynamics at the apical versus the basal surfaces of these cells, even though the authors state at least four times that their findings were "surprising," this reviewer would have predicted that finding considering that there may be no evidence that these T4 epithelial cells have strong apical-basal polarity that might affect membrane organization. The authors should explain why they feel the results were surprising. This reviewer feels that unless the authors could use an epithelial cell with high polarity and distinct differences between apical and basal plasma membrane content, the results seem to be what would be predicted for a non-strongly polarized epithelial or fibroblastic cell in which apical and ventral membranes may not differ except at sites of focal adhesions.

We have moved the sections describing the compartmentalization of the basal PM and the hop diffusion of Cy3-DOPE, transferrin receptor, and EGF receptor before and after the addition of the ligand EGF, from the original Article manuscript to the revised Tools manuscript. However, we will address this point raised by Reviewer 2 here.

Prior to this study, we had not been able to perform ultrafast observations of PM molecules in the basal PM, even using 40-nm gold particles (using ultrafast bright-field microscopy; this method using 40-nm gold particles as probes was the only way to perform ultrafast single-particle tracking at ≥ 10 kHz), because these gold particles cannot enter the space between the basal PM and the coverslip. With the development of ultrafast single fluorescent-molecule imaging (SFMI) using the < 1 nm fluorescent probes described in this study (Tools manuscript), scientists can now perform ultrafast observations of molecular dynamics in the basal PM. The question of whether the compartmentalization and hop diffusion in the basal PM are different from those in the apical PM was frequently raised when we presented our ultrafast single-particle tracking data of hop diffusion in the apical PM at various meetings. Therefore, we would say that the questions of whether the basal PM is compartmentalized in the same way as the apical PM and whether molecules in the basal PM undergo hop diffusion similar to those in the apical PM have been long-term enigmas, even in non-polarized cells.

Indeed, we were surprised to find the almost identical characteristics of the basal and apical PMs because, even in non-polarized cells, the architecture of the basal PM facing the substrate (coverslip) within distances less than 40 nm might be quite different from that of the apical PM, which only faces the cell culture medium (the statement about the gap size less than 40 nm is based on the observations that gold particles of 40 nm in diameter do not bind to transferrin receptors and phospholipids located in the basal PM, whereas they bind to those in the apical PM, and that when colloidal-gold-bound membrane molecules in the apical PM reach the cell edge, which is the interface between the apical and basal PMs, they generally cannot enter the basal PM or they stop diffusing after they slightly enter the basal PM). This point is now described in the **Discussion of the revised Tools manuscript (the last paragraph on p. 22)**. Meanwhile, we deleted the expression "surprising" from the manuscript except for one instance, where we want to explain why we used another method to confirm the results of the direct hop diffusion observations (**the last paragraph on p. 18**).

The organization of the apical PM of strongly polarized epithelial cells would be entirely different from that of the basal PM or that of non-polarized cells, due to the presence of dense microvilli. Therefore, although the dynamics of PM molecules there would be quite interesting, it is beyond the scope of the present research. Here, we believe we should concentrate on the basal PM, which is our main subject matter.

2. Considering next the characterization of the fluid membrane in focal adhesions, the findings do provide this new information. However, it is not clear why the compartment size is changed. Do the authors see differences in membrane-associated actin (which might be obscured by the actin bundles inserting into focal adhesions, and thus might require TIRF microscopy)?

Unfortunately, we cannot experimentally show that the compartment boundaries in the FAs' fluid membrane regions are produced by the actin filaments as in the bulk PM, as described in the third paragraph of "TfR undergoes hop diffusion inside the FA" (**p. 17**); i.e., "attempts to directly observe the

effects of actin depolymerizing drugs failed, because at the concentrations where their effects were detectable, the cells became round and some did not survive”.

The observation of the PM-associated actin filaments using optical microscopy such as TIRF microscopy, has been difficult (in the literature, one can find papers stating that they observed individual actin filaments underlying the PM, but the evidence was insufficient; by the way, throughout the present research, we used TIRF microscopy, particularly for the observations described in this Article manuscript, and so this is not an issue for us at all). The largest problem is that cortical actin filaments densely exist three-dimensionally near the PM and by just observing the images (even using EM images), it is impossible to tell which actin filaments (and which parts of single actin filaments) are associated with the PM. This is why we needed to use electron tomography to perform the 3D reconstruction of the structures on the PM cytoplasmic surface, in order to identify the actin filaments that are located within ≈ 8 nm from the PM inner surface (Morone et al., JCB, 2006). We tried to use this method for identifying actin filaments on the FA cytoplasmic surface, but due to the presence of very dense actin filament bundles, we could not obtain the actin structures there.

Therefore, we only speculate that the actin filament meshes on the FA cytoplasmic surface are finer (**p. 18, the second paragraph from the bottom**).

The characterization of paxillin islands by ultrafast PALM is interesting, with a useful quantification of island diameter. Although not absolutely essential, readers in the field will wonder whether these findings are specific for just this immortalized retinal epithelial cell line and not for other more-studied cell-type focal adhesions in the literature.

To address this point raised by Reviewer 2, we performed new investigations using mouse embryonic fibroblasts (MEFs), which have been widely employed in FA studies. Specifically, we extensively used paxillin knock-out MEFs rescued with mEos3.2-paxillin (cloned; **pp. 10-11**). The new results are described in **Figs. 5-7 and related text**. In terms of the diameters of the paxillin islands, they are quite similar between the human T24 epithelial cells and MEFs (**Figs. 4 G and 5 D**).

3. The demonstration of integrin hop diffusion is not surprising, nor is the immobilization at sites of integrin-based adhesion in focal adhesions. What would be useful to the field is a clarification of the nature, extent, and duration of the immobilization events, which seem to involve a range of effects.

When we presented these results in scientific meetings, quite a few researchers exhibited their surprise (perhaps with some doubt) and interest in our findings that the FA's fluid membrane regions, which represent the majority of the FA area, are compartmentalized. Many cell biology textbooks now show our model of the PM compartmentalized by actin filaments, but to the best of our knowledge, none show compartmentalized FAs or those dotted with FA-protein islands (archipelago architecture).

Immobilization of integrins at the paxillin islands might not be surprising, but we believe that somebody must unequivocally show it to make it common, solid knowledge, rather than keeping it as an educated guess, speculation, and implication.

The nature, extent, and duration of the immobilization events for both integrins $\beta 3$ and $\beta 1$ have already been described in our previous paper (Tsunoyama et al., *Nat. Chem. Biol.*, 2018), but in the previous work, we could not determine where the immobilization events occurred in the FA. Therefore, we performed these experiments in the present study, and found that the integrins are anchored (immobilized) on paxillin islands (and so, probably the FA-protein islands). This issue was pointed out by Reviewer 1 in her/his point 1, "the key results on integrin diffusion in the 2nd manuscript is perhaps somewhat overshadowed by the authors's own previous work in 2018 Nature Chemical Biology Tsunoyama et al."

Immobilization for only 0.19 s in the example shown seems quite brief, and one wonders what it signifies. An in-depth consideration of findings for integrins seems needed, even if there cannot yet be any direct correlation with paxillin due to the lack of two cameras for tracking two fluorescent labels.

Basically, the integrin data seem too limited for a JCB paper, since it has been known for decades by FRAP that integrins have immobilization in focal adhesions. A more in-depth analysis might enhance interest.

A specific question is whether all integrin molecules studies have a short immobilization duration in the order of seconds, or whether some key anchoring integrin molecules have considerably longer periods, which should be possible to quantify. If long-lived integrin adhesions exist, would they contribute to altering compartment size and the behavior of adjacent molecules?

Immobilization for 0.19 s is just one typical example (the image sequence is reproduced here in **Fig. 9 A-a** in the revised manuscript). We selected this image sequence because it nicely shows a typical sequence of events for an integrin $\beta 3$ molecule, starting from the entry from the bulk PM into the FA region, to immobilization, resuming diffusion, and finally moving away from the FA. To avoid confusion, in the new **Fig. 6 A-b** in the revised manuscript, we now show another typical case where integrin $\beta 3$ is immobilized in the FA at time 0, and remained immobilized till the end of the recording.

In addition, in the **Discussion** section (**second paragraph on p. 24**), we added the following sentences to clarify this point and its significance. "Previously, using the video-rate SFMI, Tsunoyama et al. (2018) found that both integrins $\beta 3$ and $\beta 1$ undergo temporary (< 80-s) immobilizations in the FA, but perform distinct functions in the FA formation, maintenance, and disintegration (also refer to the following publications: Roca-Cusachs et al., 2009; Rossier et al., 2012; Schiller et al., 2013). In the growing phase, integrin $\beta 1$ initially exhibited longer immobilizations, while integrin $\beta 3$ did so when the FA was in the mature steady phase. In the process of FA disintegration, the prolonged immobilizations of integrin $\beta 1$ were reduced first, while integrin $\beta 3$ continued exhibiting longer immobilizations for some time. In the present research, we unequivocally demonstrated that these immobilizations occur at the FA-protein islands, clearly demonstrating the functional importance of the FA-protein islands (**Fig. 9**)."

Furthermore, in the **Results** section directly describing our observations, to avoid confusion, we added the following paragraph (**first paragraph on p. 20**). "Tsunoyama et al. (2018) previously found that integrins $\beta 3$ and $\beta 1$ undergo temporary immobilizations in the FA in the time scales of 0.66 s \sim 79 s and 0.5 \sim 43 s, respectively (exponential lifetimes), and by using integrin $\beta 3$ we have shown here that 68% of these temporary immobilizations occur on the paxillin-enriched islands. The remaining 32% of the temporary immobilization might occur in other FA-protein islands containing <6 paxillin copies. Therefore, we conclude that the cell linkage to the extracellular matrix primarily occurs through the integrin molecules

mediating the linkage of the FA-protein islands to the extracellular matrix. Although each integrin molecule might contribute to the binding for periods on the order of one to several 10s of seconds, multiple integrin molecules would be dynamically and continually recruited to the FA-protein islands, exchanging with those located outside the FA-protein islands. As a result, the FA-protein islands would remain linked to the extracellular matrix for much longer durations. The dynamic linkage of the integrins via the paxillin islands would facilitate the rapid control of FA formation and disintegration (Shibata et al., 2012; Rossier et al., 2012; Tsunoyama et al., 2018; Orré et al., 2021).”

Taken together, we believe that these results represent useful advances in our understanding of integrin dynamics compared to previous studies that relied on FRAP, due to the higher spatial resolution and direct observations of single-molecule behaviors (their distributions), rather than simple descriptions of the average behavior (which might be confounded by many diverse events).

To contribute further to FA research, we have generated new data that are presented in **Figs. 5-7 and related text** (we already gave some descriptions of these new data in our response to Reviewer 2’s general comment).

Please note the following way in which we highlighted the new results and the results moved from the previous Tool’s manuscript to this revised Article manuscript.

The main text has been comprehensively rewritten and, therefore, no highlighting was done.

Figs. 2, 3, and 4 A-F are the figures moved from the original Tools manuscript.

They are surrounded by **blue rectangles**. The titles of their captions are **highlighted in cyan**.

Newly produced figures (mostly by performing more experiments) are indicated by **green rectangles**. They are **Figs. 1 D, 4 H, 5, 6, 7, and 9 A-b and Figs. S2 - S4**, and in their captions, only their titles are **highlighted in green**. **Videos 1, 2, and 5** are new, and in their captions, their titles are **highlighted in green**.

Fig. 4 G is a revised figure using the newly-developed method to correct for the effect of the single-molecule localization error on the diameter estimation of paxillin-enriched islands, and thus is shown by an **orange rectangle**.

Fig. 1’s caption had to be rewritten entirely due to the organizational changes of these two manuscripts, and so it is **highlighted in yellow**.

Figs. 1, 8, and 9 are basically the same figures used in the original Article manuscript, with small modifications indicated by **green rectangles**. The captions have been revised accordingly and are indicated by **yellow highlighting**.

April 27, 2023

RE: JCB Manuscript #202110162R

Prof. Akihiro Kusumi
Okinawa Institute of Science and Technology Graduate University
Membrane Cooperativity Unit
Onna-son
Okinawa 904-0495
Japan

Dear Prof. Kusumi:

Thank you for submitting your revised manuscript entitled "Ultrafast single-molecule imaging reveals focal adhesion nano-architecture and molecular dynamics". The reviewers all now support publication so we would be happy to publish your paper in JCB pending final revisions necessary to meet our formatting guidelines (see details below). In your final revision, please be sure to address reviewer #1's final minor concerns.

A. MANUSCRIPT ORGANIZATION AND FORMATTING:

- 1) Text limits: Character count for Articles is < 40,000, not including spaces. Count includes abstract, introduction, results, discussion, and acknowledgments. Count does not include title page, figure legends, materials and methods, references, tables, or supplemental legends.
- 2) Figures limits: Articles may have up to 10 main text figures.
- 3) Figure formatting: Scale bars must be present on all microscopy images, including inset magnifications. Molecular weight or nucleic acid size markers must be included on all gel electrophoresis.
- 4) Statistical analysis: Error bars on graphic representations of numerical data must be clearly described in the figure legend. The number of independent data points (n) represented in a graph must be indicated in the legend. Statistical methods should be explained in full in the materials and methods. For figures presenting pooled data the statistical measure should be defined in the figure legends. Please also be sure to indicate the statistical tests used in each of your experiments (either in the figure legend itself or in a separate methods section) as well as the parameters of the test (for example, if you ran a t-test, please indicate if it was one- or two-sided, etc.). Also, if you used parametric tests, please indicate if the data distribution was tested for normality (and if so, how). If not, you must state something to the effect that "Data distribution was assumed to be normal but this was not formally tested."
- 5) Abstract and title: The abstract should be no longer than 160 words and should communicate the significance of the paper for a general audience. The title should be less than 100 characters including spaces. Make the title concise but accessible to a general readership.
- 6) Materials and methods: Should be comprehensive and not simply reference a previous publication for details on how an experiment was performed. Please provide full descriptions in the text for readers who may not have access to referenced manuscripts.
- 7) Please be sure to provide the sequences for all of your primers/oligos and RNAi constructs in the materials and methods. You must also indicate in the methods the source, species, and catalog numbers (where appropriate) for all of your antibodies. Please also indicate the acquisition and quantification methods for immunoblotting/western blots.
- 8) Microscope image acquisition: The following information must be provided about the acquisition and processing of images:
 - a. Make and model of microscope
 - b. Type, magnification, and numerical aperture of the objective lenses
 - c. Temperature
 - d. Imaging medium
 - e. Fluorochromes

- f. Camera make and model
- g. Acquisition software
- h. Any software used for image processing subsequent to data acquisition. Please include details and types of operations involved (e.g., type of deconvolution, 3D reconstitutions, surface or volume rendering, gamma adjustments, etc.).

9) References: There is no limit to the number of references cited in a manuscript. References should be cited parenthetically in the text by author and year of publication. Abbreviate the names of journals according to PubMed. Supplemental references are not permitted.

10) Supplemental materials: There are strict limits on the allowable amount of supplemental data. Articles may have up to 5 supplemental figures. Please also note that tables, like figures, should be provided as individual, editable files. A summary of all supplemental material should appear at the end of the Materials and methods section.

13) ORCID IDs: ORCID IDs are unique identifiers allowing researchers to create a record of their various scholarly contributions in a single place. At resubmission of your final files, please consider providing an ORCID ID for as many contributing authors as possible.

Please note that JCB now requires authors to submit Source Data used to generate figures containing gels and Western blots with all revised manuscripts. This Source Data consists of fully uncropped and unprocessed images for each gel/blot displayed in the main and supplemental figures. Since your paper includes cropped gel and/or blot images, please be sure to provide one Source Data file for each figure that contains gels and/or blots along with your revised manuscript files. File names for Source Data figures should be alphanumeric without any spaces or special characters (i.e., SourceDataF#, where F# refers to the associated main figure number or SourceDataFS# for those associated with Supplementary figures). The lanes of the gels/blots should be labeled as they are in the associated figure, the place where cropping was applied should be marked (with a box), and molecular weight/size standards should be labeled wherever possible.

Journal of Cell Biology now requires a data availability statement for all research article submissions. These statements will be published in the article directly above the Acknowledgments. The statement should address all data underlying the research presented in the manuscript. Please visit the JCB instructions for authors for guidelines and examples of statements at (<https://rupress.org/jcb/pages/editorial-policies#data-availability-statement>).

B. FINAL FILES:

****It is JCB policy that if requested, original data images must be made available to the editors. Failure to provide original images upon request will result in unavoidable delays in publication. Please ensure that you have access to all original data images prior to final submission.****

****The license to publish form must be signed before your manuscript can be sent to production. A link to the electronic license to publish form will be sent to the corresponding author only. Please take a moment to check your funder requirements before choosing the appropriate license.****

Thank you for this interesting contribution, we look forward to publishing your paper in Journal of Cell Biology.

Sincerely,

Joerg Bewersdorf, PhD
Monitoring Editor

Andrea L. Marat, PhD
Senior Scientific Editor

Journal of Cell Biology

Reviewer #1 (Comments to the Authors (Required)):

Comments

In this revision the author has substantially revised and reorganized their manuscript, presented additional technical characterizations and analysis, and expanded the investigations to additional adhesion proteins. In particular, in addition to demonstrating the impressive super-resolution imaging capability enabled by their camera, they also perform systematic characterization of the camera performance with respect to fluorophore lifetime, excitation intensity, and localization accuracy analysis which I believe will help serve as a useful instructive example for the field.

The expanded analysis included the investigation into the nanocluster organization of important focal adhesion proteins including paxillin, talin, FAK, and integrin b1 and b3. These results in a better definition of the nanocluster organization of these proteins, as well as potential differences between proteins. Although the analysis was performed in the presence of endogenous proteins which complicate stoichiometric analysis, they carefully characterize the expression level and take the necessary precaution in the interpretation. Intriguing observation that they made include the potential hierarchical organization of nanoclusters, i.e. sub-micron scale (300 nm) of the ~30 nm nanoclusters which they infer from the correlation analysis of proteins such as talin, FAK, and paxillin. These additional data augmented the previous data on TfR diffusion in the membrane which further corroborate granular organizations of focal adhesions.

Taken together, in my opinion, that manuscript is a tour de force body of work in cellular biophysics, which advance technical ability, and present new conceptual advances on how adhesions are organized at the molecular scale. As the notion of nanocluster organization as subunits of adhesive structures are becoming appreciated in various systems beyond integrins, and are now being reported in cadherins, notch receptors etc., this study is timely. While I would like to see additional molecular insights into the observed nanoscale organization, given the lengths and the highly technical scope of the current study, I believe further molecular dissection may best be left for follow-up studies. On the whole, I am largely satisfied with the revision and support its acceptance to the Journal of Cell Biology. I believe this study will prove foundational in the long run and encourage the authors to disseminate this capability and expand their studies to explore a broader range of proteins in the future.

Minor comments:

Figure 2A, and similar. It may help to put the Laser power on the left rather than on the right of the figures

The supplementary movies mostly show close up view of adhesions. Since the authors discuss large field-of-view live PALM

imaging as one of the major capabilities, it would be helpful to include example movies to demonstrate this.

Towards open data sharing, the authors may consider sharing their representative raw data on platform such as <https://shareloc.xyz/#/>

Ouyang, Wei, et al. "ShareLoc-an open platform for sharing localization microscopy data." *Nature Methods* (2022): 1-3.

Reviewer #2 (Comments to the Authors (Required)):

The authors of this resubmitted manuscript have responded by revisions that have substantially expanded the data, specific conclusions, and overall interest of this companion paper to a Tools submission. The findings provide unprecedented detail and understanding of the nano-scale organization and dynamics of focal adhesions. Although various publications over recent years have been providing more and more insight into the complex substructure and internal dynamics of focal adhesions, this study represents a next step in understanding, with intriguing details about the heterogeneity of paxillin and other focal adhesion protein islands, their organization into groupings ("archipelagos") restricted by putative actin fences that are smaller than bulk plasma membrane actin barriers, and insight into a range of integrin associations that are remarkably short and variable. Overall, the substantial additions to this study, including the use of ultra-fast versions of PALM and dSTORM with dual color information in combination with single-molecule tracking make this paper a major new contribution to the field. Unusually, this normally very critical reviewer could not identify any remaining points of concern. Publication is recommended with high priority.

May 4, 2023

Joerg Bewersdorf, Ph.D.
Monitoring Editor

Andrea L. Marat, Ph.D.
Senior Scientific Editor

Re: JCB manuscript #202110162RR, entitled,
"Ultrafast single-molecule imaging reveals focal adhesion nano-architecture and molecular dynamics"

Dear Joerg and Andrea,

Thank you very much for accepting our manuscripts (this manuscript and #202110160RR). We are extremely glad to know that these manuscripts have been accepted. We now strongly hope that these manuscripts will be published back-to-back in the Journal of Cell Biology.

We have addressed the minor points raised by Reviewer 1 in the revised manuscript (#202110162RR) as follows.

Minor comments:

Figure 2A, and similar. It may help to put the Laser power on the left rather than on the right of the figures.

Done for Figures 2A and S3B.

The supplementary movies mostly show close up view of adhesions. Since the authors discuss large field-of-view live PALM imaging as one of the major capabilities, it would be helpful to include example movies to demonstrate this.

The new video has been produced addressing this point. Please see new Video 1.

Towards open data sharing, the authors may consider sharing their representative raw data on platform such as <https://shareloc.xyz/#/>
Ouyang, Wei, et al. "ShareLoc-an open platform for sharing localization microscopy data." *Nature Methods* (2022): 1-3.

We have not placed the raw data on this platform, but will seriously consider doing this.

Thank you very much again for accepting our manuscripts.

Sincerely yours,

Aki (Akihiro Kusumi)

Professor

Membrane Cooperativity Unit

Okinawa Institute of Science and Technology Graduate University (OIST)

e-mail: akihiro.kusumi@oist.jp